# The human ribosome modulates multidomain protein biogenesis by delaying cotranslational domain docking

Grant A. Pellowe ®[1,5], Tomas B. Voisin ®[1], Laura Karpauskaite[1], Sarah L. Maslen[2], Alžběta Roeselová ®[1], J. Mark Skehel ®[2], Chloe Roustan[3], Roger George[3], Andrea Nans ®[3], Svend Kjær ®[3], Ian A. Taylor ®[4] & David Balchin ®[1] ✉

Proteins with multiple domains are intrinsically prone to misfold, yet fold efficiently during their synthesis on the ribosome. This is especially important in eukaryotes, where multidomain proteins predominate. Here we sought to understand how multidomain protein folding is modulated by the eukaryotic ribosome. We used hydrogen–deuterium exchange mass spectrometry and cryo-electron microscopy to characterize the structure and dynamics of partially synthesized intermediates of a model multidomain protein. We find that nascent subdomains fold progressively during synthesis on the human ribosome, templated by interactions across domain interfaces. The conformational ensemble of the nascent chain is tuned by its unstructured C-terminal segments, which keep interfaces between folded domains in dynamic equilibrium until translation termination. This contrasts with the bacterial ribosome, on which domain interfaces form early and remain stable during synthesis. Delayed domain docking may avoid interdomain misfolding to promote the maturation of multidomain proteins in eukaryotes.

Most proteins contain more than one domain[1]. Although functionally advantageous, combining domains into a single polypeptide often compromises refoldability[2–11], necessitating cellular mechanisms tailored to multidomain protein biogenesis. A fundamental solution is to couple folding to translation on the ribosome. Cotranslational folding shapes protein maturation in several ways. Vectorial synthesis can separate folding into elementary steps[12–14], while interactions with the ribosome surface can destabilize native folds[15–17] and tethering to the ribosome can stabilize unique folding intermediates[18–20]. In the case of multidomain proteins, cotranslational folding has been suggested to avoid misfolding by favoring sequential folding of individual domains[21,22]. Nonetheless, domains do not necessarily behave as independent units during cotranslational folding. Detailed studies of the multidomain bacterial protein EF-G have revealed complex interdependencies between folding domains on the ribosome[23–26]. How domain–domain interactions are modulated by the ribosome is poorly understood.

Multidomain proteins are more frequent in eukaryotic compared to prokaryotic proteomes[27], indicating increased evolutionary pressure to optimize their biogenesis. Indeed, several multidomain proteins were shown to fold more efficiently in eukaryotes than bacteria[28]. Experiments in cell-free translation systems have further suggested that eukaryotic and prokaryotic ribosomes differ fundamentally in their ability to promote multidomain protein folding[22,29]. Although they are similar overall, bacterial and eukaryotic ribosomes differ in the architecture of their exit tunnels[30,31] and several eukaryote-specific ribosomal proteins cluster near the exit port where nascent polypeptides emerge into the cytosol[32]. Whether species-specific features of ribosomes directly influence cotranslational folding is not clear.

[1]Protein Biogenesis Laboratory, The Francis Crick Institute, London, UK. [2]Proteomics Science Technology Platform, The Francis Crick Institute, London, UK. [3]Structural Biology Science Technology Platform, The Francis Crick Institute, London, UK. [4]Macromolecular Structure Laboratory, The Francis Crick Institute, London, UK. [5]Present address: Aston Institute for Membrane Excellence, Aston University, Birmingham, UK. ✉e-mail: david.balchin@crick.ac.uk

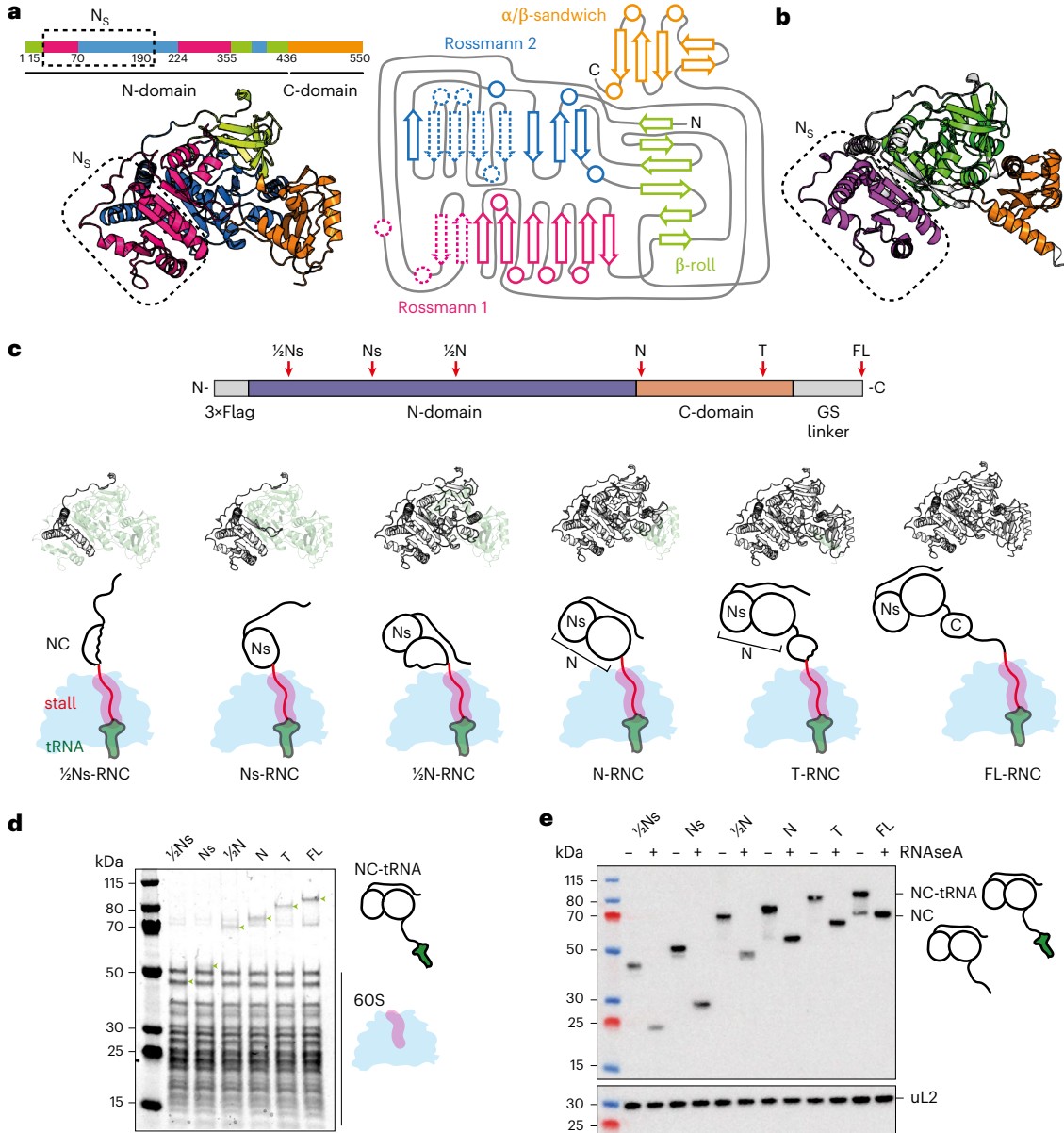

**Fig. 1 | FLuc RNCs. a**, Structure and domain architecture of FLuc. The structure (left) is predicted by AlphaFold2 (refs. 63,64) and the topology map (right) is based on PDB 1LCI (ref. 38). Elements corresponding to the Ns domain are indicated by dashed lines on the topology map. Subdomains constituting distinct folds are colored green (β-roll), magenta (Rossmann fold 1), blue (Rossmann fold 2) or orange (α/β sandwich). **b**, AlphaFold2-predicted structure of FLuc, colored by InterPro-annotated domains. **c**, Design of FLuc stalling constructs. Residue numbers: ½N$_S$, 1–123; N$_S$, 1–208; ½N, 1–388; N, 1–458; T, 1–528; FL, 1–550 + 50 aa. Positions at which XBP1u+ was inserted are indicated by red arrows. **c**, Schematic diagram of RNCs. The fraction of FLuc synthesized in each RNC is colored white. **d**, Coomassie-stained SDS–PAGE of purified RNCs. Bands corresponding to 60S ribosomal proteins and the NC linked to peptidyl-tRNA (green arrows) are indicated. Experiments were repeated using three independent protein purifications, with similar results. **e**, Anti-FLuc immunoblot of purified RNCs. RNase and EDTA treatment confirmed that the NCs are covalently linked to peptidyl-tRNA. Bottom, immunoblot against ribosomal protein uL2 as a loading control. Experiments were repeated using three independent protein purifications, with similar results (Extended Data Fig. 1).

To study how eukaryotic ribosomes shape multidomain protein folding, we focused on firefly luciferase (FLuc) as a nascent chain (NC) model. FLuc is a conformationally labile two-domain protein with a complex topology, the efficient biogenesis of which strongly depends on the cellular environment. FLuc refolds extremely slowly ($t_{1/2} \approx 75$ min) and inefficiently (yield: 10–50%) from denaturant in vitro and populates aggregation-prone intermediates[3,33]. Although the Hsp70 chaperone system substantially accelerates FLuc folding ($t_{1/2} \approx 4$ min)[3,34], FLuc maturation is optimal when coupled to translation on the eukaryotic ribosome, where its folding is synchronized with synthesis ($t_{1/2} \approx 1$ min)[21,35,36].

Understanding why cotranslational folding is efficient requires molecular insight into NC conformation. However, the structural dynamics of ribosome-tethered NCs are challenging to resolve[37] and eukaryotic NCs have not been characterized because of the absence of suitable approaches. Here, we extend peptide-resolved hydrogen–deuterium exchange mass spectrometry (HDX-MS) to human ribosome–NC complexes (RNCs)[19]. In combination with cryo-electron microscopy (cryo-EM) and orthogonal biochemical approaches, this allowed us to describe the local conformational landscape and ribosome contacts of NCs at specific points in their synthesis. We find that FLuc (sub)domains fold interdependently on the human ribosome.

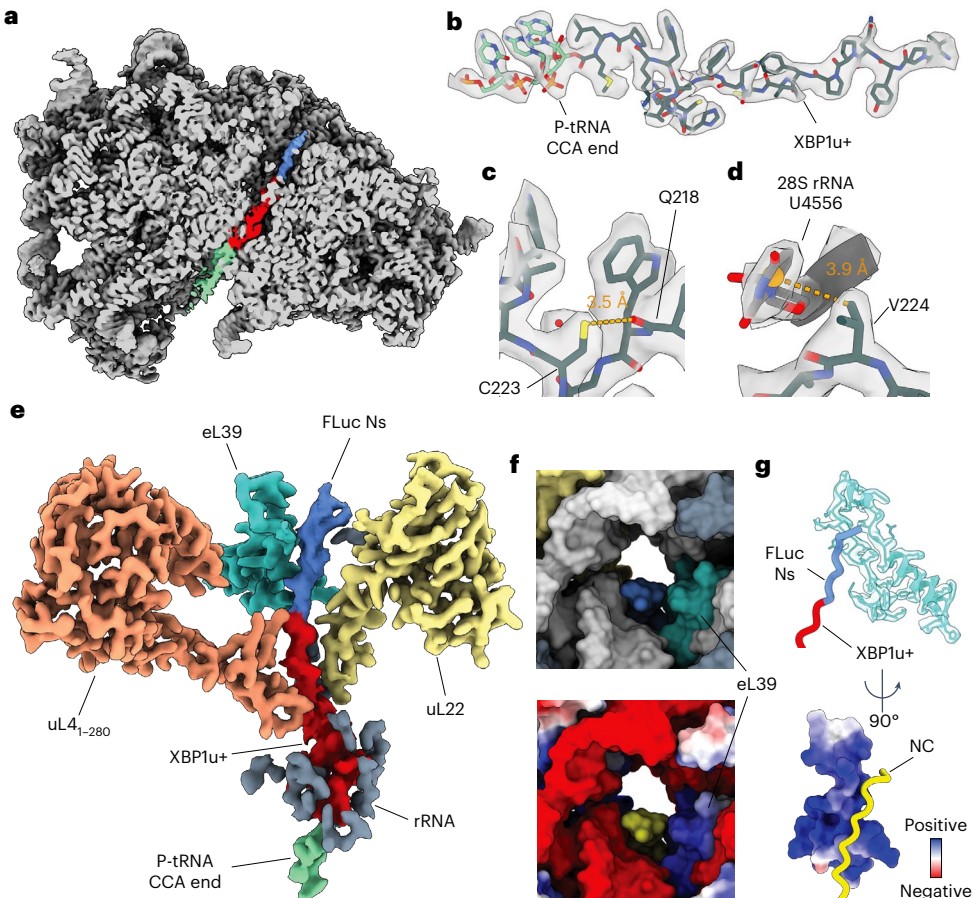

**Fig. 2 | Structure of Ns-RNC. a**, Cross-section of the consensus map of Ns-RNC, with the 60S ribosome in gray, density corresponding to the P-site tRNA in green, the XBP1u+ arrest peptide in red and FLuc in blue. **b**, Map–model overlay of the XBP1u+ arrest peptide (black) and the CCA end of the P-tRNA (green). **c**, Intrachain contact in XBP1u+ not observed in XBP1u. The thiol sulfur of C223 is within 3.5 Å of the backbone oxygen of Q218 in XBP1u+. **d**, Ribosome contact involving XBP1u+ not observed for XBP1u. The center of the aromatic ring of U4556 (28S rRNA) is within 3.9 Å of the nearest side-chain methyl carbon from XBP1u+ V224. **e**, Cryo-EM density of the NC within the exit tunnel. XBP1u+ is colored red and FLuc is in blue. rRNA (gray) is only shown if within 4 Å of the NC. **f**, Top, surface representation of the exit tunnel vestibule showing the NC (blue), uL22 (yellow), eL39 (turquoise), rRNA (white) and other ribosomal proteins (gray). Bottom, same view colored by electrostatic potential (blue, positive; red, negative) with the NC in yellow. **g**, Top, close-up view of the NC (red, XBP1u+; blue, FLuc) in proximity of eL39. Bottom, view rotated by 90° and colored according to electrostatic potential (blue, positive; red, negative) with the NC in yellow (Extended Data Figs. 3 and 4 and Supplementary Table 1).

Folding of the N-terminal domain is templated by its partner interface, promoting native interdomain contacts. Folded domains then detach as translation progresses, potentially helping to avoid entrenching misfolded states. This contrasts with the bacterial ribosome, which permits stable docking of N-terminal domains and interferes with folding of C-terminal domains.

## Results

### Design and preparation of FLuc RNCs

FLuc consists of 550 residues and was initially described to contain two domains: a large N-terminal domain containing the active site, connected by a flexible linker to a smaller C-terminal domain[38] (Fig. 1a). The N-domain has a complex topology and can be subdivided into three discontinuous subdomains with distinct folds: two inverted Rossman-like folds and a β-barrel roll (β-roll). Despite containing only part of each Rossmann fold, the N-terminal part of the N-domain (Ns, residues 15–190) was shown to form a protease-stable substructure during translation but not refolding from denaturant, suggesting that it may behave as an independent folding unit[21]. Consistent with this, InterPro divides FLuc into three consensus domains with residues 54–186 constituting an independent domain (CATH superfamily 3.40.50.980) (Fig. 1b).

To characterize folding intermediates of FLuc on the human ribosome, we first developed an approach to prepare suitable quantities of homogeneously stalled RNCs, analogous to previous work in bacteria[19,39–41]. To stall translation, we designed an arrest-enhanced variant of XBP1u, guided by a previous mutagenesis study[42]. The resulting XBP1u+ was around fourfold more efficient at stalling FLuc translation in human cells than the previously characterized XBP1u[43] (Extended Data Fig. 1a). Stalling positions were chosen on the basis of the domain architecture of FLuc (Fig. 1c). The RNCs exposed half of the Ns subdomain (½Ns-RNC, residues 1–123), the entire Ns (Ns-RNC, 1–208), half of the N-domain (½N-RNC, 1–388) or the entire N-domain (N-RNC, 1–458). To mimic the stage of folding immediately before translation termination, we additionally created T-RNC (1–528), in which the C-terminal 22 amino acids of FLuc was replaced with the stalling sequence. As a control, we designed an RNC where the full-length FLuc sequence was extended from the ribosome by a 50-amino-acid GS-rich linker (FL-RNC).

We expressed the stalling constructs in suspension-adapted HEK293 cells and purified RNCs using an N-terminal 3×Flag tag on the NC. We rigorously purified the RNCs to remove loosely bound interactors including chaperones, allowing us to isolate the effect of the ribosome on NC folding. High-salt purification also removed the 40S subunit,

facilitating downstream HDX-MS analyses, which are limited by sample complexity. NCs resolved by SDS–PAGE were sensitive to RNase, indicating that they were covalently linked to peptidyl-tRNA (Fig. 1d,e). FL-RNC was enzymatically active, indicative of native folding (Extended Data Fig. 1b). MS showed that the purified RNCs were substantially depleted of 40S ribosomal proteins, as expected, but retained the NC at near-stoichiometric levels (Extended Data Fig. 1c,d).

Aside from 60S ribosomal proteins and FLuc, the only other abundant proteins in the RNCs were EIF6, which is established to bind isolated 60S subunits[44], and Hsp70 (HspA1A) (Extended Data Fig. 1e and Supplementary Table 1). These each copurified at ~20% occupancy, irrespective of NC length, arguing against NC-specific binding. To locate Hsp70 on RNCs, we used crosslinking MS (XL-MS). We identified crosslinks between Hsp70 and the N-terminal part of ribosomal protein eL24, which is distant from the exit tunnel (Extended Data Fig. 2a). eL24 bridges the small and large ribosomal subunits and its N terminus is expected to be flexible in the absence of 40S. Crosslinks to eL24 stemmed from the substrate-binding domain and 'lid' of Hsp70, consistent with a substrate-like interaction (Extended Data Fig. 2b). Hsp70 did not crosslink to the NC in any sample, suggesting that it is not directly bound to nascent FLuc in the high-salt-purified RNCs.

## Structure of Ns-RNC

To further characterize the stalled RNCs, we solved the structure of Ns-RNC to a global resolution of 2.2 Å by cryo-EM (Fig. 2a and Extended Data Figs. 3a–d and 4a). Ns-RNC was chosen because the Ns subdomain was previously shown to fold cotranslationally[21]. As observed for rabbit ribosomes stalled on XBP1u^S255A (ref. 42) stalling by XBP1u+ does not induce large conformational changes in the 60S ribosome. The P-site tRNA is poorly resolved relative to the ribosome, because of high flexibility in the absence of the small subunit (Fig. 2a and Extended Data Fig. 4e). However, we observe clear density for the P-tRNA CCA tail in our consensus map (Fig. 2b and Extended Data Fig. 4f), confirming attachment of the NC.

The stalling sequence is well resolved in our map, allowing us to model all side chains and two ordered water molecules near R221 and W226 (Fig. 2b and Extended Data Fig. 4b–d). XBP1u+ adopts a similar conformation to that previously observed for XBP1u, including a turn involving residues W219 to W226 (ref. 42) (Extended Data Fig. 4h). Compared to wild-type XBP1u, XBP1u+ contains four substitutions (L216I, Q223C, P224V and S225A; Ns-RNC numbering) that increase its resistance to NC folding-induced release[42], two of which we can rationalize using our structure. The Q223C substitution places a thiol sulfur within 3.5 Å of the backbone carbonyl oxygen of Q218, allowing the formation of a hydrogen bond that likely stabilizes the W219–W226 turn[45,46] (Fig. 2c). The P224V substitution positions one of the side-chain methyl carbons within 3.9 Å of the aromatic ring of 28S rRNA residue U4556, compatible with a CH–π interaction[47,48] (Fig. 2d). Thus, stabilization of the arrest peptide conformation and increased interactions with the ribosome may both contribute to the increased stalling efficiency of XBP1u+.

Our map shows continuous density for the NC in the ribosomal exit tunnel. Clear side-chain density allowed us to confidently model the entire XBP1u+ and eight FLuc residues N-terminal to the stalling sequence (Fig. 2b,e and Extended Data Fig. 4g). Despite the volume available to the NC past the constriction point[30,31], the path of nascent FLuc is biased toward one side of the tunnel, where it occupies a groove formed by rRNA and the eukaryote-specific ribosomal protein eL39 (Fig. 2f,g). The positively charged inner surface of eL39 confers a mixed-charge character to the groove, suggesting that the trajectory of the NC may be guided by charge effects (Fig. 2f,g). FLuc adopts a partially compacted structure in the exit tunnel, forming a left-handed helix with three residues per turn and a ~3-Å rise per residue, resembling a κ-helix[49] (Fig. 2e and Extended Data Fig. 4g). The same residues make up an unstructured coil in native FLuc, suggesting that confinement in the eL39–rRNA groove may contribute to stabilizing non-native secondary structure in the NC. A similar phenomenon was previously noted for the all-β protein CspA, which showed density in the exit tunnel consistent with an α-helix[50].

The absence of clear density for FLuc beyond the exit tunnel indicated that the emerging NC is conformationally dynamic. Consistent with this, we identified frequent crosslinks between NCs and solvent-exposed residues on the ribosome surface (Extended Data Fig. 2c,d). uL29, directly at the tunnel vestibule, crosslinked to all NCs. Longer NCs also crosslinked to ribosomal proteins further from the tunnel exit, including eL22, uL22, uL24 and eL38.

## Sequence of folding events during FLuc synthesis

We next used HDX-MS to probe the conformation of FLuc on the ribosome. We measured peptide-resolved deuterium uptake of NCs and used isolated (off-ribosome) FL-FLuc as a reference for the native state (Fig. 3a). The Flag tag did not affect deuterium uptake of FL-RNC and was retained (Extended Data Fig. 5a). Sequence coverage of NCs was >83% and most peptides were detected across different RNCs, allowing quantitative comparison between states (Extended Data Fig. 5b and Supplementary Data 3). HDX-MS data for each RNC are summarized in Fig. 3a–g and representative peptides are shown in Fig. 3h for quantitative comparison. Data were measured in triplicate, with an average s.d. of 0.11 Da across the entire dataset. Note that a difference in uptake of 0.5 Da would correspond to protection from exchange of a single amide hydrogen, considering an average back exchange of 50% (Supplementary Data 3).

We found that peptides in ½Ns were deprotected by 2–6 Da relative to the same region in FL-FLuc, indicating that ½Ns was unfolded (Fig. 3b). Ns was also globally unfolded and peptides from this subdomain reached near-native levels of deuterium exchange (within 0.5 Da) only when a larger part of the N-domain was synthesized (½N-RNC) (Fig. 3c,d). The interface between Ns and N remained deprotected by 1–3 Da until the N-domain was complete (N-RNC), stabilizing the β-roll that connects the N-domain to the extreme N terminus of FLuc (Fig. 3e). At the final stage of translation, before termination, the extreme C-terminal residues of FLuc are within the ribosomal exit tunnel and, therefore, unavailable to fold with the rest of the NC.

**Fig. 3 | Sequence of folding events during FLuc synthesis. a**, HDX-MS analysis of FLuc NCs. Difference in deuterium uptake, after 3 min of deuteration, between FLuc NCs and native FL-FLuc. Larger values indicate more deuteration of NCs compared to FL-FLuc. Peptides forming the Ns:N interface and β-roll are indicated. Data represent the mean ± s.d. (n = 3–5 independent labeling reactions, depending on the construct; Supplementary Data 3). Every second peptide is labeled. **b–g**, Difference in relative fractional uptake (ΔRFU), after 3 min of deuteration, between each NC and FL-FLuc. Data are mapped onto the AlphaFold 2 model of FL-FLuc. Darker red indicates increased deuteration of NCs compared to FL-FLuc. Regions without peptide coverage are colored yellow. **h**, Deuterium uptake, after 3 min of deuteration, for representative peptides from NC and isolated FL-FLuc. Data represent the mean ± s.d. (n = 3–5

independent labeling reactions, depending on the construct; Supplementary Data 3). Dashed lines are guides for the eye only. P values were calculated for the difference in uptake between N-RNC and T-RNC peptides, between FL-RNC and FL-FLuc peptides and at the N:Ns interface and β-roll (one-way unpaired t-test with Welch's comparison). **i**, Schematic of FLuc intermediates on the ribosome. Folding of Ns is delayed until the native interface with N is synthesized (½N-RNC). Ns and N then stably associate when N is completely synthesized and close to the ribosome surface (N-RNC). During subsequent C-domain synthesis (T-RNC), Ns:N and β-roll are destabilized. Native interdomain contacts are recovered when the C-domain emerges fully from the ribosome (FL-RNC) (Extended Data Fig. 5 and Supplementary Data 3).

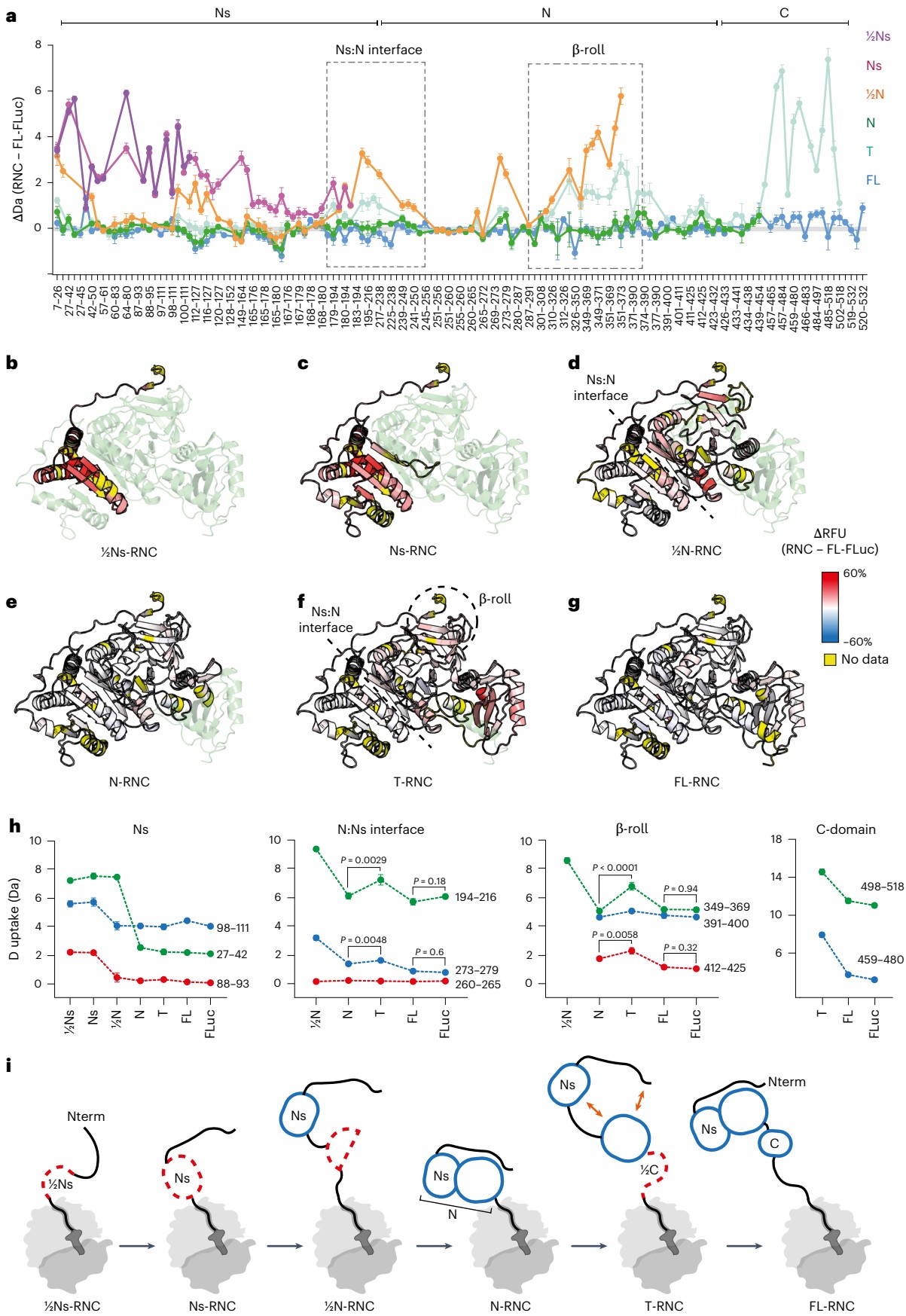

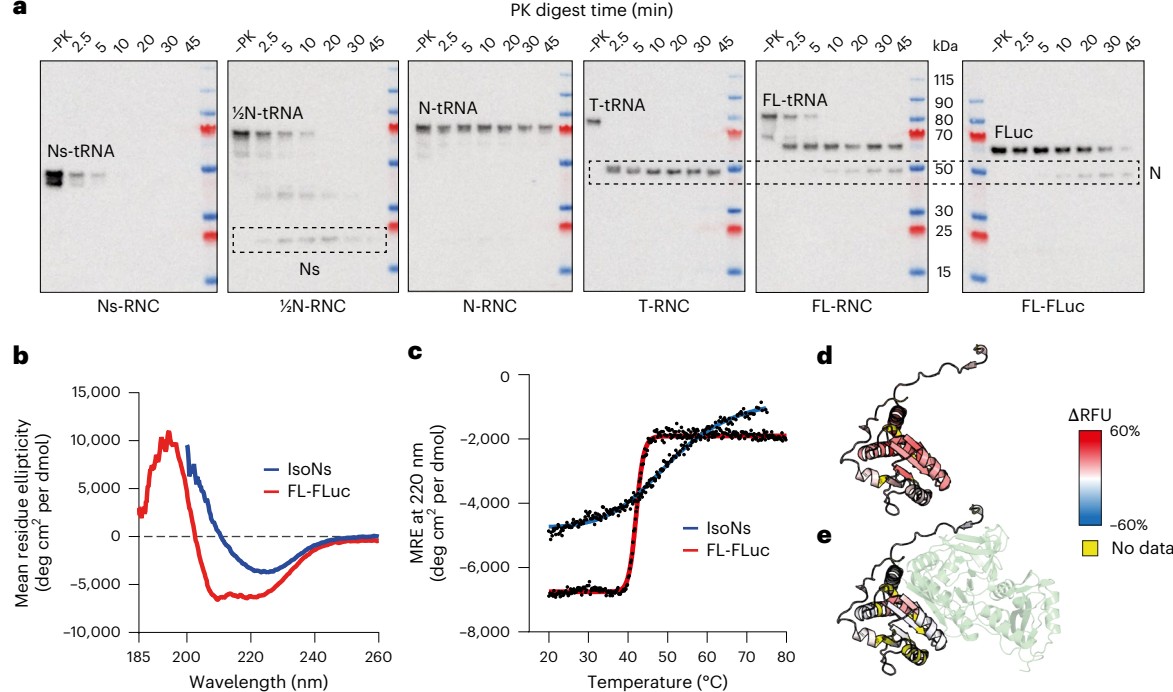

**Fig. 4 | Folding and docking of the Ns subdomain is delayed relative to synthesis. a**, Limited proteolysis of FLuc RNCs. RNCs were treated with PK for different times and probed by immunoblotting using a monoclonal antibody against the FLuc N terminus. Experiments were repeated using three independent protein purifications, with similar results. **b**, Secondary structure of isolated Ns (isoNs). Circular dichroism spectroscopy of isoNs and FL-FLuc. Spectral deconvolution showed 4.4% α-helix and 38.5% β-sheet for isoNs, in contrast to 24.7% α-helix and 23.5% β-sheet for Ns (residues 1–190) in the context

of FL-FLuc, calculated on the basis of the crystal structure (PDB 1LCI). **c**, Thermal denaturation of FL-FLuc ($T_m$ = 42 °C) and isoNs ($T_m$ = 50.5 °C). Data were fitted to sigmoidal functions in GraphPad Prism. **d**, Conformational dynamics of isoNs. The ΔRFU, after 3 min of deuteration, between isoNs and FL-FLuc. Darker red indicates increased deuteration of isoNs compared to FL-FLuc. **e**, As in **d**, for Ns-RNC compared to isoNs. Darker red indicates increased deuteration of Ns-RNC compared to isoNs (Extended Data Fig. 6 and Supplementary Data 3).

In our T-RNC construct that mimics this step, C-domain peptides were highly deprotected (up to 7 Da) relative to FL-FLuc, as expected. Surprisingly, the N-domain β-roll and the Ns:N interface of T-RNC were also deprotected by 1–3 Da compared to RNCs, mimicking earlier stages of translation (Fig. 3f,h). Extending the C terminus of FLuc from the ribosome on a 50-amino-acid linker (FL-RNC) resulted in native-like folding of C-domain peptides and recovered the stability of N-domain peptides to within 0.5 Da of FL-FLuc (Fig. 3g). Together, these data resolve length-dependent local folding of partially synthesized FLuc on the human ribosome (Fig. 3i). We find that Ns folding is delayed relative to synthesis and the NC is destabilized before translation termination.

### Folding and docking of the Ns subdomain is delayed relative to synthesis

Our HDX-MS analysis indicated that stable folding of Ns was triggered by partial synthesis of the larger N-domain. As an orthogonal approach to identify stable subdomains, we performed limited proteolysis of RNCs using proteinase K (PK). Ns-RNC was rapidly digested by PK, consistent with our conclusion that it does not fold independently on the ribosome (Fig. 4a). Digestion of ½N-RNC resulted in weak accumulation of the previously described ~22-kDa intermediate corresponding to Ns[21], suggesting that Ns is stabilized by interactions with the N-domain and consistent with HDX-MS showing that Ns is near-natively folded after synthesis of residues 1–388 (Fig. 3a,d,h). The Ns intermediate did not accumulate upon digestion of longer RNCs or FL-FLuc. Instead, we observed a stable intermediate consistent in size with the full N-domain (Fig. 4a). N-RNC was resistant to PK, as expected on the basis of HDX-MS (Fig. 3a,e).

To test whether Ns behaves as an independent folding unit, we purified isolated Ns (isoNs, residues 1–190) off the ribosome

(Extended Data Fig. 6a,b). Unexpectedly, isoNs was dimeric at nanomolar–micromolar concentrations, as judged by size-exclusion chromatography coupled to multiangle laser light scattering (SEC–MALLS) and mass photometry (Extended Data Fig. 6c,d). Circular dichroism spectroscopy showed that isoNs had altered secondary structure, with reduced α-helix and increased β-sheet content compared to Ns in the context of FL-FLuc (Fig. 4b). Moreover, isoNs showed a shallow transition during thermal denaturation, consistent with low global unfolding cooperativity (Fig. 4c). In line with this, HDX-MS showed that isoNs was globally unfolded relative to native Ns in FL-FLuc (Fig. 4d). When compared to Ns-RNC, isoNs was protected at several peptides forming the interface with the N-domain in FL-FLuc (Fig. 4e). We speculate that isoNs may, therefore, dimerize through the same interface to form a non-native intermolecular β-sheet.

In summary, we find that Ns folds cotranslationally, in agreement with previous reports[21,35]. Our data are also consistent with the time course of Ns formation in synchronized translation reactions[35]. A discrete protease-resistant species corresponding to Ns was shown to form early and then disappear upon incorporation into FL-FLuc. Using HDX-MS analysis of purified RNCs and isoNs, we further show that Ns is not an independent folding unit but is held on the ribosome in a folding-competent state until a larger part of the discontinuous Rossmann fold is available (Fig. 1a). The Ns:N interface stabilizes later, when the entire N-domain is synthesized.

### Domain interfaces are destabilized before translation termination

Our HDX-MS analysis indicated that the N-domain was natively folded in N-RNC but partially destabilized in T-RNC. As an orthogonal measure of NC stability, we used fluorescein-5-maleimide (F5M)

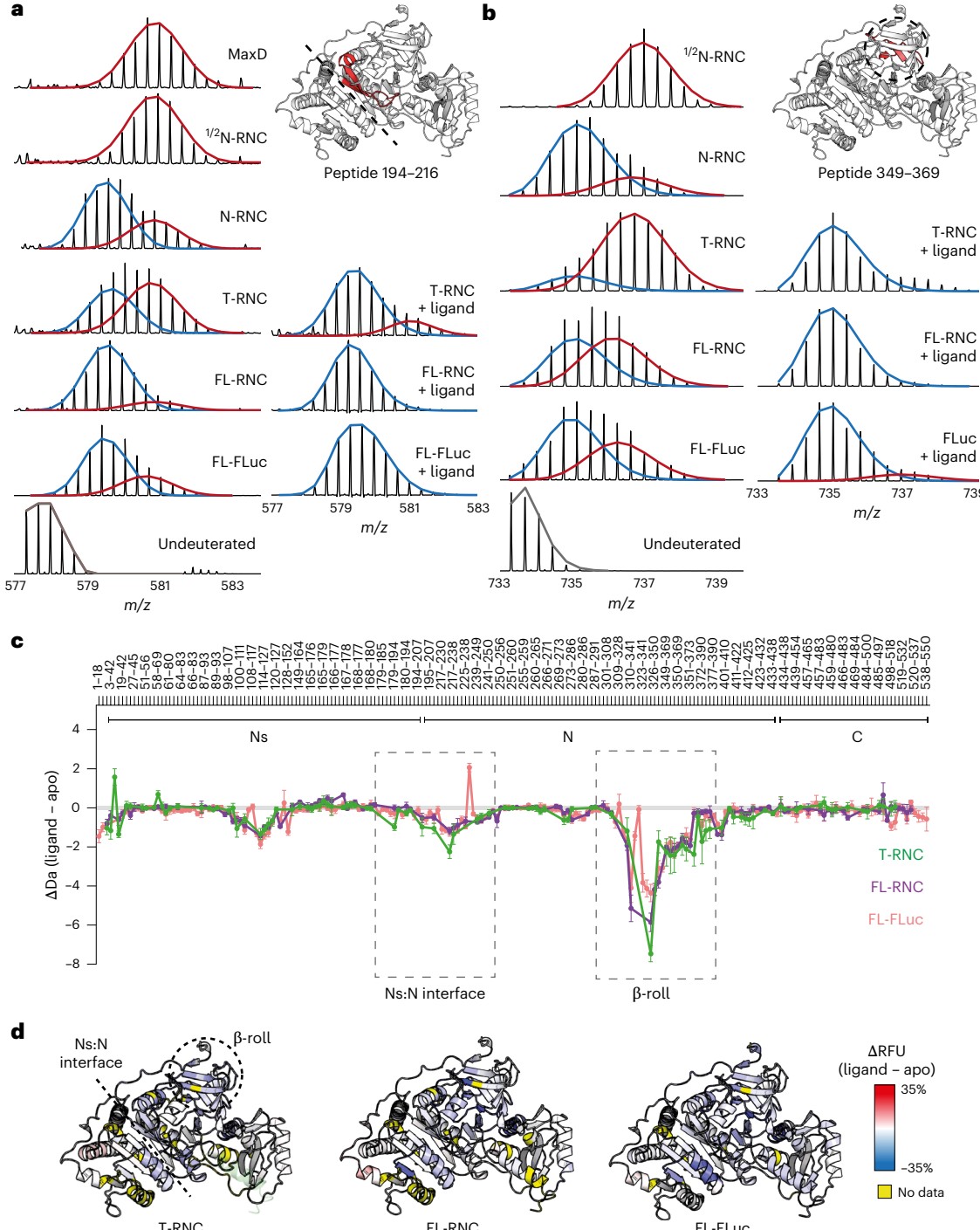

**Fig. 5 | FLuc domain interfaces are destabilized before translation termination. a,b,** Concerted HDX at FLuc domain interfaces. Mass spectral envelopes for peptide 194–216 at the Ns:N interface (**a**) and 349–369 in the β-roll (**b**) of FLuc. Isotopic distributions were fitted to single or bimodal gaussian distributions using HX-Express3.0 (ref. 65). The high-exchanging population is colored red and the low-exchanging population is colored blue. Except for the maximally deuterated sample (MaxD), samples were deuterated for 3 min. In the +ligand condition, samples were incubated with 50 μM PBT and 1 mM ATP

before deuteration. **c,** Effect of PBT and ATP (ligand) on FLuc conformational dynamics. Difference in deuterium uptake, after 3 min of deuteration, between FLuc samples with and without ligand. Smaller values indicate less deuteration of the ligand-bound samples. Data represent the mean ± s.d. (*n* = 3–5 independent labeling reactions, depending on the construct; Supplementary Data 3). Every third peptide is labeled. **d,** The ΔRFU, after 3 min of deuteration, between FLuc samples with and without ligand. Regions colored blue are protected upon ligand binding (Extended Data Fig. 7 and Supplementary Data 3).

to probe the accessibility of native cysteine residues in FLuc. All cysteines are in the N-terminal domain and are buried in the native state (Extended Data Fig. 7a). N-RNC and FL-RNC were labeled by F5M with similar kinetics, whereas T-RNC was labeled significantly faster, consistent with increased cysteine exposure (Extended Data Fig. 7b–d).

N-domain destabilization did not depend on the specific C-terminal sequence, as replacing C-domain residues with a glycine and serine stretch of equivalent length resulted in identical cysteine labeling kinetics (Extended Data Fig. 7c,d). Destabilization of T-RNC was not reflected by increased susceptibility of the N-domain to proteolysis (Fig. 4a).

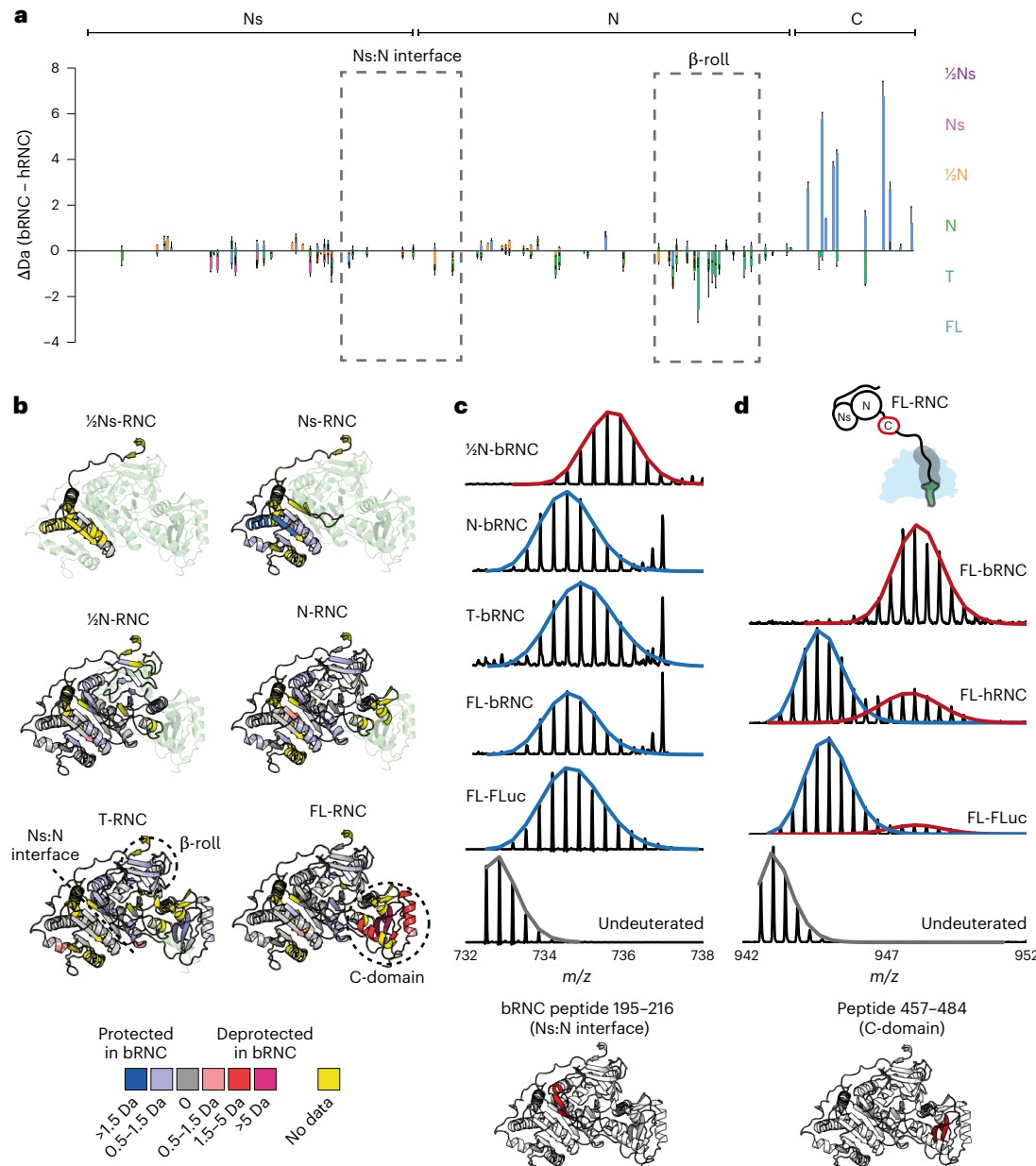

**Fig. 6 | Human and bacterial ribosomes differentially affect the conformational ensemble of FLuc NCs. a**, Comparison of human and bRNCs. Difference in deuterium uptake, after 3 min of deuteration, between FLuc NCs on the human and bacterial ribosome. Positive values indicate more deuteration on the bacterial ribosome. Negative values indicate more deuteration on the human ribosome. Data represent the mean ± s.d. ($n = 3–5$ independent labeling reactions, depending on the construct; Supplementary Data 3 and 5). **b**, As in **a**, mapped onto the structure of FL-FLuc. Regions that are protected in bRNCs compared to hRNCs are colored blue, whereas deprotected regions are colored red. Regions without peptide coverage are colored yellow. **c**, Mass spectral envelopes for peptide 195–216 at the Ns:N interface of bRNCs and FL-FLuc. The high-exchanging population is colored red and the low-exchanging population is colored blue. **d**, Mass spectral envelopes for peptide 457–484 in the C-domain of FL-RNCs and FL-FLuc, colored as in **c** (Extended Data Fig. 9 and Supplementary Data 3 and 5).

This could be because of limited protease accessibility to cleavage sites at FLuc domain interfaces or result from the fact that the destabilizing C-terminal sequence is cleaved immediately after adding protease.

Destabilization of the N-domain in T-RNC was primarily localized to the β-roll and Ns:N interface (Fig. 3a,f,h). Mass spectra for peptides in these regions displayed EX1 kinetics and were multimodal, indicating that a subset of residues within the peptide is slow to refold relative to the timescale of HDX (Fig. 5a,b). The peptides map to similar structural motifs, consisting of two antiparallel β-strands and a connecting loop. Because the loops pack against structured elements at the Ns:N interface and β-roll:N interface in native FLuc, they are likely to be substantially deprotected if these interfaces are not formed.

Multimodal behavior was dependent on NC length. Only the unfolded conformation was detected in ½N-RNC, whereas the folded population predominated in N-RNC and FL-RNC. T-RNC showed intermediate behavior but was predominantly unfolded. If the unfolded state sampled by T-RNC is in equilibrium with the native state, stabilizing the native state would be expected to alter the population distribution. To test this, we added the active site ligands phenobenzothiazine (PBT) and ATP to T-RNC. Note that, although T-RNC is not enzymatically active, it has a complete enzyme active site housed in the N-domain. Adding ligands protected the β-roll and Ns:N interface regions of T-RNC, FL-RNC and FL-FLuc from deuterium exchange (Fig. 5c,d), and shifted multimodal spectra toward the folded

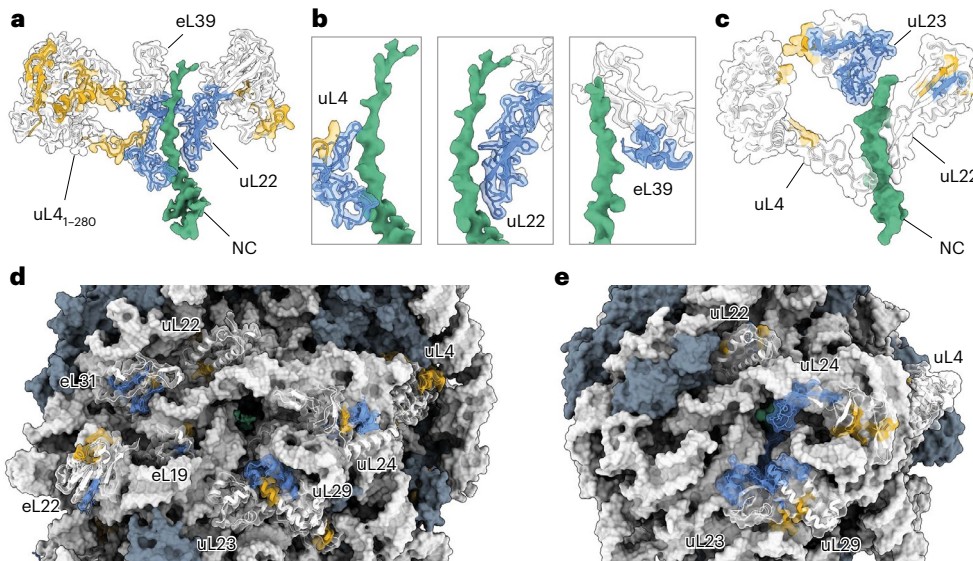

**Fig. 7 | NC interaction sites on human and bacterial ribosomes. a**, HDX-MS analysis of human ribosomal proteins lining the exit tunnel. Difference in deuterium uptake, after 3 min of deuteration, between human RNCs and empty 80S ribosomes. Sites protected by NCs (>0.5 Da) are colored blue on the structure of Ns-RNC. Regions that do not change in uptake are colored white and regions without peptide coverage are colored yellow. Only residues 1–280 of uL4 are shown for clarity. **b**, Close-up views of uL4, uL22 and eL39, colored as in **a**.

**c**, HDX-MS analysis of bacterial ribosomal proteins lining the exit tunnel. Difference in deuterium uptake, after 3 min of deuteration, between bRNCs and empty 70S ribosomes. Protection is mapped as in **a** on the structure of a SecM-stalled ribosome (PDB 8QOA)[66]. **d**, As in **a**, for proteins on the human ribosome surface. **e**, As in **c**, for proteins on the bacterial ribosome surface (Extended Data Fig. 10 and Supplementary Data 6 and 7).

population (Fig. 5a,b). The high-exchange states populated by T-RNC are, therefore, in equilibrium with the native state and not stably mis-folded or kinetically trapped.

Together, these data show that the distribution of states populated by nascent FLuc is altered by synthesis of a C-terminal unstructured sequence, which shifts the equilibrium toward the unfolded state and destabilizes preformed domain interfaces.

### Human and bacterial ribosomes differentially affect the NC conformational ensemble

To understand whether cotranslational folding intermediates differ between human and bacterial ribosomes, we generated equivalent bacterial RNCs (bRNCs) for FLuc in *Escherichia coli* (Extended Data Fig. 8a–c). bRNCs were purified from Δ*tig* cells to avoid copurifying Trigger factor and RNC composition was verified by MS (Supplementary Data 4). Stalling positions were offset by 12 amino acids to account for the difference in length of the stalling sequence between XBP1u+ (22 amino acids) and the bacterial arrest peptide SecM[Str] (10 amino acids). HDX-MS showed that the overall sequence of folding events is similar on bacterial and human ribosomes, including delayed folding of Ns (Extended Data Fig. 9a,b). However, N-domain peptides were protected on the bacterial compared to human ribosome at all NC lengths, indicating that the environment of backbone hydrogens is altered (Fig. 6a,b). Furthermore, the bacterial ribosome modu-lated the domain docking equilibrium. The Ns:N interface was not destabilized in T-bRNC, indicating that the unfolded conformation is sampled much less frequently than on the human ribosome (Figs. 6c and 5a). Similar behavior was observed for the β-roll (Extended Data Fig. 9c). The bacterial ribosome, therefore, shifts the conformational equilibrium of the N-domain of nascent FLuc to favor the domain-docked state.

The C-domain in FL-RNC showed the opposite effect and was substantially deprotected on the bacterial compared to human ribosome (Fig. 6a,b). Similar to the Ns:N interface and β-roll, pep-tides in the C-domain were bimodal, indicative of a slow confor-mational exchange between folded and unfolded states (Fig. 6d).

Whereas the unfolded conformation was rarely sampled on the human ribosome and in isolated FL-FLuc, the same peptide was predomi-nantly unfolded on the bacterial ribosome. The C-domain was none-theless folding competent, as FL-bRNC was enzymatically active (Extended Data Fig. 8d), albeit slightly less so than the equivalent FL-hRNC (Extended Data Fig. 1b). Previous comparisons of FLuc bio-genesis efficiency in bacteria and eukaryotes arrived at conflicting conclusions[29,51]. However, the observation that FLuc folding is partially post-translational in *E. coli*[29] is consistent with our finding that the bacterial ribosome interferes with folding of the C-domain of FLuc near the ribosome surface.

Interactions with the ribosome surface may bias the NC con-formational ensemble[15–17,52,53]. To identify interaction sites, we used HDX-MS to analyze the conformational dynamics of human and bacte-rial ribosomal proteins, in the presence and absence of different NCs (Fig. 7a–e and Extended Data Fig. 10a,b). The tunnel-facing surface of eukaryote-specific eL39 on the human ribosome was protected by up to 2 Da from deuterium exchange, consistent with our cryo-EM structure of Ns-RNC. Protection was not specific to particular NCs, suggesting that the interaction with eL39 does not depend on the sequence of the NC. In addition, NCs protected sites on human ribosomal proteins uL22, eL22, uL29, eL19 and eL31. eL19 and eL31 are unique to eukary-otes and form part of the binding site for the ribosome-associated protein βNAC[54]. In bacteria, the most prominent interaction sites were extended loops on uL23 and uL24. These loops protrude into the ves-tibule and are absent from the equivalent human proteins. Together, these data argue that species-specific ribosomal elements bind NCs and potentially modulate their conformation.

### Discussion

Here, we characterized sequential cotranslational folding intermedi-ates on the human ribosome. We traced the path of the NC through the exit tunnel, identified NC interactions with the ribosome surface and showed how local folding dynamics change during synthesis. This analysis of eukaryotic RNCs revealed differences in NC folding com-pared to bacterial ribosomes.

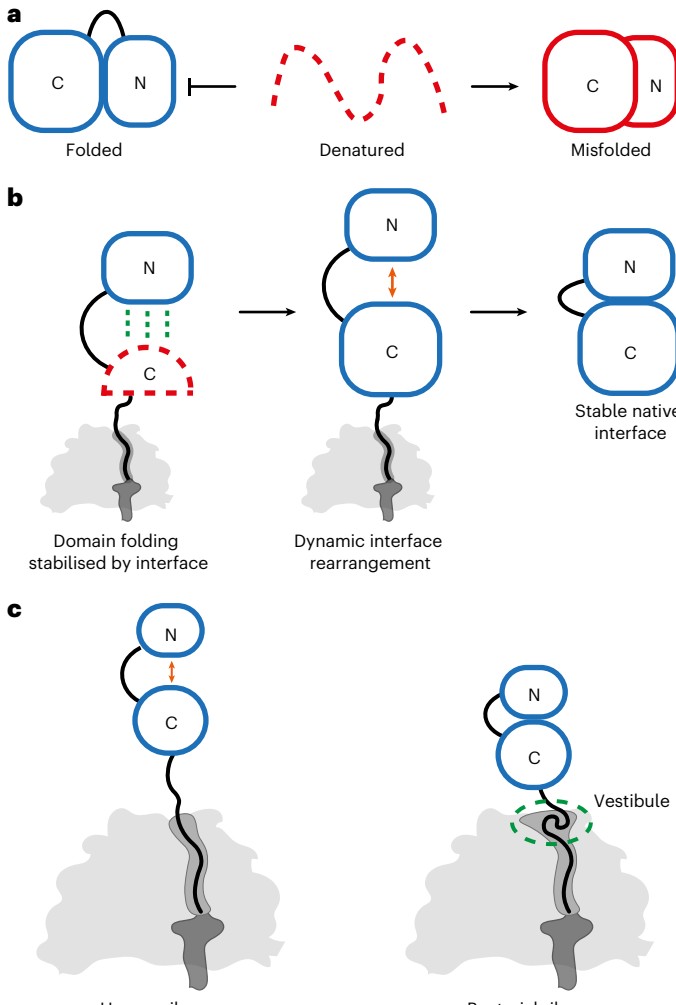

**Fig. 8 | Multidomain protein biogenesis modulated by the ribosome.**
**a**, Interdomain misfolding in vitro. Denatured multidomain proteins tend to spontaneously misfold in vitro, forming compact non-native states[3]. **b**, Factors promoting multidomain protein biogenesis. Domain folding is stabilized when the partner interface emerges from the ribosome, thereby avoiding misfolding with elements that are not yet synthesized. Later in translation, dynamic rearrangement of domain interfaces on the ribosome may avoid entrenching non-native contacts. **c**, The wide vestibule of bacterial ribosomes accommodates collapsed or partially folded NCs, preventing unstructured C-terminal sequences from destabilizing domain interfaces.

interactions[3–5,9,55] (Fig. 8a). The same folding vulnerability was illustrated in our study by the tendency of isoNs to form a non-native dimer when the native interface is unavailable.

How do multidomain proteins fold efficiently in vivo? Previous work showed that interdomain misfolding can be circumvented by separating domain folding on the ribosome[22] or rescued by the Hsp70 chaperone system[3,34,56]. Here, we suggest two additional factors that optimize de novo folding of multidomain proteins (Fig. 8b). First, (sub)domain folding is triggered when the native domain interface first becomes available. This might allow the correct interface to form before additional sequence, encoding competing interactions, is synthesized. Such a mechanism may be less relevant for simpler multidomain proteins with continuous domain architectures. Second, domain interfaces are partially destabilized by the emergence of unstructured (not yet folded) nascent polypeptide from the exit tunnel. A similar phenomenon was observed for EF-G[23] and DHFR[19] and may be explained by the entropic cost of tethering a disordered polypeptide to a folded domain[57,58]. This allows for increased sampling of conformational space during synthesis, which could help to avoid kinetic traps or recover from transient non-native interactions between domains. Such a mechanism is in principle sequence independent (Extended Data Fig. 7c,d) and, thus, might be a general feature of multidomain protein biogenesis. Chaperones are also involved in FLuc biogenesis[35] and may have a role in stabilizing the nascent Ns before the Ns:N interface is satisfied by native intramolecular contacts. Future research will need to establish how exactly folding on the human ribosome is modulated by molecular chaperones and explore these phenomena for other model proteins.

### Different folding environments on human and bacterial ribosomes

Multidomain proteins often fold more efficiently in eukaryotes compared to bacteria[22,28,29]; however, whether the ribosome contributes directly has not been clear. Here, we show how different ribosomes can shape the conformational ensemble of a model NC. In eukaryotes, nascent domains are preferentially undocked and the C-domain folds close to the ribosome. In bacteria, N-domain interfaces are stable throughout translation and the complete C-domain is held in an unfolded conformation. Together with differences in elongation kinetics, delayed folding of C-terminal domains may contribute to the general tendency of multidomain proteins to complete folding after translation when expressed in bacteria[22,29].

What differentiates human and bacterial ribosomes? Human ribosome exit tunnels are both narrower and shorter than their bacterial counterparts[30], likely disfavoring cotranslational folding in the exit tunnel (Figs. 7 and 8c). Moreover, although tertiary structure can form in the bacterial exit tunnel as it widens near the exit port[12,50,59], the equivalent vestibule in humans is constricted by the eukaryote-specific tunnel protein eL39. We showed using HDX-MS that diverse sequences directly contact eL39 and using cryo-EM that the NC follows a narrow groove between eL39 and rRNA without sampling substantially compact structures. We speculate that exit tunnel architecture shapes overall NC folding by dictating the conformation of the C-terminal part of the NC. Unstructured C-terminal sequences outside the exit tunnel of the ribosome can destabilize folded domains[19,23,57,58], thereby amplifying NC interdomain dynamics on the human ribosome (Fig. 8c). On the bacterial ribosome, collapse of C termini in the wide vestibule may insulate already folded domains from entropic destabilization (Fig. 8c). Indeed, the bacterial ribosome was recently shown to reduce NC entropy to favor partial folding[20]. The extent of entropic destabilization would be expected to depend on the stage of translation and stability of the nascent domain. In cases where a substantial unstructured C-terminal segment is exposed outside the exit tunnel of the bacterial ribosome, Trigger factor was shown to protect upstream folded domains from denaturation[23].

We found that FLuc does not fold in a simple domain-by-domain fashion on the ribosome. As a result of the discontinuous architecture of the Rossmann fold, the Ns subdomain folds only when a larger part of the N-domain is synthesized. Ns then latches onto the N-domain when the β-roll is complete. Subdomain folding is, therefore, stabilized by interactions across domain interfaces. Although Ns initially docks against N, the Ns:N and β-roll interfaces are destabilized during subsequent synthesis of the C-domain. As a result, domain interfaces are maintained in dynamic equilibrium between docked and undocked states, delaying stable domain association until release from the ribosome.

### Optimizing multidomain protein biogenesis

Combining domains in a single polypeptide expands protein function but introduces several folding challenges[6,8]. First, domain interfaces must be established with the correct geometry. Second, the coexistence of two unfolded domains entails the risk of interdomain misfolding. These challenges manifest during attempted refolding in vitro, which is complicated by non-native interdomain

**Article**

Specific interactions with the ribosome surface may also influence folding. On the bacterial ribosome, the NC engages prokaryote-specific loops on uL23 and uL24 that protrude into the vestibule and are established to affect the onset of folding[52,60]. Eukaryote-specific rRNA expansion segments[32] might also contribute to creating a unique folding environment on the human ribosome. Translation elongation is faster in bacteria than eukaryotes and it is possible that this also contributes to the difference in folding outcomes[61], which may persist even in stalled complexes at equilibrium. Indeed, previous work noted that the yield of active FLuc is sensitive to codon usage, especially in regions encoding the Ns:N interface[62].

## Online content

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

## Methods

### Plasmids and constructs

Stalling constructs were synthesized in mammalian expression vector pTwist CMV BG WPRE Neo (Twist Bioscience). The constructs contained an N-terminal 3×Flag tag and 3C protease site upstream of FLuc (subcloned from pGL2$^+$) and a downstream stalling sequence XBP1u+ (VPYQPPFICQWGRHCVAWKPLMN). Compared to XBP1u[43], XBP1u+ contains the following substitutions: L216I, Q223C, P224V, and S225A (Ns-RNC numbering). The open reading frame is under transcriptional control of the CMV promoter and β-globin and Woodchuck hepatitis virus post-transcriptional regulatory elements were included to enhance protein expression. pTwist vectors were amplified in DH5α strain *E. coli* in Luria–Bertani (LB) medium and maintained with 100 μg ml$^{-1}$ ampicillin. Truncations and modifications of the stalling construct were produced by deletion PCR using a Q5-SDM kit or Gibson assembly (New England Biolabs). DNA for transfection was prepared using a PureLink HiPure plasmid maxi-prep kit (Thermo Fisher Scientific) from 250-ml cultures.

### Human RNC purification

**Transient transfection and cell harvest.** Expi293F suspension cells (Thermo Fisher Scientific, A14527) were maintained at $0.4 \times 10^6$ cells per ml twice a week in FreeStyle 293 expression medium (Thermo Fisher Scientific) at 37 °C and 8% $CO_2$ with shaking at 125 rpm in vented Erlenmeyer flasks (Corning). Maximum volumes for culture were as follows: 30 ml of culture in 125-ml flask, 50 ml of culture in 250-ml flask, 100 ml of culture in 500-ml flask, 200 ml of culture in 1-l flask and 600 ml of culture in 2-l flask. Cell number and viability were estimated using a ViCell-XR cell counter (Beckman Coulter) and transfections were only carried out if viability was >95%. Cells were discarded at passage 20. Cells were split to $2 \times 10^6$ cells per ml 1 day before transfection and were diluted to $3.3 \times 10^6$ cells per ml on the day of transfection in nine tenths of the final transfection volume (final cell count: $3 \times 10^6$ cells per ml). Polyethylenimine (PEI)–DNA complexes were mixed at a 3:1 mass ratio, with 1 μg of plasmid DNA per 1 ml of $3 \times 10^6$ cells per ml to be transfected. PEI (Polysciences) (1 mg ml$^{-1}$ stock, pH 8.0) was diluted to 30 μg ml$^{-1}$ in half of 1/10 of the final transfection volume into Opti-MEM reduced-serum medium with GlutaMAX (Thermo Fisher Scientific) and plasmid DNA was diluted into the same volume of Opti-MEM, before incubating for no longer than 5 min at room temperature. The diluted PEI and DNA were mixed and incubated for 20 min at room temperature and then added dropwise with swirling to the transfection ready cells and incubated as above for expression of RNCs. Cells were harvested after 24 h of transient expression by centrifugation at 300$g$. Cells were subsequently washed in ice-cold PBS with cOmplete EDTA-free protease inhibitor tablet (Roche). Harvested cells were used immediately for RNC purification.

**Cell lysis and affinity purification.** All buffers were made using RNase-free and DNase-free molecular-biology-grade $H_2O$.

Cells were weighed and gently resuspended at 1/100 cell culture volume in ice-cold 80S lysis buffer (50 mM Tris-HCl pH 7.5, 0.5% NP-40 alternative, 5 mM MgCl$_2$, 25 mM KCl, 0.2 M arginine–HCl, 1 mM PMSF, 1 mM DTT, 2 μl per 10 ml of RNasIN and EDTA-free cOmplete protease inhibitor tablet) with a Pasteur pipette. RNase-free DNase I (Qiagen) was added at 10 μl per 1 g of wet cell weight and cells were incubated on ice for 30 min to promote lysis. Cell debris was removed by centrifugation at 14,000$g$ for 30 min at 4 °C and the clarified lysate was layered onto a 35% sucrose cushion (35% w/v RNA-free and DNAse-free sucrose, 50 mM HEPES–NaOH pH 7.5, 500 mM KCl, 5 mM MgCl$_2$, 0.2 M arginine–HCl, 1 mM PMSF and RNasIN) to enrich for ribosomes. Layered samples were centrifuged at 70,000 rpm in a Beckmann TLA-110 rotor for 2 h or 55,000 rpm for 4 h in Beckmann Ti-70 rotor at 4 °C. Supernatants were discarded and the clear, straw-colored pellets containing ribosomes and RNCs were resuspended with gentle agitation overnight in RNC

buffer (20 mM HEPES–NaOH pH 7.5, 500 mM KCl, 5 mM magnesium acetate, 10 mM NH$_4$Cl, 10% v/v glycerol, 1 mM PMSF and 0.2 M arginine–HCl) at 4 °C.

Resuspended ribosomal pellets were incubated with RNC buffer-equilibrated agarose anti-Flag M2 affinity gel (Thermo Fisher Scientific) for 3 h at 4 °C on a spinning wheel (50 μl of beads for 50 ml of culture). The unbound fraction was collected with a 7,000$g$ centrifugation step at 4 °C along with subsequent washes (3× 15 min) of RNC buffer. The 3×Flag-tagged RNCs were eluted from the resin with 1× 150 ng μl$^{-1}$ 3×Flag peptide (Peptide Chemistry STP, Francis Crick Institute) diluted in RNC buffer for 1 h at 4 °C.

### Generation and purification of 70S RNCs

bRNCs were prepared using construct design and high-salt purification methods detailed previously[19,67] with some minor changes. The Δ*tig* BL21 (from J. Christodoulou) cells were transformed with RNC constructs containing an N-terminal 3×Flag tag and C-terminal arrest-enhanced SecM stalling sequence (WWWPRIRGPPGS)[68] and grown in ZYM-5052 autoinduction medium at 37 °C for 4 h, before cooling to 18 °C for expression overnight. bRNCs were truncated at the following positions: ½Ns-bRNC, 1–135; Ns-bRNC, 1–220; ½N-bRNC, 1–400; N-bRNC, 1–470; T-RNC, 1–540; FL-bRNC, 1–550 + GS$_{50}$-SecM. FLuc length was altered to account for the relative lengths of SecM and XBP1u+ to maintain total construct length between orthogonal RNCs. All purification buffers additionally contained 0.2 M arginine–HCl and were bound to and eluted from anti-Flag M2 affinity gel, as for human RNC purifications.

### Purification of 80S ribosomes

Crude 80S ribosomes were purified as for human RNCs with some minor differences. Lysis was performed in 0.5 M KCl lysis buffer without arginine. Additionally, 2.5 mM puromycin was added during lysis to remove NCs from ribosome. Cell debris was removed by centrifugation at 14,000$g$ for 20 min; then, ribosomes were isolated from the supernatant by sucrose cushion ultracentrifugation as described above. Ribosome pellets were washed twice and resuspended in RNC buffer.

### RNC quality control

RNC concentration was estimated by measuring its absorbance at 260 nm ($A_{260}$), where $1 A_{260} = 20$ pmol ml$^{-1} = 0.02$ μM 80S ribosomes[69] or $1 A_{260} = 24$ pmol ml$^{-1} = 0.024$ μM 60S or 70S ribosomes[39]. The mass of 80S ribosomes was calculated using a molecular weight of 4.5 MDa. RNCs were aliquoted and flash-frozen for storage at −80 °C.

RNC integrity was checked by running 1–2.5 pmol of RNC in reducing Laemmli sample buffer (BioRad) on 4–12% Bis–Tris gels (Thermo Fisher Scientific) in MES–SDS running buffer. Samples were not boiled. Then, 0.5 μl of 1/10 diluted RNaseA and 0.5 μl of 0.5 M EDTA pH 8.0 were added to a 10-μl RNC sample to digest tRNA, for confirmation of ribosome stalling[39]. Gels were either stained with Coomassie to check characteristic 60S or 80S ribosomal banding or transferred (BioRad transblot turbo) to prepacked nitrocellulose membranes (BioRad) for western blot. NCs were blotted using anti-N-terminal FLuc (Abcam, ab185923, lot GR3317915-2). Membranes were blocked at room temperature for 1 h or overnight at 4 °C with 1% (w/v) milk in PBS with 0.1% (v/v) Tween-20 (PBST). Antibodies were used at 1:1,000–10,000 dilution, with a goat anti-Rabbit horseradish peroxidase secondary antibody (Abcam, ab205718) at 1:10,000 after first washing excess primary away with PBST. Membranes were exposed using SuperSignal West Pico PLUS chemiluminescent substrate (Thermo Fisher Scientific) on an Amersham 600 imager.

### Purification of recombinant FL-FLuc and off-ribosome truncations

His–SUMO–FLuc was cloned into a pET19 vector for expression in *E. coli*. Isolated truncations were produced by deletion PCR using

the Q5 site-directed mutagenesis kit (New England Biolabs) and contained an N-terminal His–SUMO tag. His–SUMO–FLuc constructs were transformed into BL21 cells and a single colony was used to seed a starter culture of LB maintained with 100 μg ml⁻¹ ampicillin. The starter culture was used to inoculate ZYM-5052 autoinduction medium[70] with 100 μg ml⁻¹ ampicillin, which was incubated for 4–6 h (until cloudy) at 37 °C before cooling to 18 °C for expression over 12–16 h. Cell pellets were harvested and washed in PBS before resuspension in lysis buffer (10 mM Tris pH 8.0, 10 mM imidazole, 300 mM NaCl, 5 mM β-mercaptoethanol, 1 mM PMSF and EDTA-free cOmplete protease inhibitor tablet) and storage at −80 °C.

For purification using an AKTA Pure system (Cytiva), resuspended pellets were thawed and incubated for 20 min at RT with 5 μl of Benzonase nuclease (Sigma). Cells were lysed with one pass through a cell disruptor (Constant Systems) at 25 kpsi at 4 °C and cell debris was removed with a 66,000g centrifugation for 30 min, 4 °C. Supernatants were loaded onto a 5-ml HisTrap HP column (Cytiva) equilibrated with binding buffer (10 mM Tris pH 8.0, 10 mM Imidazole, 300 mM NaCl, 5 mM β-mercaptoethanol and 1 mM PMSF). Nonspecifically bound proteins were removed by washing with 10 mM imidazole in binding buffer, before elution with 25 column volumes over a gradient from 10 to 300 mM imidazole. Fractions containing FLuc were treated with 8 μg of Ulp1 protease (prepared in-house) per 2 mg of protein during dialysis overnight into gel-filtration buffer (20 mM Tris pH 7.5, 150 mM NaCl, 5 mM β-mercaptoethanol and 0.2 M arginine–HCl). Cleaved His–SUMO was removed using reverse immobilized metal affinity chromatography in binding buffer containing no imidazole. Eluted fractions containing pure FLuc were concentrated using Vivaspin 50-kDa protein concentrators (Sigma) and injected directly onto a 16/60 Superdex 200pg column (Cytiva) equilibrated in gel-filtration buffer for further cleanup. FLuc fractions of >95% purity were pooled and buffer-exchanged into FLuc storage buffer (25 mM Tris-acetate pH 7.8, 1 mM EDTA, 0.2 M ammonium sulfate, 15% glycerol, 30% ethylene glycol and 2 mM TCEP) and flash-frozen for storage at −80 °C.

Before use, the protein was buffer-exchanged into the appropriate assay buffers using 7-kDa Zeba spin desalting columns (Thermo Fisher Scientific) as per the manufacturer's instruction and the product was centrifuged at 21,000g for 20 min at 4 °C to remove potential aggregated protein.

## Luciferase activity assays
RNC or FLuc were diluted to 0.2 μM in 10-μl reactions with assay buffer (20 mM HEPES pH 7.5, 100 mM KCl, 5 mM magnesium acetate, 1 mM DTT and 0.2 M arginine–HCl) and equilibrated at 25 °C for 10 min with either 5% (v/v) DMSO and 50 μM 2-phenylbenzothiazole (PBT) in DMSO or 12.5 μg ml⁻¹ RNaseA. Then, 50 μl of luciferase assay reagent (Promega) was added to each reaction and luminescence was immediately measured using a Glomax 20/20 luminometer (Promega) with delay time of 2 s. The relative luminescence units were corrected for pmol of protein in each reaction, calculated using $A_{260}$ (for RNC) or $A_{280}$ (for FLuc) measurements. Postreaction samples were concentration-normalized for gel loading and immunoblotted for FLuc NCs (anti-N-terminal FLuc) to confirm whether the tRNA-bound NCs remained intact.

## PK digestion assays
RNC or FLuc at 0.1 μM in HDX labeling buffer was treated with 0.5 μl of 8 mg ml⁻¹ 3C protease for 30 min on ice to remove 3×Flag. Next, 4 μl of 5 mM PMSF was prechilled on ice to quench PK (Sigma). Then, 30 μl of cleaved RNCs were incubated at 10 °C and 4 μl of the reaction was removed and quenched into 4 μl of PMSF as a −PK control. Then, 3 μl of PK was added to the reaction at a concentration of 1 ng μl⁻¹ to initiate the digest reaction. At 2.5, 5, 10, 20 and 30 min, 4 μl was removed from the reaction and quenched in prechilled PMSF. Finally, 8 μl of reducing Laemmli loading dye was added to each sample and proteins

were separated on 4–12% Bis–Tris SDS–PAGE gels, transferred to nitrocellulose membranes and western-blotted for FLuc.

Amounts of tRNA-bound FLuc remaining at each time point were measured by densitometry using ImageJ[71] and normalized to the −PK control. Data were plotted as a single-phase decay using GraphPad Prism 10.

## Cysteine labeling
RNC or FLuc was buffer-exchanged using Zeba 7-kDa spin desalting columns (Thermo Fisher Scientific) or diluted to 0.1 μM into HDX labeling buffer containing no DTT. For urea-denatured controls, 8 M urea was included in the HDX labeling buffer and incubated at RT for 1 h to unfold the protein. Stocks of F5M (Thermo Fisher Scientific) were produced in DMF and diluted 200-fold into HDX buffer before equilibration at 10 °C. Labeling was initiated by mixing 3 μl of F5M (0.1 mM final) with 30 μl of sample and incubated at 10 °C in a Thermomixer (Eppendorf) with gentle agitation at 300 rpm. After 30 s, 2.5 min, 10 min, 30 min and 60 min, 5 μl of the reaction was quenched with 5 μl of Laemmli sample buffer containing β-mercaptoethanol. Samples were resolved on a 15-well 4–12% Bis–Tris SDS–PAGE gel in MES–SDS running buffer. Fluorescently labeled bands were detected on a Typhoon-9500 with 488-nm laser excitation and 750 gain. No background correction was applied.

Labeled tRNA bands were identified by western blot of the same gel when probed for luciferase. Any loading errors were identified here and omitted from further analysis. Raw counts from fluorescent gels were quantified using ImageJ[71], normalized to largest degree of labeling in each reaction and fitted to a one-phase exponential plateau in GraphPad Prism 10. Labeling rates were calculated for individual reactions and plotted for comparison between samples.

Statistical significance between labeling rates was tested using an ordinary one-way analysis of variance between different conditions from at least three independent biological repeats. All data presentation and statistical analysis were carried out in GraphPad Prism 10.

## Cryo-EM
**Grid preparation and data collection.** Lacey carbon Au 300-mesh grids (Agar Scientific) were glow-discharged in air for 30 s with a GloQube instrument (Quorum Technologies). Then, 6 μl of purified Ns-RNCs at a concentration of 0.1 μM were applied to the grids before blotting with a Vitrobot Mark IV (Thermo Fisher Scientific) at 22.5 °C and 80% with a blot time of 3 s and a blot force of −1. Grids were plunge-frozen in liquid ethane and stored in liquid nitrogen.

Movies were collected on a 300-kV Titan Krios EM instrument (Thermo Fisher Scientific) with a Falcon 4i direct electron detector (Thermo Fisher Scientific) and a Selectris energy filter (Thermo Fisher Scientific) operating at a slit width of 10 eV. A total of 35,616 movies were collected with EPU version 3.8.1.7603 using aberration-free image shift and fringe-free illumination at a nominal magnification of ×130,000 for a pixel size of 0.95 Å per pixel. Data collection details are in Supplementary Table 1.

**Cryo-EM image processing.** Cryo-EM image processing is summarized in Extended Data Fig. 3. Movies were motion-corrected in RELION 4 (ref. 72) using MotionCor2 (ref. 73) and contrast transfer function (CTF) parameters were estimated with CTFFIND4 (ref. 74). Exposures were then transferred to cryoSPARC (version 4.4.1)[75] and curated on the basis of CTF fit and overall motion. In total, 29,080 exposures were selected and particle picking was performed in cryoSPARC with the blob picker utility. Particles were downsampled by a factor of 4 and extracted before two rounds of two-dimensional (2D) classification. Particles from well-resolved 2D classes were selected for ab initio reconstruction (one class) and homogeneous refinement. Particles were reextracted with a downsampling factor of 2 and used for homogeneous refinement followed by heterogeneous refinement

(four classes). Particles refining into a class with obvious signs of alignment on the Lacey carbon edge were excluded from further processing while good particles were retained for homogeneous refinement followed by three-dimensional (3D) classification (six classes, two O-EM epochs, 6-Å target resolution and principal component analysis initialization). Particles from the two best-resolved classes (788,554 particles) were taken forward for processing in RELION 5, where they were extracted with a downsampling factor of 2 and subjected to 3D refinement with Blush regularization[76]. This map was used as a reference to visualize flexibility of the P-site tRNA by performing 3D classification (four classes, $T = 100$, ten iterations, Blush regularization and no alignment) using a mask around the tRNA-binding site (Extended Data Fig. 4). All particles from the reference map were reextracted without downsampling and subjected to 3D refinement with blush regularization. Postprocessing, CTF refinement and particle polishing were carried out before 3D refinement (with Blush regularization) to yield the final consensus reconstruction.

**Model building.** The model of the human 80S ribosome (Protein Data Bank (PDB) 6QZP)[77] was rigid-body fitted into the consensus map in WinCoot (version 0.9.8.93)[78]. Regions of the model corresponding to the small subunit or regions that were absent in our map were removed and the XBP1u model and CCA tail of the P-site tRNA (PDB 6R5Q)[42] were rigid-body fitted independently into the density. The relevant XBP1u+ substitutions (L216I, Q223C and P224V) were introduced and the eight C-terminal residues of FLuc were added in WinCoot. Two ordered water molecules were modeled into clear density near the NC (interacting with the backbone nitrogen of R221 and carbonyl oxygen of W226; Extended Data Fig. 4f,g). Density for a third water molecule was seen near the side chain of K227 but the map quality was not high enough for unambiguous assignment of the water and side-chain densities (Extended Data Fig. 4h); hence, the water molecule was not modeled. The entire model was locally refined using Isolde (version 1.7.1)[79] in ChimeraX (version 1.7.1)[80]. Ions and water molecules were modeled and checked in WinCoot manually. The model was subjected to real-space refinement with phenix.real_space_refine using PHENIX (version 1.21)[81] for eight macrocycles using default parameters, with a custom covalent bond between the carbonyl carbon of XBP1u+ M230 and the 3′ oxygen of P-tRNA A76. Model validation was performed in PHENIX version 1.21 (Supplementary Table 1).

### Circular dichroism spectroscopy
Isolated Ns and FLuc were buffer-exchanged into 25 mM Na-phosphate pH 7.8, 10 mM magnesium acetate and 2 mM DTT using Zeba 7-kDa spin desalting columns. Isolated Ns was diluted to 0.15 mg ml⁻¹ and scanned in a 1-mm quartz cell from 260 nm to 200 nm in 0.2-nm increments using a Jasco J-815 circular dichroism spectrometer at 20 °C. FLuc at 0.82 mg ml⁻¹ was scanned in a 0.2-mm demountable cell from 260 nm to 180 nm in 0.2-nm increments. For both proteins, 25 scans were taken, averaged and corrected using a buffer blank recorded in the same cell. Machine units were converted to mean residue ellipticity using an average mean residue weight of 110 g mol⁻¹. Spectra were deconvoluted using the CDSSTR program[82] and compared to the SPD40 protein database. Secondary-structure content was estimated from PDB 1LCI for FLuc[38] and isolated NS in the context of folded FLuc using the KCD server[83].

For thermal melts, both proteins were diluted to 0.15 mg ml⁻¹ in a 1-mm quartz cell. Circular dichroism was measured at 220 nm over a temperature ramp from 20 °C to 75 °C for Ns and 20 to 95 °C for FLuc.

### SEC–MALLS
SEC–MALLS was used to determine the oligomeric state of isolated Ns. Then, 100 µl of Ns at 0.1 or 0.2 mg ml⁻¹ was applied to a Superdex 200, 10/30 increase GL column (Cytiva) equilibrated in 20 mM HEPES pH 7.5, 100 mM KCl, 5 mM magnesium acetate, 200 mM arginine–HCl, 1 mM DTT and 2 mM sodium azide at a flow rate of 1 ml min⁻¹.

Scattered light intensity and protein concentration were recorded using a DAWN-HELIOS-II laser photometer and an OPTILAB-TrEX differential refractometer respectively. The weight-averaged molecular mass of the eluate was determined using combined data from both detectors in the ASTRA software.

### Mass photometry
Isolated Ns was diluted to 50 nM in HDX labeling buffer (20 mM HEPES pH 7.5, 100 mM KCl, 5 mM magnesium acetate, 200 mM arginine–HCl and 1 mM DTT). Solution phase masses were measured on a TwoMP mass photometer (Refeyn). Videos were recorded on precleaned, high-sensitivity microscope slides. Mass calibration was determined using urease, BSA and aldolase covering a mass range of 66 to 544 kDa. Each standard was used at 2–5 nM and diluted tenfold into PBS for data acquisition. For Isolated Ns, 50 nM protein was diluted tenfold in the HDX labeling buffer. Videos were recorded for 1 min and analyzed using DiscoverMP version 2.5 software.

### XL-MS
RNCs (approximately 200 µl of 0.4 µM) for XL-MS were buffer-exchanged into XL buffer (20 mM HEPES pH 7.5, 100 mM KCl, 5 mM magnesium acetate and 1 mM DTT) using Zeba 7-kDa spin desalting columns. For ligand-bound conformations, 50 µM PBT and 5 mM ATP were included in this buffer. XL was initiated by the addition of 1 mM MS-cleavable DSBU (Thermo Fisher Scientific) at 25 °C for 1 h with gentle agitation. XL was quenched with the addition of 20 mM Tris-HCl pH 8.0 and incubated for a further 15 min. XL-MS data were collected and processed as described previously[67]. Crosslinked proteins were reduced (10 mM DTT), alkylated (50 mM iodoacetamide) and digested with trypsin (enzyme-to-substrate ratio of 1:100, 1 h, 25 °C) followed by a further digestion (enzyme-to-substrate ratio of 1:20, 18 h, 37 °C). Tryptic peptides were fractionated into 5 fractions (10%, 20%, 30%, 40% and 80% acetonitrile in 10 mM NH₄HCO₃, pH 8) using high-pH reverse-phase chromatography on TARGA C18 columns (Nest Group), lyophilized and resuspended in 1% formic acid and 2% acetonitrile. Samples were then subjected to nanoscale capillary liquid chromatography (LC)–MS/MS analysis using the Vanquish Neo ultrahigh-performance LC (UPLC) instrument (Thermo Fisher Scientific Dionex) at a flow rate of 250 nl min⁻¹, a C18 PepMap Neo nanoViper trapping column (5 µm, 300 µm × 5 mm, Thermo Fisher Scientific Dionex), an EASY-Spray column (50 cm × 75 µm inner diameter, PepMap C18, 2-µm particles, 100 Å pore size; Thermo Fisher Scientific) and quadrupole Orbitrap MS instrument (Orbitrap Exploris 480, Thermo Fisher Scientific). Acquisition parameters were set to data-dependent mode using a top ten method, recording a high-resolution full scan ($R = 60,000$, $m/z$ 380–1,800) followed by higher-energy collision dissociation (HCD) of the ten most intense MS peaks, excluding ions with precursor charge states of 1+ and 2+.

For data analysis, Xcalibur raw files were converted to MGF format using Proteome Discoverer 2.3 and used directly as input files for MeroX[84]. Searches were performed against an ad hoc protein database containing sequences of the 80S ribosome, HspA1A, EIF6 and FL-RNC, as well as a set of random decoys sequences generated by the software. Crosslinks were analyzed using XiView[85] and intra-FLuc crosslinks were visualized using PyLink Viewer[86]. Crosslinks were filtered by a match score > 50.

### Mass spectrometry analysis of RNC composition
RNCs or ribosomes were purified as above in high-salt RNC buffer and processed and collected as detailed previously[19,67]. Then, 5–10 µg of RNC or protein was separated in 8 mm on NuPAGE 1 mm 12% Bis–Tris 10-well or 12-well SDS–PAGE gels before Coomassie blue staining (quick Coomassie stain, Generon). Bands were destained in extraction buffer (50% acetonitrile, 100 mM ammonium bicarbonate and 5 mM DTT; 16 h, 4 °C). Samples were then alkylated (40 mM chloroacetamide,

160 mM ammonium bicarbonate and 10 mM TCEP; 20 min, 70 °C), dehydrated in 100% acetonitrile and air-dried at 37 °C, followed by digestion with trypsin (Promega). Digests were loaded onto Evotips and tryptic peptides were eluted with the '30 SD' gradient using an Evosep One high-performance LC instrument fitted with a 15-cm C18 column into a Lumos Tribrid Orbitrap MS instrument using a nanospray emitter operated at 2,200 V. The Orbitrap was operated in data-dependent acquisition mode with precursor ion spectra acquired at 120,000 resolution in the Orbitrap and MS/MS spectra in the ion trap at 32% HCD collision energy in 'TopS' mode. Dynamic exclusion was set to ±10 ppm over 15 s, automatic gain control was set to 'standard' and max injection time was set to 'dynamic'. The vendors universal method was adopted to schedule the ion trap accumulation times.

Raw files were processed using MaxQuant 2.0.3.0 and Perseus with a recent download of the UniProt *Homo sapiens* reference proteome (UP000005640) or *E. coli* reference proteome (UP000000625), and the sequence of the FL-RNC based on UniProt Q27758_PHOPY, including the N-terminal tag, C-terminal linker and XBP1u+ or SecM[Str] stalling sequence. A decoy dataset of reverse sequences was used to filter false positives, with both peptide and protein false discovery rate set to 1%. Quantification of individual proteins was achieved using iBAQ (intensity-based absolute quantification) and values were normalized to the mean iBAQ of all 50S or 70S proteins in each sample. Therefore, each protein could be quantified as a percentage per ribosome in the sample.

## HDX-MS

**RNC labeling, quench and digestion.** RNC labeling, quench and offline digest were carried out as previously described[19,67] with some differences. In brief, 60 μl of 0.2–0.5 μM RNC, FLuc or ribosome was labeled in 20 mM HEPES pH 7.5, 100 mM KCl, 5 mM magnesium acetate, 1 mM DTT and 0.2 M arginine–HCl made up with water or $D_2O$. Deuterated buffers contained 98% $D_2O$. Labeling was initiated using 0.5-ml Zeba 7-kDa spin desalting columns, preequilibrated with labeling buffer as per the manufacturer's instructions. Exchanged proteins were eluted by centrifugation at 1,000*g* for 2 min at 25 °C and incubated at 25 °C with 350-rpm shaking in an Eppendorf ThermoMixer. Then, 50 μl of exchanged material was quenched at 3 min or 60 min with 20 μl of quench buffer (175 mM sodium phosphate pH 2.1, 17.5 mM TCEP and 5 M guanidinium chloride, adjusted to pH to 1.2 with orthophosphoric acid; final pH after quench of 2.3) on ice and immediately added to 20 μl of 50% pepsin POROS slurry[67] preincubated at 10 °C. Offline RNC digest was carried out at 10 °C for 2 min with 450-rpm shaking, with brief vortexing every 30 s. The digested peptides were eluted through a 0.2-μm PVDF centrifugal filter (Durapore) at 0 °C for 30 s at 16,000*g* before flash-freezing and storage at −80 °C or immediate thaw and injection into a Synapt G2-Si (Waters).

Maximally deuterated samples were prepared as previously described[87]. Undeuterated samples were digested as above and hydrogen-containing solvent was removed overnight using a Savant ISS110 Speedvac (Thermo Fisher Scientific). The pellets containing peptides were resuspended in 50 μl of deuterated labeling buffer for 4 h at 40 °C, before adding 20 μl of quench buffer on ice and incubating at 10 °C for 2 min before flash-freezing in liquid nitrogen.

Labeled and quenched peptides were thawed and immediately injected into a Waters M-class UPLC instrument coupled to a Synapt G2-Si HDMS[E] MS instrument. All chromatographic elements were held at 0 °C in the UPLC chamber. Peptides were trapped and desalted on a VanGuard precolumn trap (2.1 mm × 5 mm Acquity UPLC BEH C4 1.7 μm (Waters)) for 4 min at 200 μl min⁻¹. Peptides were eluted in a gradient of 3–30% acetonitrile in 0.1% formic acid over 25 min across at a flow rate of 90 μl min⁻¹, and separated using an Acquity UPLC HSS T3 1.8 μm × 5 mm 1.8-μm LC column (Waters). Mass spectra were acquired using Waters Synapt G2-Si in ion-mobility (IMS) mode. The MS instrument was calibrated with direct infusion of a solution of glutamate fibrinopeptide at 50 fmol μl⁻¹ at 5 μl min⁻¹. A conventional electrospray source was used and the instrument was scanned over a range of $m/z$ 50–2,000. The instrument was configured as follows: capillary at 3 kV, trap collision energy at 4 V, sampling cone at 40 V, source temperature of 80 °C and desolvation temperature of 180 °C. All experiments were carried out under identical experimental conditions in triplicate and were not corrected for back exchange; therefore, the results are reported as relative.

After each injection, a trap column and LC wash was injected containing 1.5 M guanidinium chloride, 4% (v/v) acetonitrile and 0.8% (v/v) formic acid, with a peptide trapping time of 3 min and eluted with three repeat ramps of 5–95% acetonitrile over 12 min at 90 μl min⁻¹.

**Data analysis.** Peptides were identified from HDMS[E] analyses of 3–8 replicate undeuterated control samples across each RNC, off-ribosome FLuc and empty 80S ribosomes using PLGS 3.0.3 (Waters). Peptide masses were identified from searches using nonspecific cleavage of a custom database containing the sequence of FL-RNC based on UniProt Q27758_PHOPY and all ribosomal proteins identified and extracted from proteomics datasets. In addition, the database contained HspA1A and EIF6, which were also identified in RNC samples. Searches used the following parameters: no missed cleavages, no post-translational modifications (although methionine oxidation was allowed), a low energy threshold of 135, an elevated energy threshold of 35, a lock mass window of 0.25 Da, three minimum fragment ion matches per peptide and seven minimum fragment ion matches per protein. No false discovery rate correction was performed. Peptides identified in PLGS for each RNC were aggregated and further filtered using DynamX 3.0 (Waters) using a maximum sequence length of 40, 0.05 minimum products per amino acid and a minimum of one consecutive product identified. Only peptides that were identified in the majority of undeuterated replicates and in off-ribosome FLuc were retained. In addition, the resulting FLuc peptide list was searched in DynamX against control data collected for empty 80S ribosomes. Putative FLuc peptides that were detected in empty 80S were removed. Each mass spectrum was manually inspected for spectral overlap and signal to noise and peptides of unacceptable quality were removed. The relative amount of deuterium was determined with the software by subtracting the centroid mass of the undeuterated from the deuterated spectra at each time point for each condition. Spectra showing multimodal uptake kinetics were further analyzed using HX-Express3 (ref. 88). Uptake values were used to generate all uptake graphs and difference maps using GraphPad Prism 10.1.1 (Supplementary Data 3, 5, 6 and 7). Data were collected in at least technical triplicate. The s.d. associated with every differential HDX measurement is reported in Supplementary Data 3, 5, 6 and 7. The average s.d. across all peptides was 0.11 Da, with an interquartile range of 0.04–0.14 Da for hRNCs. The s.d. was 0.10 Da for bRNCs with an interquartile range of 0.03–0.13 Da.

## Reporting summary

Further information on research design is available in the Nature Portfolio Reporting Summary linked to this article.

## Data availability

Proteomic analysis used the UniProt *H. sapiens* (UP000005640) or *E. coli* (UP000000625) reference proteome. All MS data were deposited to the ProteomeXchange Consortium through the PRIDE partner repository with the following dataset identifiers: hRNC composition, PXD055069; XL-MS, PXD055251; HDX-MS analysis of hRNCs, PXD055280; bRNC composition, PXD055060; HDX-MS analysis of bRNCs, PXD055476. The cryo-EM map was deposited to the Electron Microscopy Data Bank (EMDB) and the coordinates were deposited to the PDB under the following accession numbers: EMD-51611 and PDB 9GUL. Any other data and materials can be obtained from the corresponding authors upon request. Source data are provided with this paper.

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

## Acknowledgements

We thank D. Briggs for help with cell culture, S. Mouilleron for purified 3C protease, S. Garcia-Manyes for the Ulp1 vector, C. Soudy for help with preparing pepsin beads, S. Kunzelmann for help with circular dichroism deconvolution, S. Howell for proteomic analysis, G. Kelly for preliminary NMR analysis, E. Couves for cryo-EM advice and all members of the Protein Biogenesis Lab for help and discussion. This work was supported by funding from UK Research and Innovation (FoldingMap, EP/X020428/1, to D.B.) and the Francis Crick Institute, which receives its core funding from Cancer Research UK (CC2025, CC1063 and CC1068), the UK Medical Research Council (CC2025, CC1063 and CC1068) and the Wellcome Trust (CC2025, CC1063 and CC1068).

## Author contributions

G.P. performed the biochemical and HDX-MS experiments and analyzed the data. T.V. collected and analyzed the cryo-EM data. L.K. contributed to optimizing the RNC purification. A.R. contributed to optimizing the HDX-MS workflows. C.R. and S.K. assisted with the cell culture. R.G. performed the mass photometry. S.L.M. and J.M.S. collected and processed the XL-MS data and supported the HDX-MS experiments. A.N. performed the Krios data collection. I.A.T. performed the SEC–MALLS. D.B. conceptualized and supervised the project and wrote the manuscript together with G.P. and T.V.

## Funding

## Competing interests

The authors declare no competing interests.

## Additional information

**Extended data** is available for this paper at https://doi.org/10.1038/s41594-025-01676-5.

**Correspondence and requests for materials** should be addressed to David Balchin.

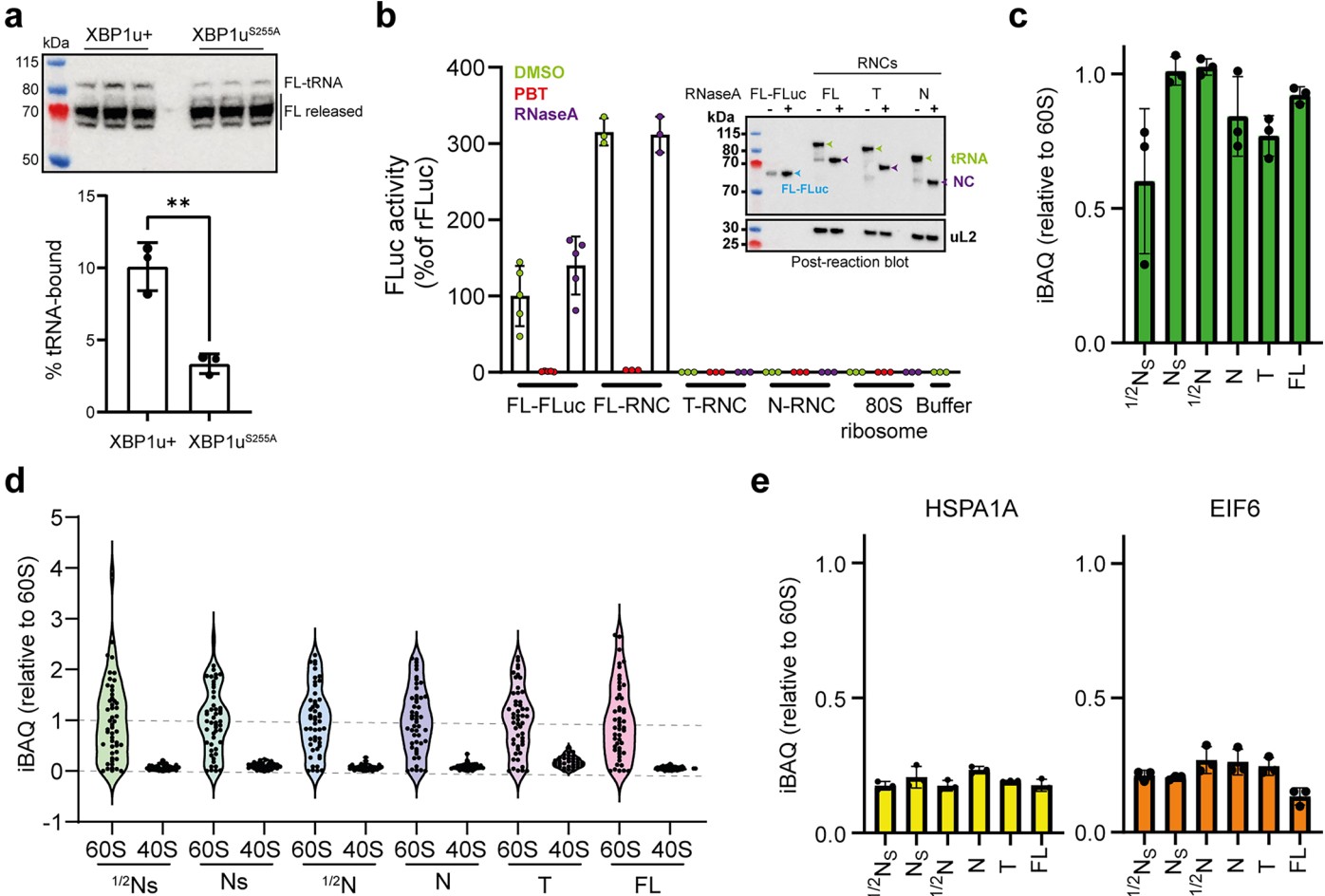

**Extended Data Fig. 1 | Quality control of FLuc RNCs. a**, Comparison of stalling efficiency between XBP1u+ and XBP1u$^{S255A}$. FL-RNC constructs prepared with either stalling sequence were expressed in Expi293 cells. Clarified lysates were resolved by SDS-PAGE and immunoblotted using an antibody against the N-terminus of FLuc. The intensity of the NC-tRNA band was measured using ImageJ, and expressed as a percentage of the total FLuc signal. Data represent the mean ± s.d.; n = 3 biological replicates. p = 0.0030**, two-tailed unpaired t-test. **b**, FL-RNC is enzymatically active. Luminescence activity is expressed as a percentage of the activity of FL-FLuc. Where indicated, samples were treated with RNase/EDTA to release the NC from ribosomes, or PBT to inhibit FLuc activity. Inset: control immunoblot (anti-FLuc) of post-reaction samples, showing that the NC-tRNA is intact. Note that the apparent difference in enzyme activity between FL-FLuc and FL-RNC may result from differences in concentration, which

is estimated using different approaches for off-ribosome and RNC samples (see Methods). Data represent the mean ± s.d.; n = 5 technical replicates for FL-FLuc, and n = 3 technical replicates for RNCs. **c**, Fractional occupancy of FLuc in RNCs. Intensity-based absolute quantification (iBAQ) of NC peptides in each RNC, normalised to the average iBAQ of 60S ribosomal proteins. Data represent the mean ± s.d.; n = 3 technical replicates. **d**, Quantification of ribosomal proteins in RNCs. iBAQ values for each protein were normalised to the average iBAQ of 60S ribosomal proteins. 60S and 40S proteins are grouped separately. Data represent the mean ± s.d.; n = 3 technical replicates. **e**, Fractional occupancy of HspA1A (Hsp70) and EIF6 in RNCs. iBAQ values for HspA1A or EIF6 peptides in each RNC were normalised to the average iBAQ of 60S ribosomal proteins. Data represent the mean ± s.d.; n = 3 technical replicates. See also Data S1.

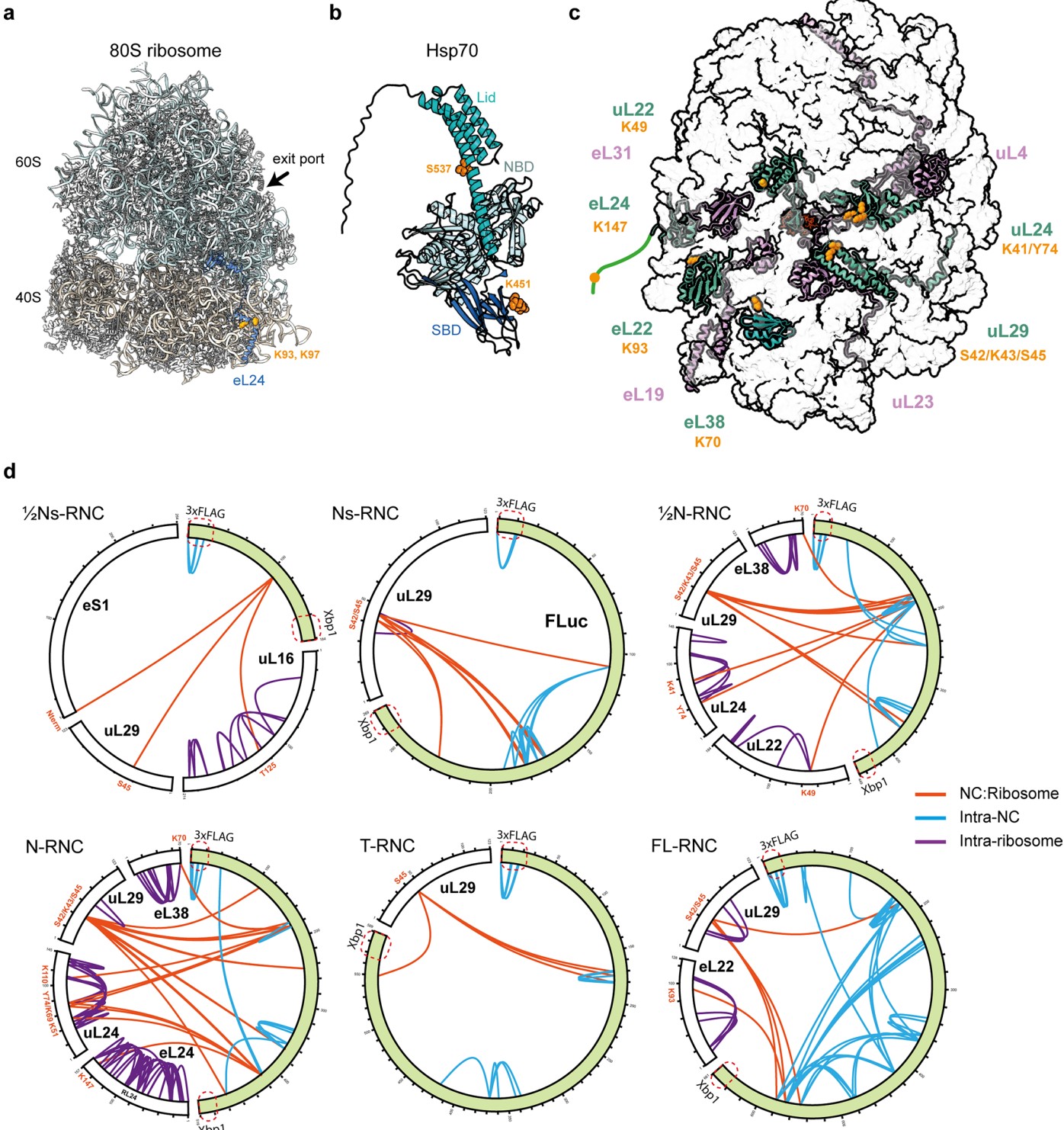

**Extended Data Fig. 2 | Crosslinking-mass spectrometry (XL-MS) analysis of RNCs. a**, Hsp70 crosslink sites on the ribosome (PDB 8QOI ref. 89). Residues on eL24 (blue) which crosslink to Hsp70 are shown as orange spheres. The ribosomal exit port is indicated. **b**, Resides on Hsp70 that crosslinked to eL24 are shown as orange spheres. Hsp70 is shown in the "open" (domain-docked) conformation predicted by Alphafold2 (AF-P0DMV8-F1). **c**, NC crosslink sites on the ribosome.

Ribosomal protein residues that crosslinked to any FLuc NC are shown as orange spheres on the structure of Ns-RNC. The N-terminal tail of eL24 containing K147 is not resolved in the structure. **d**, Map of RNC crosslinks identified by MS. NC:ribosome crosslinks (orange), intra-NC crosslinks (blue) and intra-ribosome crosslinks (purple) are shown. See also Supplementary Data S2.

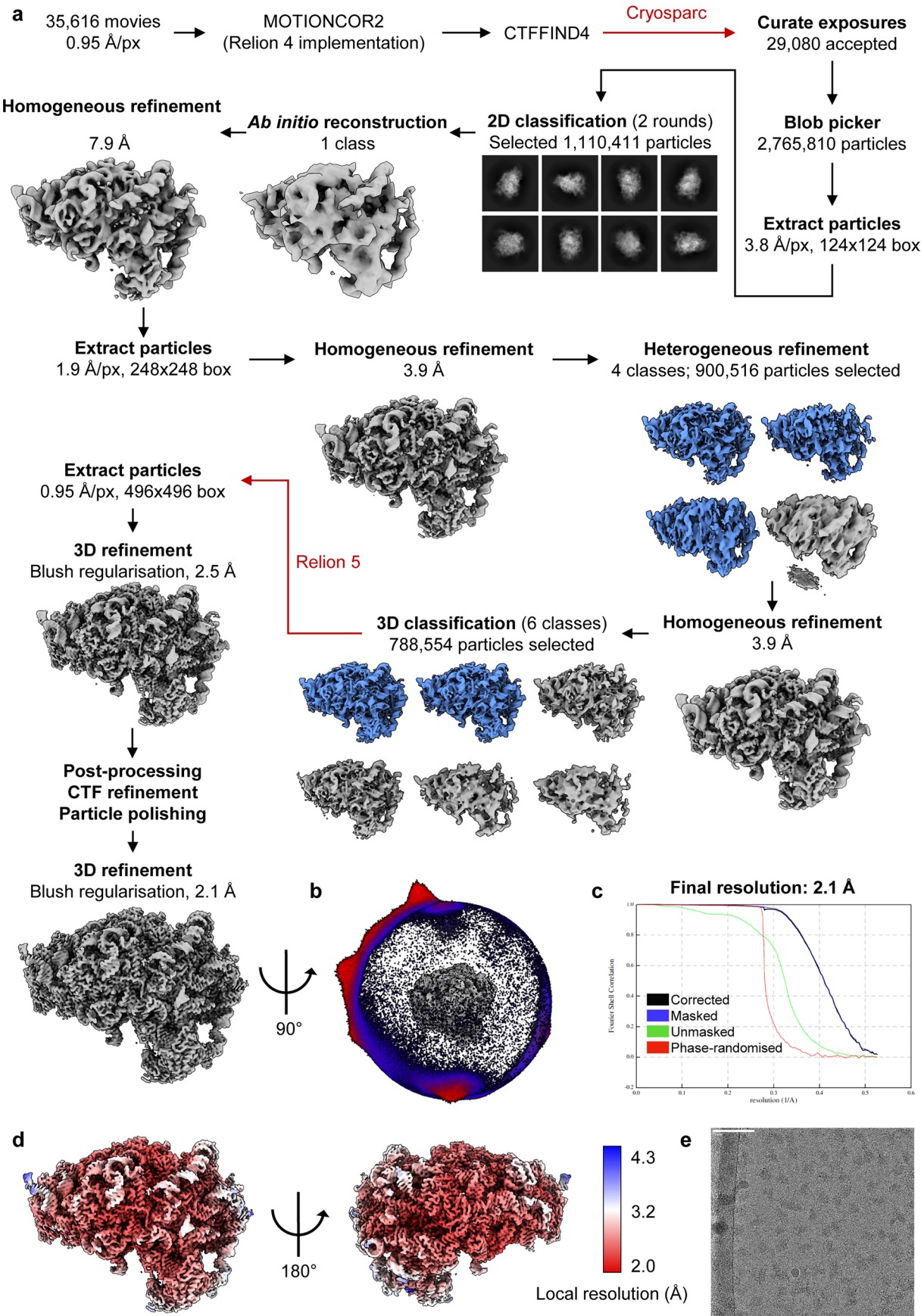

**Extended Data Fig. 3 | CryoEM image processing. a**, Schematic of the cryoEM processing pipeline including representative 2D classes and the final reconstruction. **b**, Angular distribution of the final cryoEM map. **c**, Fourier Shell Correlation curve of the cryoEM reconstruction. **d**, CryoEM map coloured according to local resolution, from 2.0 Å (red) to 4.3 Å (blue). **e**, Representative micrograph from 29,080 in total (white scale: 75 nm).

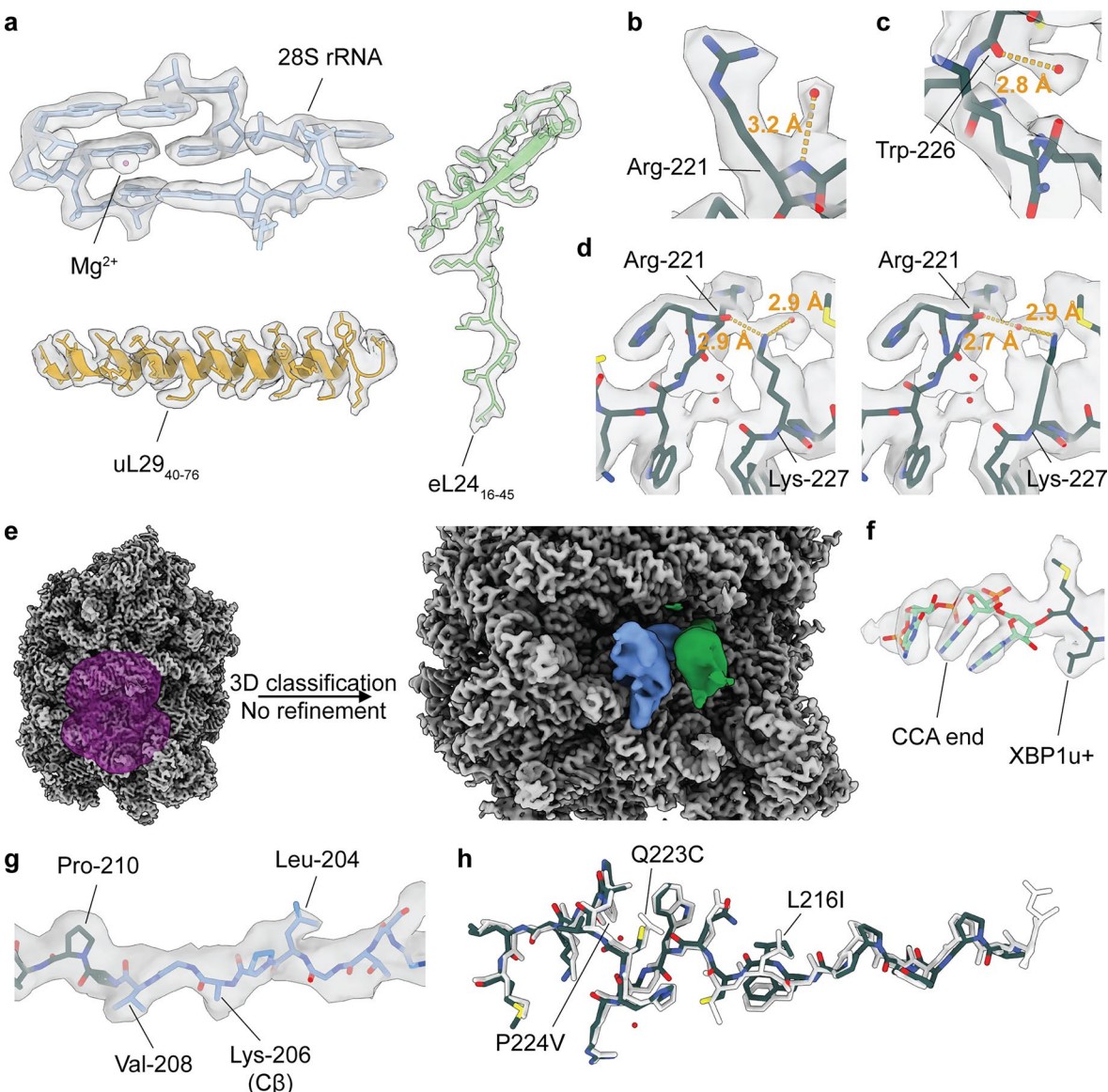

**Extended Data Fig. 4 | Quality of the cryoEM reconstruction and model building. a**, Map-model overlay of example regions of the structure. **b**, Map-model overlay of XBP1u+ showing an ordered water molecule interacting with the backbone nitrogen of Arg-221. **c**, Map-model overlay of XBP1u+ showing an ordered water molecule interacting with the carbonyl oxygen of Trp-226. **d**, Map-model overlay of XBP1u+ with the side chain of Lys-227 modelled to form a hydrogen bond with the carbonyl oxygen of Arg-221 and an ordered water molecule (left) and with the ordered water molecule placed between Arg-221 and Lys-227 (right). As the position of the water molecule could not be unambiguously determined, it was not modelled in the final structure. Distances in orange in B, C and D correspond to the hydrogen bonds marked by the orange lines. **e**, 3D classification with a mask around the tRNA sites (purple) shows P-tRNA flexibility. The P-tRNA density from two example classes (blue, green) is shown in the context of the consensus map (grey). **f**, Continuous density between the P-tRNA CCA end and XBP1u+ in the consensus map. **g**, Map-model overlay of the nascent chain with the side chain densities used to trace the backbone labelled. Due to the lack of density, the Lys-206 was trimmed to its Cβ carbon. **h**, Structural alignment of the XBP1u$^{S225A}$ arrest peptide (white) and the stalling-enhanced XBP1u+ (coloured).

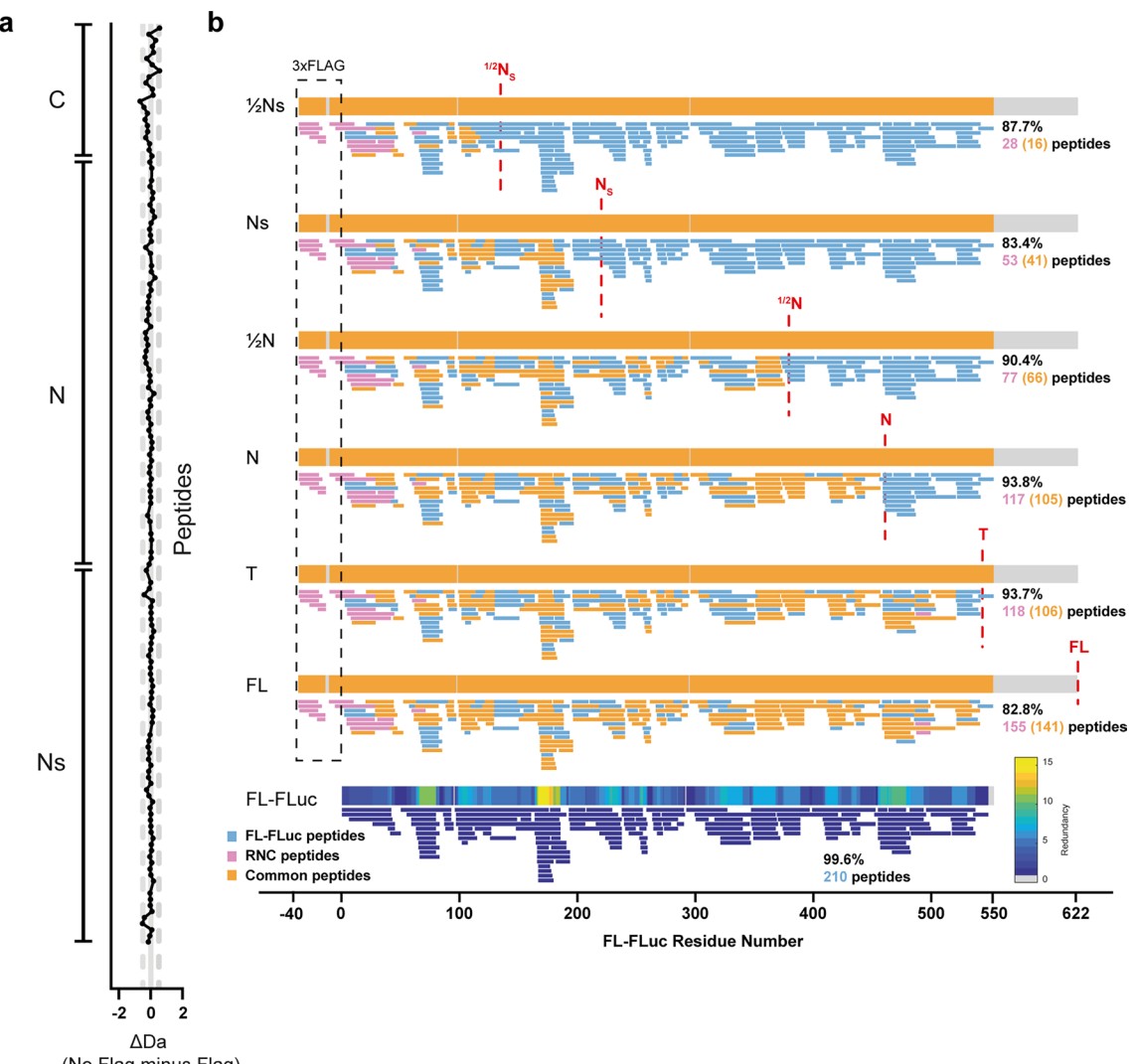

**Extended Data Fig. 5 | HDX-MS analysis of RNCs. a**, FLAG tag does not measurably affect the conformation of FL-RNC. FL-RNC was treated with 3 C protease to remove the 3xFLAG tag, then analysed by HDX-MS using a 3 min deuteration pulse. **b**, Peptide coverage maps for RNCs and FL-FLuc. Peptides found in FL-FLuc are coloured blue, peptides found in RNCs are coloured pink, and peptides common to both an RNC and FL-Fluc are coloured orange. Plots were made using Deuteros[90].

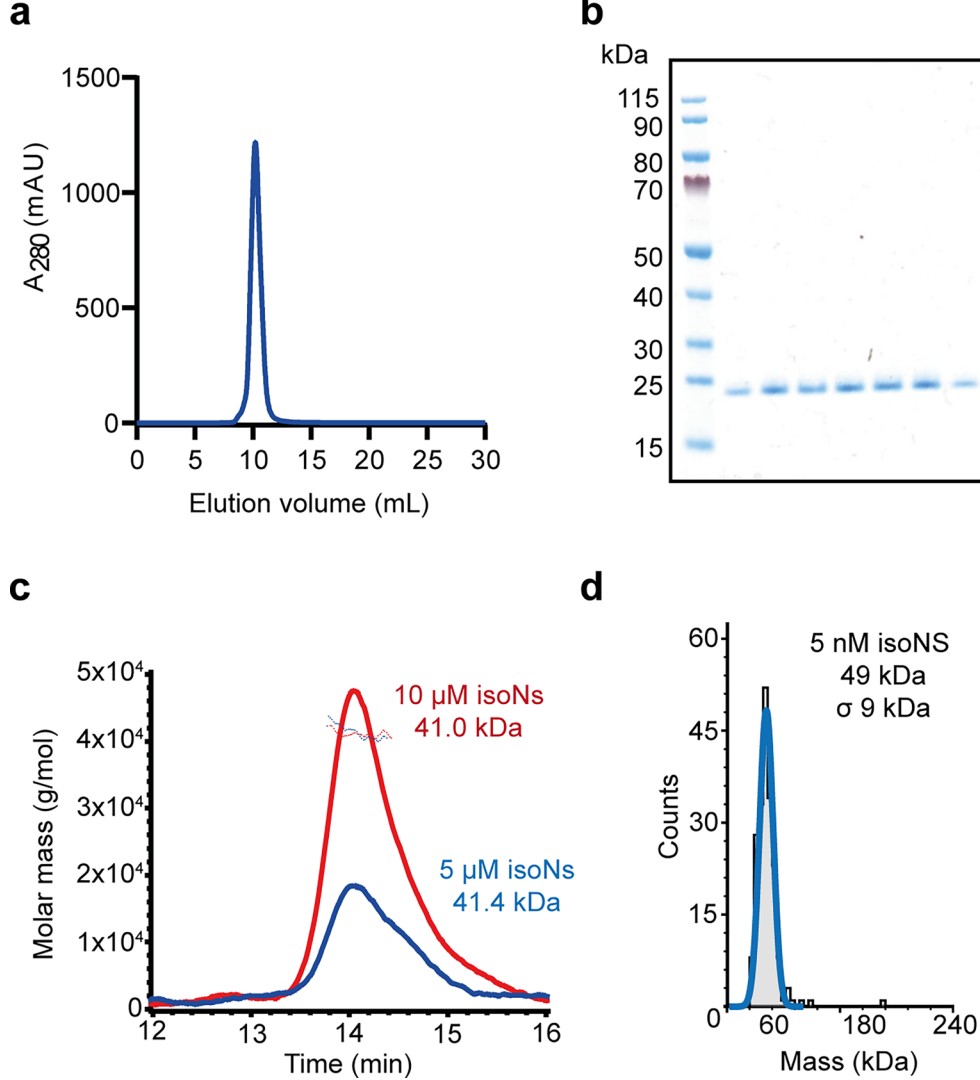

**Extended Data Fig. 6 | Characterisation of isolated Ns (isoNs). a**, Gel filtration analysis using a Superdex 75 column indicates that purified isoNs is homogeneous. **b**, Fractions from the gel filtration peak in **a** were resolved by SDS-PAGE and stained with Coomassie. The experiment was repeated twice using independent protein purifications, with similar results. **c**, SEC-MALLS analysis of isoNs at 0.1 or 0.2 mg/mL (~5 or 10 μM) indicates that isoNs is dimeric.

Chromatograms are differential refractometer output, and points represent calculated molar mass. The expected mass of the monomer is ~21.5 kDa. **d**, Analysis of 5 nM isoNs by mass photometry indicated an absolute mass of 49 kDa, σ 9 kDa, consistent with a dimer. Note that a monomeric peak at ~21 kDa is not expected to be resolved.

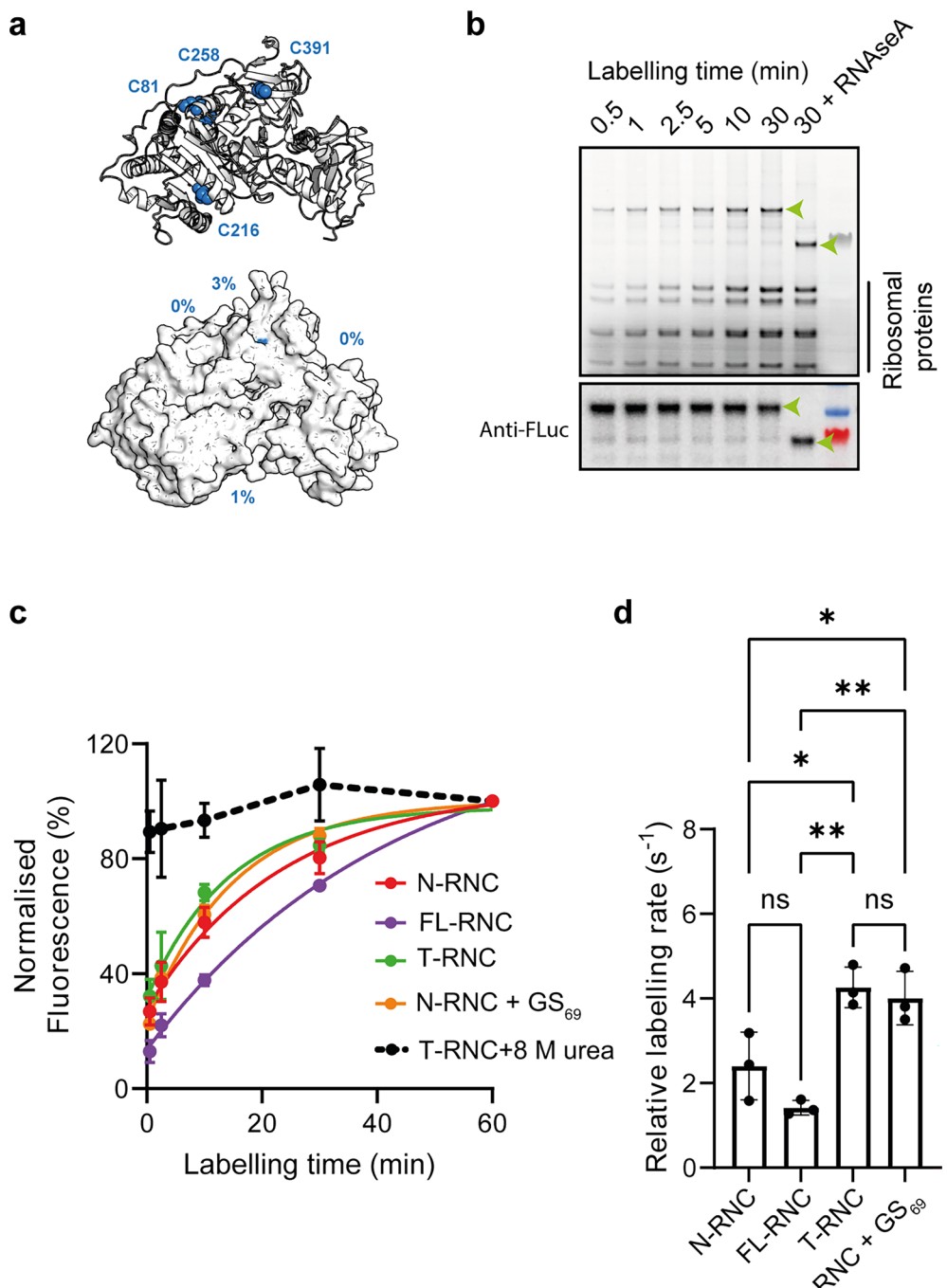

**Extended Data Fig. 7 | Analysis of NC conformation by cysteine labelling.**
**a**, Cys are buried in the native state of FL-FLuc. Cys residues in FLuc are shown as blue spheres. % solvent accessibility of each residue was calculated using PyMol. **b**, Fluorescein-5-malemide (F5M) labelling of T-RNC. Reactions were resolved by SDS-PAGE and imaged for in-gel fluorescence. The band corresponding to the NC is indicated using a green arrow. Below, the same gel was probed using an antibody against FLuc as a loading control. **c**, Cys labelling kinetics. The degree of NC labelling was quantified from gels as in **b** using ImageJ, and normalised to the maximum value. As a control, T-RNC was denatured in 8 M urea before

labelling. Data were fit to a single-exponential equation. Data represent the mean ± s.d.; n = 3-6 independent labelling reactions, depending on the construct. See source data for exact n. and the resulting labelling rates are shown in **d**. Cysteine labelling rates. Data represent the mean ± s.d.; n = 3 independent labelling reactions. P-values were determined using a 1-way ANOVA. N-RNC vs T-RNC, p = 0.0167*; N-RNC vs FL-RNC, p = 0.2264 ns; N-RNC vs N-RNC + GS69, p = 0.0351*; T-RNC vs FL-RNC, p = 0.0013**; T-RNC vs N-RNC + GS69, p = 0.9461 ns; FL-RNC vs N-RNC + GS69, p = 0.0023**.

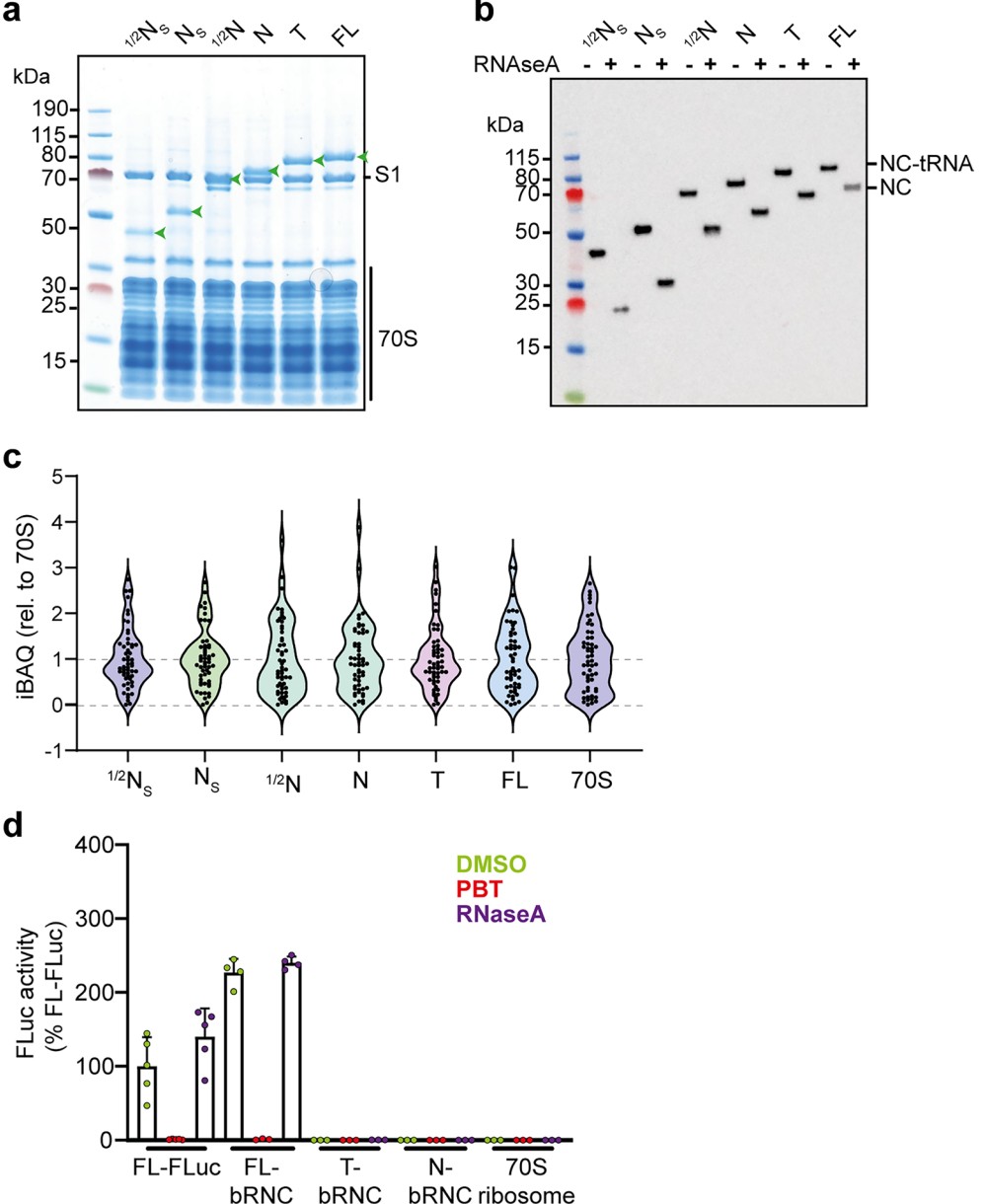

**Extended Data Fig. 8 | Quality control of FLuc bacterial RNCs (bRNCs).**
**a**, Coomassie-stained SDS-PAGE of purified bRNCs. Bands corresponding to 70S ribosomal proteins, ribosomal protein S1, and the NC linked to peptidyl-tRNA (green arrows) are indicated. The experiment was repeated using 3 independent purifications, with similar results. **b**, Anti-FLuc immunoblot of purified RNCs. RNase/EDTA treatment confirms that the NCs are covalently linked to peptidyl-tRNA. The experiment was repeated using 3 independent purifications, with similar results. **c**, Spread of iBAQ values for 70S ribosomal proteins in each bRNC. Purified empty ribosomes (70S) are shown as a control. **d**, FL-bRNC is enzymatically active. Luminescence activity is expressed as a percentage of the activity of FL-FLuc. Where indicated, samples were treated with RNaseA/EDTA to release the NC from ribosomes, or PBT to inhibit FLuc activity. Data represent the mean ± s.d.; n = 5 technical repeats for FL-FLuc and n = 3 technical repeats for bRNCs and ribosomes. See also Data S4.

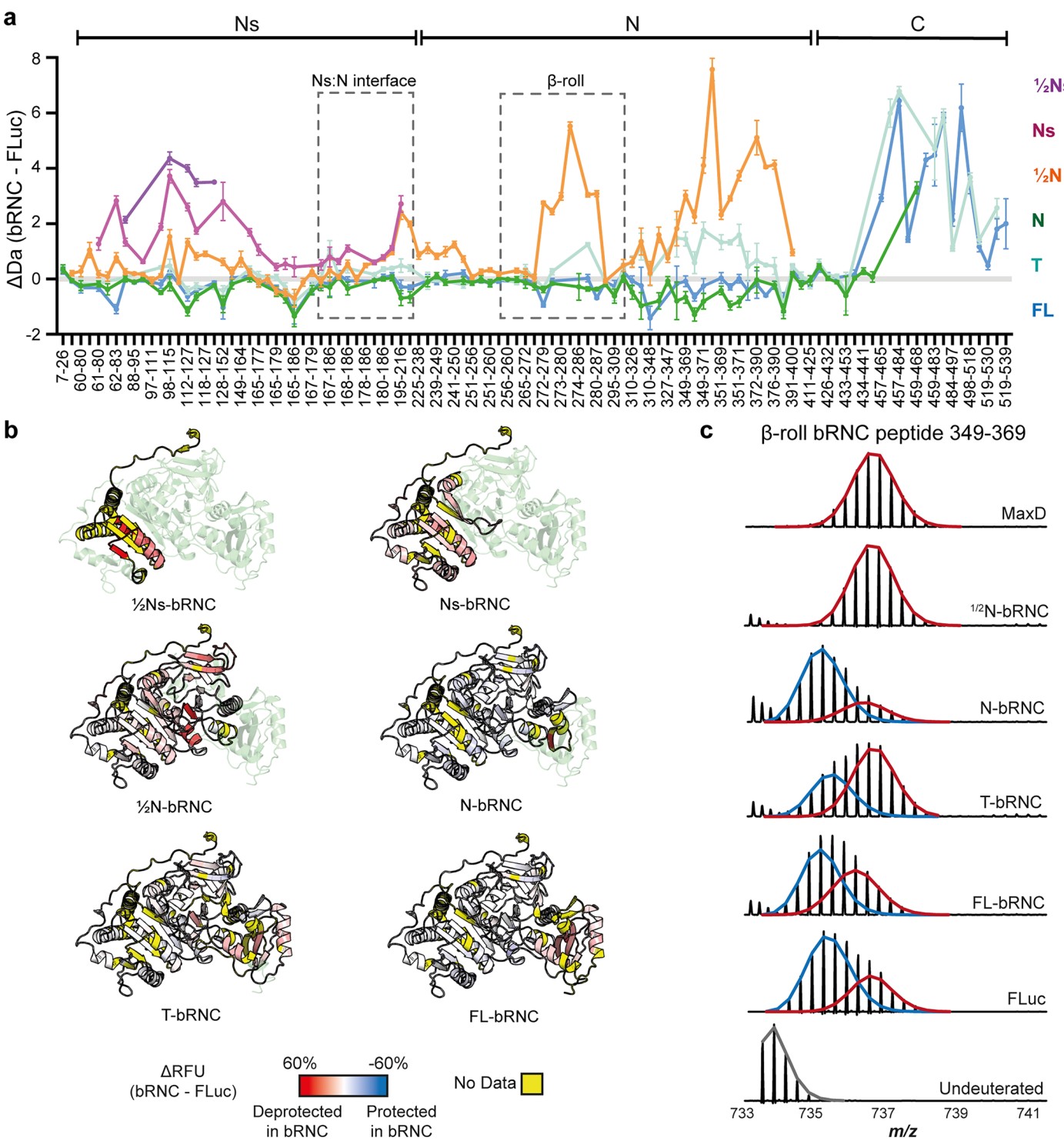

**Extended Data Fig. 9 | HDX-MS analysis of bRNCs. a**, Difference in deuterium uptake, after 3 min deuteration, between FLuc NCs on the bacterial ribosome and native FL-FLuc. Larger values indicate more deuteration of NCs compared to FL-FLuc. Data represent the mean ± s.d.; n = 3 independent labelling reactions. Every 2nd peptide is labelled. **b**, Difference in relative fractional uptake (ΔRFU), after 3 min deuteration, between each NC and FL-FLuc. Data are mapped onto the Alphafold2 model FL-FLuc. Darker red indicates increased deuteration of NCs

compared to FL-FLuc. Regions without peptide coverage are coloured yellow. **c**, Mass spectral envelopes for peptide 349-369 in the β-roll of FLuc and bRNCs. Isotopic distributions were fit to single or bimodal gaussian distributions using HX-Express3.0[65]. The high-exchanging population (deprotected) is coloured red and the low-exchanging population (protected) is coloured blue. Except for the maximally deuterated sample (MaxD), samples were deuterated for 3 min. See also Data S5.

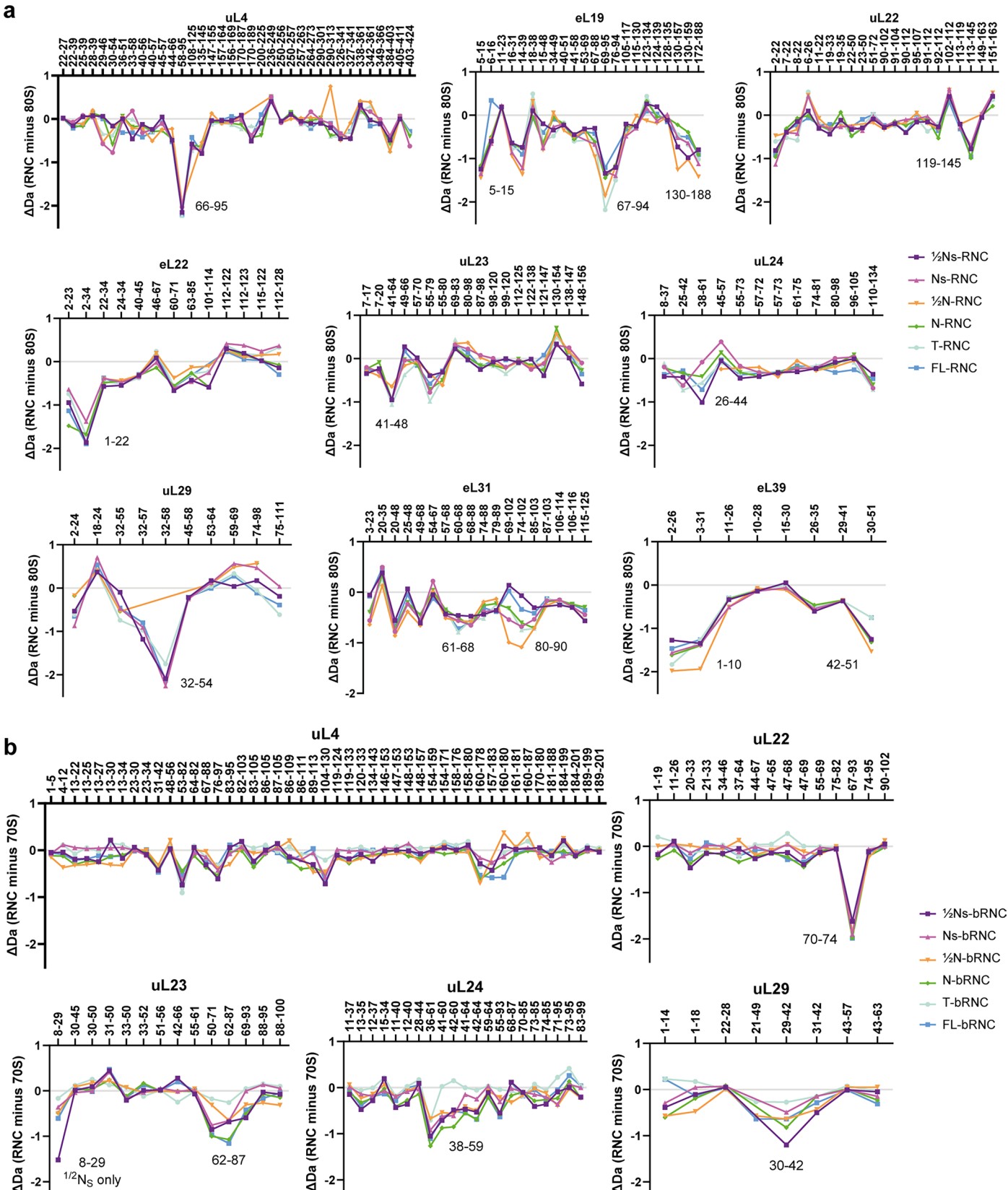

**Extended Data Fig. 10 | HDX-MS analysis of human and bacterial ribosomal proteins. a**, Difference in deuterium uptake, after 3 min deuteration, between ribosomal proteins in empty human 80S ribosomes and different human RNCs. **b**, As in **a**, for bRNCs compared to empty 70S ribosomes. See also Supplementary Data S6 and S7.

# Reporting Summary

## Statistics

For all statistical analyses, confirm that the following items are present in the figure legend, table legend, main text, or Methods section.

| n/a | Confirmed | |
|---|---|---|
| ☐ | ☒ | The exact sample size (*n*) for each experimental group/condition, given as a discrete number and unit of measurement |
| ☐ | ☒ | A statement on whether measurements were taken from distinct samples or whether the same sample was measured repeatedly |
| ☐ | ☒ | The statistical test(s) used AND whether they are one- or two-sided<br>*Only common tests should be described solely by name; describe more complex techniques in the Methods section.* |
| ☒ | ☐ | A description of all covariates tested |
| ☒ | ☐ | A description of any assumptions or corrections, such as tests of normality and adjustment for multiple comparisons |
| ☐ | ☒ | A full description of the statistical parameters including central tendency (e.g. means) or other basic estimates (e.g. regression coefficient) AND variation (e.g. standard deviation) or associated estimates of uncertainty (e.g. confidence intervals) |
| ☐ | ☒ | For null hypothesis testing, the test statistic (e.g. *F*, *t*, *r*) with confidence intervals, effect sizes, degrees of freedom and *P* value noted<br>*Give P values as exact values whenever suitable.* |
| ☒ | ☐ | For Bayesian analysis, information on the choice of priors and Markov chain Monte Carlo settings |
| ☒ | ☐ | For hierarchical and complex designs, identification of the appropriate level for tests and full reporting of outcomes |
| ☒ | ☐ | Estimates of effect sizes (e.g. Cohen's *d*, Pearson's *r*), indicating how they were calculated |

*Our web collection on statistics for biologists contains articles on many of the points above.*

## Software and code

Policy information about availability of computer code

| | |
|---|---|
| Data collection | HDX-MS data were collected using a Synapt G2Si (Waters). Proteomics data were collected using a Lumos Tribrid Orbitrap. CryoEM data were collected using EPU v3.8.1.7603 (ThermoFisher Scientific) on a Titan Krios equipped with a Falcon 4i detector and Selectris energy filter. |
| Data analysis | PLGS 3.0 (Waters), Dynamx 3.0 (Waters) and HXexpress were used for analysis of HDX-MS data. Proteomics data were analysed using Maxquant 2.5 and Perseus 1.6.2.3. Western blots were analysed in ImageJ. Other data were analysed using Graphpad Prism 9. CryoEM initial particle picking, reconstruction, classification and refinement in Cryosparc v4.4.1. Further refinement in Relion 5. Model building using WinCoot v0.9.8.93, Isolde v1.7.1 and ChimeraX v1.7.1. Model refinement in Phenix v1.21. |

For manuscripts utilizing custom algorithms or software that are central to the research but not yet described in published literature, software must be made available to editors and reviewers. We strongly encourage code deposition in a community repository (e.g. GitHub). See the Nature Portfolio guidelines for submitting code & software for further information.

## Data

Policy information about availability of data

All manuscripts must include a data availability statement. This statement should provide the following information, where applicable:

- Accession codes, unique identifiers, or web links for publicly available datasets
- A description of any restrictions on data availability
- For clinical datasets or third party data, please ensure that the statement adheres to our policy

All mass spectrometry data have been deposited to the ProteomeXchange Consortium via the PRIDE partner repository with the dataset identifiers PXD055069; PXD055060; PXD055251; PXD055280; PXD055476. Proteomic analysis used the Uniprot E. coli reference proteome (UP000000625).
The electron density reconstructions and structure coordinates were deposited to the Electron Microscopy Database (EMDB) and to the PDB under the following accession codes: EMD-51611 and PDB: 9GUL. Structural models used to initiate model building were accessed from the PDB: PDB 6QZP and PDB 6R5Q.

## Research involving human participants, their data, or biological material

Policy information about studies with human participants or human data. See also policy information about sex, gender (identity/presentation), and sexual orientation and race, ethnicity and racism.

| | |
|---|---|
| Reporting on sex and gender | Not applicable |
| Reporting on race, ethnicity, or other socially relevant groupings | Not applicable |
| Population characteristics | Not applicable |
| Recruitment | Not applicable |
| Ethics oversight | Not applicable |

Note that full information on the approval of the study protocol must also be provided in the manuscript.

# Field-specific reporting

Please select the one below that is the best fit for your research. If you are not sure, read the appropriate sections before making your selection.

☒ Life sciences    ☐ Behavioural & social sciences    ☐ Ecological, evolutionary & environmental sciences

For a reference copy of the document with all sections, see nature.com/documents/nr-reporting-summary-flat.pdf

# Life sciences study design

All studies must disclose on these points even when the disclosure is negative.

| | |
|---|---|
| Sample size | No statistical methods were used to calculate sample sizes. Biochemical experiments were repeated at least three times, and in some cases using independent protein purifications. 3-5 replicates of HDX-MS data were collected to account for technical variability in pipetting, considering prior evidence in the literature supporting the high technical reproducibility of these experiments, and consistent with community guidelines for HDX MS (Masson, G. R. et al. Recommendations for performing, interpreting and reporting hydrogen deuterium exchange mass spectrometry (HDX-MS) experiments. Nat Methods 16, 595–602 (2019)). In some cases, replicate data were also acquired for independent purifications of the protein complexes to assess biological variability. For other experiments (PK assays, Cys labelling), the high reproducibility and large effect sizes indicate that the number of replicates was sufficient. For structural studies, 788,554 particles were picked from 29,080 movies. No sample size calculation was performed for the cryoEM data; the sample size is sufficient for the resolution of the map reported here. |
| Data exclusions | For structural studies, electron micrograph movies with substantial drift and crystalline ice were excluded. This is a pre-established standard in the cryoEM community. Picked particles were excluded based on 2D and 3D classification. Excluded particles were those belonging to classes that lacked high-resolution structural features; this is a pre-established standard in the cryoEM community. No other data were excluded. |
| Replication | All experiments except for structure determination were confirmed using replicate measurements. CryoEM single particle analysis involves averaging many independent particles. |
| Randomization | No randomization was performed, as samples were not grouped. |
| Blinding | No blinding was performed, as samples were not grouped. Moreover, MS data were analysed using standard pipelines which minimise the possibility of subjective interpretation. |

# Reporting for specific materials, systems and methods

We require information from authors about some types of materials, experimental systems and methods used in many studies. Here, indicate whether each material, system or method listed is relevant to your study. If you are not sure if a list item applies to your research, read the appropriate section before selecting a response.

## Materials & experimental systems

| n/a | Involved in the study |
|---|---|
| ☐ | ☒ Antibodies |
| ☐ | ☒ Eukaryotic cell lines |
| ☒ | ☐ Palaeontology and archaeology |
| ☒ | ☐ Animals and other organisms |
| ☒ | ☐ Clinical data |
| ☒ | ☐ Dual use research of concern |
| ☒ | ☐ Plants |

## Methods

| n/a | Involved in the study |
|---|---|
| ☒ | ☐ ChIP-seq |
| ☒ | ☐ Flow cytometry |
| ☒ | ☐ MRI-based neuroimaging |

## Antibodies

| | |
|---|---|
| Antibodies used | Anti-FLuc (Abcam, ab185923, lot# GR3317915-2). Diluted 1:1000 |
| Validation | DOI: 10.1093/nar/gkab1241 <br><br> Also validated in the current manuscript using purified FLuc. |

## Eukaryotic cell lines

Policy information about cell lines and Sex and Gender in Research

| | |
|---|---|
| Cell line source(s) | Expi293F suspension cells (ThermoFisher) |
| Authentication | The cell line was not authenticated |
| Mycoplasma contamination | The cell line tested negative for mycoplasma |
| Commonly misidentified lines (See ICLAC register) | *Name any commonly misidentified cell lines used in the study and provide a rationale for their use.* |

## Plants

| | |
|---|---|
| Seed stocks | *Report on the source of all seed stocks or other plant material used. If applicable, state the seed stock centre and catalogue number. If plant specimens were collected from the field, describe the collection location, date and sampling procedures.* |
| Novel plant genotypes | *Describe the methods by which all novel plant genotypes were produced. This includes those generated by transgenic approaches, gene editing, chemical/radiation-based mutagenesis and hybridization. For transgenic lines, describe the transformation method, the number of independent lines analyzed and the generation upon which experiments were performed. For gene-edited lines, describe the editor used, the endogenous sequence targeted for editing, the targeting guide RNA sequence (if applicable) and how the editor was applied.* |
| Authentication | *Describe any authentication procedures for each seed stock used or novel genotype generated. Describe any experiments used to assess the effect of a mutation and, where applicable, how potential secondary effects (e.g. second site T-DNA insertions, mosiacism, off-target gene editing) were examined.* |

