## [Peer Review File · Nature Structural & Molecular Biology]

The human ribosome modulates multidomain protein biogenesis by delaying cotranslational domain docking

Corresponding Author: Dr David Balchin

Version 0:

Decision Letter:

18th Oct 2024

Dear Dr. Balchin,

Thank you again for submitting your manuscript "The human ribosome modulates multidomain protein biogenesis by delaying cotranslational domain docking". I apologize for the delay in this decision. As you know, I have been assigned as your handling editor in Dr. Osman's absence. Please do not hesitate to get in touch with me if you have any questions.

I'm writing to let you know that we have decided to send your manuscript for peer review.

I am re-opening the manuscript submission link for you to resubmit your manuscript with all the associated files needed for the peer review process directly to our system, at your convenience. Please see below for details regarding the required materials. Please follow the link at the bottom of this email to upload the documents.

We want to ensure that the methods and statistics reporting in our papers are of the highest quality. To that end, we ask authors to fill out a Reporting Summary that collects information on experimental design and reagents, as well as an editorial Policy Checklist, which confirms compliance with our editorial policies, including the declaration of Competing Interests. If your paper includes ChIP-seq, flow cytometry or MRI data, we ask you take special care to complete those sections of the Reporting Summary as this data will aid greatly in the review of your manuscript.

These documents can be found by following the links below:

Reporting Summary:

Editorial Policy Checklist: <https://www.nature.com/documents/nr-editorial-policy-checklist.pdf>

Please be aware of our guidelines on digital image standards.

*****IMPORTANT*****

In order for us to proceed with the peer review process of your manuscript, we require you to provide accession numbers and reviewer tokens to access sequencing data sets. Please add this information to your manuscript file.

Please note we require official wwPDB validation reports for newly described atomic structures, as noted in the policy checklist. We also request that authors provide cryo-EM maps, half-maps and models, to help the reviewers in assessing the work. We recommend the use of figshare integration into our systems, which allows for provision of anonymous access links for the referees (<https://www.springernature.com/gp/authors/research-data/figshare-integration>).

Alternatively, please upload .zip folders directly with the submission. To ensure the ease of reviewer access to the data, please specify in the Data Availability section, where the files can be found (provide a figshare link or direct the reader to the manuscript files).

Please note that we will be unable to proceed with the peer review process in the absence of these data and documents. Please let me know if anything is unclear about the requirements, or you have any concerns sharing your data.

Additionally, I would like to kindly request that you provide the code used to analyse the data to the reviewers, if used. In order for the reviewers to evaluate the work adequately they must be able to test the software/review the code themselves. If you have not yet provided the software, we therefore request that you provide a single compressed zip file containing the software with a readme.txt file or other user manual containing complete instructions for installing and running the software. If appropriate, please also provide example data and expected output. Sufficient material should be provided for referees to directly test the performance of the software/algorithm. If the software and materials are small enough to fit in a single compressed zip file less than 6MB in size, you may email this file directly to me. If the zip file is between 6 MB and 200 MB you may upload it to our file transfer site. If necessary, a second zip file up to 200 MB in size can be used to supply the example data. Please let me know if you need to use this option and I'll send you further details. Alternatively, you can also upload the code to GitHub and provide us with the link.

Please also fill out and return to me the code and software submission checklist that will be made available to editors and reviewers during manuscript assessment. Please note that this form is a dynamic 'smart pdf' and must therefore be downloaded and completed in Adobe Reader, instead of opening it in a web browser.

<https://www.nature.com/documents/nr-software-policy.pdf>

Please use the link below to submit the files. **Please also remember to move forward all other files associated with this version of the paper.**

Link Redacted

Sincerely,
Kat

Katarzyna Ciazynska, PhD
(she/her)
Senior Editor
Nature Structural & Molecular Biology
<https://orcid.org/0000-0002-9899-2428>

Version 1:

Decision Letter:

28th Feb 2025

Dear Dr. Balchin,

Thank you for submitting your manuscript "The human ribosome modulates multidomain protein biogenesis by delaying cotranslational domain docking". I sincerely apologize for the extreme delay in processing your manuscript, which resulted from difficulties in obtaining referees' reports. Nevertheless, we now have the comments from the 3 reviewers who have evaluated your manuscript are below. Unfortunately, after carefully considering their comments, we cannot offer to publish your manuscript in Nature Structural & Molecular Biology.

You will see that while the referees find the work of potentially interesting, they raise concerns about the strength of the novel conclusions that can be drawn at this stage. You will see that specifically reviewer #1 has concerns regarding the data on the strength of the cryo-EM data, as well as the choice of the construct itself, and the support it offers to the conclusions.

However, if further experimentation, analysis, and revisions allow you to address the referees concerns in full, we would be prepared to consider an appeal of our decision, on the condition that no related work is published in the interim or has been accepted in our journal. Please contact me to discuss an appeal and potential revision. I would be happy to schedule a call to discuss this. Please note that, until we have the opportunity to read the revised manuscript in its entirety, we cannot promise that it will be sent back for peer review.

I am sorry we could not be more positive on this occasion. I hope that you find the referees' comments useful in deciding how

best to proceed.

Sincerely,
Kat

Katarzyna Ciazynska, PhD
(she/her)
Senior Editor
Nature Structural & Molecular Biology
<https://orcid.org/0000-0002-9899-2428>

Reviewers' Comments:

Reviewer #1 (Remarks to the Author):

In this manuscript Pellowe et al. explore the co-translational folding of the multidomain protein firefly luciferase (FLuc) on both human and bacterial ribosomes. They harnessed a range of biochemical and structural biology assays, particularly mass spectrometry, to characterise the folding process and the intermediate states of FLuc on the ribosome at different stages during translation.

Overall, the manuscript is well-written, data are robust and of high quality, coherently supporting their conclusions. This is an exciting study, particularly showcasing how mass spectrometry can be utilised to reveal the dynamic process of a multidomain protein folding on the human ribosome – a process that has not been well-characterised before. Multiple orthogonal analyses, including HDX-MS, limited proteolysis and the F5M accessibility assay, consistently demonstrate delayed folding of the Ns domain and the destabilisation of the domain interfaces by the C-terminal residues on the human ribosome, whereas the data from the 70S ribosome reveal noticeably different trends. Given the interest of the field in multidomain protein folding, co-translational intermediates and related concepts such as delayed folding, domain interfaces, and the entropic contributions of the ribosome, I believe this manuscript will likely to attract well-deserved attention from the field. In principle, I would recommend publishing this manuscript in NSMB, provided my questions and suggestions below are adequately addressed.

The main weakness of the paper appears to lie in the cryo-EM data and its interpretation in the context of the main conclusions of the manuscript. The authors used Ns-RNC for their cryoEM structure to explain the structural basis of the stalling mechanism of the modified XBP1. However, this is neither particularly exciting – given its similarity to previously reported data on the original sequence – nor useful in supporting their main conclusions. Understandably the NC is dynamic for this short construct, as most of the Ns-RNC is unfolded, as evidenced by the MS data. In the manuscript why this specific length, rather than longer chains (such as T-RNC, or even N-RNC to investigate the mechanism and the NC structure for the folded Ns domain), was chosen is not explained.

The structural analysis of the visible NC (8 residues) is not convincing. This is partly due to the poor resolution caused by NC dynamics (Fig. S4), and partly because of the NC in this short construct would not fold into a stable structure. Because of this the absence of clear density (line169-170) in the vestibule and outside the tunnel exit can easily be predicted. Furthermore, the detailed structural interpretations – such as the predicted electrostatic potential-based interaction between the NC and the ribosome surface, or the location of the NC in the groove between eL39 and rRNA – are not convincing for these reasons. Also, NCs of different lengths would exhibit different local charges at this location of the tunnel, likely resulting in distinct interactions with the ribosomal tunnel. Consequently, conclusions (such as line339-341) are not supported by their cryoEM structure. Using CryoEM structure of N-RNC or T-RNC may provide far more convincing data to support their argument.

Minor comments

- L125

Enzymatic activity of RNCs in S1b. FL-RNC appears to have greater activity than isolated FLuc. Is this biologically plausible? Could this be related to any released full-length NC in the RNC sample, as suggested by the released product in Fig1e and FigS1b?

Why is there stronger signal for the FL-FLuc on the gel after RNaseA treatment?

- Fig4a

Why is there no change at all in T-RNC, despite the structural destabilisation of T-RNC observed in Fig3? Is this due to limitations of the proteolysis experiments, or could it be a result of the lack of digestion sites in those regions (e.g. Ns-N interface and b-roll)?

What are the bands in the 30-50 kDa range for 1/2N-RNC? Could these represent dimers of Ns as shown in Fig4b-e?

- line214-215

Does a shallow transition always indicate low cooperativity? Do they refer to global or local cooperativity? Is the data fit to a simple two-state unfolding kinetic model? Additional clarification and explanation would be helpful.

- line 237

Where in the figure is the partially folded population evident?

- line 249

Do the data demonstrate an alteration in the conformational ensemble? Or is this simply a change in population between two distinct states (folded and unfolded)? Is there any evidence supporting the presence of multiple conformational states revealed by the data?

- FigS9a

It would be helpful to show the two interface regions with dashed boxes as in main figures

- line346

Previous studies on the bacterial ribosome (e.g., reference 26) suggest C-terminus-mediated destabilisation of the nascent chain on bacterial ribosome. Can the 'insulation of entropic destabilisation' observed on the bacterial ribosome in this study generalised?

- HDX-MS data (Fig3a,h, Fig5d, Fig6a)

Are there uncertainties associated with these data? How many experimental repeats were performed?

- Fib5ab Ex1 data

Can these experiments be performed under different conditions (e.g., varying temperature or pulse labelling times) to provide more detailed kinetic information on the dynamics between the two states?

- line183

Why was 0.5Da chosen as the cutoff? The Ns:N interface of T-RNC shows a weak ΔDa . How can this be justified as meaningful evidence of destabilisation of this region?

- FigS2d

The cryoEM structure of Ns-RNC shows an interaction between the NC and eL39. Was this interaction observed in the crosslinking data?

- line210-219

It is not clear how the role of the ribosome in Ns folding is being explained using data from isolated proteins. The CD and HDX-MS data are derived from the dimer. Are these data intended to show that the protein cannot fold independently without the ribosome and self-assemble? Is it clear that the isolated dimer is formed by the same interface as in the ribosome-bound state?

Reviewer #3 (Remarks to the Author):

While many proteins fold efficiently when refolded from a denatured state, many at the same time are unable to do so. Protein folding *in vivo* starts cotranslationally and hence is widely believed to facilitate efficient folding of especially those proteins that are unable to refold efficiently *in vitro*. Many recent studies shed light on the general principles governing cotranslational protein folding, however, many details of this process remain poorly understood. New techniques for both measuring folding kinetics and detecting the conformations of partially folded intermediates on the ribosome emerged recently thus allowing to get important information on cotranslational folding trajectories of several single and multidomain proteins. In this manuscript, Pellowe and co-authors used cryo-electron microscopy and hydrogen/deuterium exchange-mass spectrometry to characterize the structure and dynamics of partially-synthesized intermediates of a model multidomain protein, Firefly Luciferase (FLuc), previously shown to fold cotranslationally. The authors used HEK293 cells to characterize folding intermediates of FLuc on the human ribosome and *E. coli* cells to further understand whether cotranslational folding intermediates differ between human and bacterial ribosomes. Homogeneously-stalled Ribosome-nascent chain complexes (RNCs) were obtained using an arrest-enhanced variant of XBP1u, or a bacterial arrest-enhanced SecM sequence, respectively. The authors found that FLuc nascent chain subdomains fold progressively during synthesis on the human ribosome, templated by interactions across domain interfaces and that the conformational ensemble of the nascent chain is tuned by its unstructured C-terminal segments, which keep interfaces between folded domains in dynamic equilibrium until translation termination. Interestingly, the authors found that in case of bacterial ribosomes, domain interfaces appeared to form early and remained stable during synthesis. The authors concluded that delayed domain docking may avoid interdomain misfolding to promote the maturation of multidomain proteins in eukaryotes. This is an elegant and comprehensive study providing important insights into the mechanism of protein folding in the cell.

I have only a few minor comments:

1. The authors write (page 7) that "Digestion of 1/2N-RNC resulted in weak accumulation of the previously-described ~22 kDa intermediate corresponding to Ns..." and then state that "Ns is not an independent folding unit, but is held on the ribosome in a folding-competent state until a larger part of the Rossmann fold is available". Please note that Frydman et al (Nature, 1994) in their seminal publication not only observed the formation of this N-terminal intermediate, but also described the time-dependent nature of its formation during protein synthesis and hypothesized that during the course of translation it further engages in interactions with the rest of the synthesized protein. This earlier observation needs to be mentioned and discussed in a bit more detail.

2. The authors revealed that human and bacterial ribosomes differentially affect the conformational ensemble of FLuc NCs. They have concluded that “different ribosomes can shape the conformational ensemble of a model NC.” I would be a bit more cautious with this conclusion and consider alternative explanations too. While different ribosome structures may certainly provide different folding environment for the emerging nascent chains (both inside and outside the ribosome tunnel), another important factor to consider (besides, also the different cellular environment discussed by the authors) is the local and global synthesis rates that are known to modulate protein folding in the cell (see e.g. Buhr et al, Mol. Cell, 2016, etc). It is my understanding that the same Fluc sequence was used to obtain RNCs in HEK293 cells and E. coli and hence due to different repertoire of the tRNAs (and different codon usage bias) in these systems, the differential translation kinetics of Fluc may have resulted in different (structural and functional) intermediates. Please note that Liu and colleagues previously showed that specific luciferase activity can be affected by altered dynamics of its synthesis (see Yu et al, Mol. Cell, 2016) and Kolb et al (JBC 2000) in an earlier study also demonstrated that specific luciferase activity may differ in prokaryotic and eukaryotic translation systems. Interestingly, an about 10-20% higher specific luciferase activity was observed in E. coli S30 system compared to wheat germ extract system (Kolb et al, JBC, 2000) back then. It was also shown that refolding of luciferase in E. coli and wheat germ cytosol proceeds with different efficiency (Kolb et al, JBC, 2000), suggesting that besides the difference in translation kinetics, differences in the cellular environment (mentioned by the authors) are important contributors to the process of luciferase folding. In case, I am reading Figs. S1B and S8D correctly, FL-RNCs have about 20% higher specific activity compared to FL-bRNCs. This difference additionally implies that there exists a different spatial arrangement of the ribosome bound chains between FL-RNC and FL-bRNCs in the two systems used. As shown by the authors, this conformational difference persists in the Fluc chains fully extruded out of the ribosome (Fig. 6). However, the lower specific activity in case of FL-bRNC suggests that the differential engagement of the Fluc C-terminus (required for protein activity, see Sala-Newby et al., Biochem. Biophys. Res. Commun., 1990) with the rest of the protein (C-domain was substantially deprotected in FL-bRNC compared to FL-RNC as revealed by the authors) may account for this difference. These considerations and issues related to Fluc activity need to be also discussed by the authors.

3. It is also my understanding that RNCs were purified using either the 3xFLAG-tag (in case of HEK293 cells) or the muGFP(green)-tag (in case of E. coli cells; as described previously by the authors (Wales et al, 2024)). Could this difference in the N-terminal tagging affect the initial events of luciferase folding in the different systems used? The authors found that the FLAG-tag does not measurably affect the conformation of FL-RNCs (Fig. S5), however they didn't provide any evidence regarding the muGFP(green)-tag. This issue also needs to be discussed. Obviously, the muGFP(green)-tag has been cleaved off after RNCs purification. However, before it has been cleaved off, it may have affected the folding of a (comparable in size) Ns domain, which is not an independent folding unit and due its dynamic nature may have been engaged in interactions with muGFP(green)-tag.

Minor:

Figure 8. There is no D panel, D in the text should be replaced with C.

Reviewer #4 (Remarks to the Author):

The manuscript by Pellowe et al investigates how human ribosomes influence the folding of multidomain proteins during synthesis. Using mainly cryo-electron microscopy and HDX-MS experiments on a model 550 res two-domain protein FLuc, the authors effectively illustrate how delayed domain docking helps prevent misfolding, highlighting a key difference between eukaryotic and bacterial ribosomes. The experimental work appears thorough, and speaking mainly for the HDX-MS experiments, I believe these are performed well and the data is generally adequately analyzed and interpreted. Overall, the paper provides new interesting insights into folding of multi-domain proteins and ribosome-specific folding mechanisms in eukaryotes. I find the work interesting and the results from the combination of the many techniques used (including cryo-EM, XL-MS, limited proteolysis, HDX-MS etc) convincing. I am positive towards publication provided the below comments are addressed.

Comments:

Line 183 – the authors need to substantiate and support the assertion that peptides are natively folded only when they differed in D uptake from full-length FL by less than 0.5 Da. In general it is not clear to me how the authors approached error in their HDX-MS experiments.

Line – 235 The writing on the EX1 kinetics is a bit confusing. Surely, for all proteins folded and unfolded conformations are populated to some extent. What defines EX1 kinetics is that the conversion between these states is slow on the HDX timescale. Please rewrite this part. Also, can the authors say something about what are the likely parts in each peptide that have a slow kcl? Does this make sense structurally? More discussion would add to this part.

Line 722 – I am assuming the LC used is an UPLC?

Line 750 – since the authors use HDMSe to identify peptides they should make sure that each peptide is identified in several replicate HDMSe experiments i.e 2 out of 3 or 3 out of 4 – to add further confidence to identification. MS is notorious for misidentifications if such a replicate validation threshold is not used.

** For Springer Nature Limited general information and news for authors, see <http://npg.nature.com/authors>.

Version 2:

Decision Letter:

17th Apr 2025

Dear Dr. Balchin,

Thank you for your letter concerning your manuscript "The human ribosome modulates multidomain protein biogenesis by delaying cotranslational domain docking". We have now had a chance to discuss the points you raised in detail, and we have decided to send your paper out to review. You will find a link at the bottom of this email to resubmit the manuscript with any data and materials required for re-review.

Before we reach out to reviewers, please provide the below. We also ask that you provide source data for the graphs and images in the manuscript. Please see the details below.

We want to ensure that the methods and statistics reporting in our papers are of the highest quality. To that end, we ask authors to fill out a Reporting Summary that collects information on experimental design and reagents, as well as an editorial Policy Checklist, which confirms compliance with our editorial policies, including the declaration of Competing Interests.

These documents can be found by following the links below:

Reporting Summary:

Editorial Policy Checklist:

<https://www.nature.com/documents/nr-editorial-policy-checklist.pdf>

Please complete the relevant forms and return them within 48 hours. Please note that these forms are dynamic 'smart pdfs' and must, therefore, be downloaded and completed in Adobe Reader. We will then flatten them for ease of use by the reviewers. If you would like to reference the guidance text as you complete the template, please access these flattened versions at <http://www.nature.com/authors/policies/availability.html>.

Note that you are not required to revise your paper to include the information provided in the reporting summary. However, all points on the policy checklist must be addressed; please send me a new version of the manuscript with your completed checklist if needed.

EXTENDED DATA FIGURES

Please note that all key data shown in the main figures as cropped gels or blots should be presented in uncropped form, with

molecular weight markers. These data can be aggregated into a single supplementary figure item. While these data can be displayed in a relatively informal style, they must refer back to the relevant figures. These data should be submitted with the final revision, as source data, prior to acceptance, but you may want to start putting it together at this point.

Data availability: this journal strongly supports public availability of data. All data used in accepted papers should be available via a public data repository, or alternatively, as Supplementary Information. If data can only be shared on request, please explain why in your Data Availability Statement, and also in the correspondence with your editor. Please note that for some data types, deposition in a public repository is mandatory - more information on our data deposition policies and available repositories can be found below:

<https://www.nature.com/nature-research/editorial-policies/reporting-standards#availability-of-data>

Once we receive these documents and review them to ensure that all requested information is provided, we will proceed to send your paper for review. If you have questions or anticipate delays, please let me know as soon as possible.

Please use the link below to be taken directly to the site and submit your manuscript:

Link Redacted

Sincerely,
Kat

Katarzyna Ciazynska, PhD
(she/her)
Senior Editor
Nature Structural & Molecular Biology
<https://orcid.org/0000-0002-9899-2428>

Version 3:

Decision Letter:

6th Jun 2025

Dear Dr. Balchin,

Thank you again for submitting your manuscript "The human ribosome modulates multidomain protein biogenesis by delaying cotranslational domain docking". I apologize for the delay in responding, which resulted from the difficulty in obtaining suitable referee reports. As reviewer #4 was unavailable to assess the revision, we recruited reviewer #5 to comment on technical aspects. Please see below for the comments from the 4 reviewers who evaluated your paper. In light of those reports, we remain interested in your study and would like to see your response to the comments of the referees, in the form of a revised manuscript.

You will see that reviewer #1 asks for revisions to representation of HDX MS errors, and reviewer 2 has additional comments. Please address these points in the text. Please be sure to address/respond to all concerns of the referees in full in a point-by-point response and highlight all changes in the revised manuscript text file. If you have comments that are intended for editors only, please include those in a separate cover letter.

We are committed to providing a fair and constructive peer-review process. Do not hesitate to contact us if there are specific

requests from the reviewers that you believe are technically impossible or unlikely to yield a meaningful outcome.

We expect to see your revised manuscript within 6 weeks. If you cannot send it within this time, please contact us to discuss an extension; we would still consider your revision, provided that no similar work has been accepted for publication at NSMB or published elsewhere.

Reporting Summary:

-- that unprocessed scans are clearly labelled and match the gels and western blots presented in figures. Please note that all key data shown in the main figures as cropped gels or blots should be presented in uncropped form, with molecular weight markers. While these data can be displayed in a relatively informal style, they must refer back to the relevant figures. These data should be submitted as source data with the last revision, prior to acceptance.

-- that control panels for gels and western blots are appropriately described as loading on sample processing controls

-- all images in the paper are checked for duplication of panels and for splicing of gel lanes.

-- For any revision that includes light microscopy data, we ask our authors to please include a completed light microscopy reporting table [https://www.nature.com/documents/Light_microscopy_reporting_table.xlsx] to ensure the methods are described thoroughly. The table will be available to reviewers and ultimately published should the manuscript be accepted at the journal.

EXTENDED DATA FIGURES

Data availability: this journal strongly supports public availability of data. All data used in accepted papers should be available via a public data repository, or alternatively, as Supplementary Information. If data can only be shared on request, please explain why in your Data Availability Statement, and also in the correspondence with your editor. Please note that for some data types, deposition in a public repository is mandatory - more information on our data deposition policies and available repositories can be found below:

<https://www.nature.com/nature-research/editorial-policies/reporting-standards#availability-of-data>

Link Redacted

Sincerely,

Katarzyna Ciazynska, PhD
(she/her)
Senior Editor
Nature Structural & Molecular Biology
<https://orcid.org/0000-0002-9899-2428>

Reviewers' Comments:

Reviewer #1 (Remarks to the Author):

It is clear the authors have improved the text in various areas. However, there are still a few points that remain unconvincing to me.

First, regarding the cryoEM data: the Q-score based assessment for model fitting looks good and justifies the structural modelling. However, I still cannot see that these structures support their main conclusion well. The two reasons they offer – demonstrating the integrity of one of the RNCs and insights into the stalling mechanism – do not seem particularly interesting or coherent in supporting the central findings of the manuscript. As they acknowledge, density for the nascent chain (NC) outside exit is challenging to capture, even when the NC is well folded. Contrary to their response, however, NC density outside the exit has been reported in several studies (e.g., Zhang et al. 2015 Elife; Tian et al. 2018 PNAS), though these were based on bacterial ribosomes. Even if the T-RNC or N-RNC lack NC density outside the exit as they suggest, the density inside the tunnel would be more readily visualised and more informative than what is shown for the Ns-RNC in Figure 2 – especially since the manuscript's conclusions primarily rely on HDX data of 1/2N-RNC and T-RNC.

It is useful to see statistical analysis added to some figures. However, the key HDX figures (Fig 3a and 5c) would better include uncertainties for each data point. These data underpin the main conclusions, and I do not think displaying uncertainties will make the figures overly complex. Rather, including error bars is essential for allowing readers to interpret and assess the data in a more rigorous and transparent manner.

Other responses appear reasonable.

Reviewer #2 (Remarks to the Author):

I still think the paper should be accepted eventually and I appreciate the authors comments to my first round of review. Here are some additional comments based on a close second read and my response to some of their responses in a few cases where I really do think a change is required/merited.

1. "FLuc refolds extremely slowly ($t_{1/2}$ ~75 min)." Slowly and inefficiently, I would add. (Slowly by itself implies that it eventually gets to 100% just slowly, but really even after 75 min, the corresponding author showed in the cited paper the final recovered activity is 10-50%).
2. Extended Data Fig 1B: Any comment why FL-RNC has a 3x activation relative to isolated FL-FLuc?
3. Line 136 and 139: Extended Data Fig SA, missing a number

4. I would appreciate residue numbering in Fig 3A
5. Average back exchange of 50%. This seems higher than normal for HDX-MS. Could authors comment why? Is this due to a technical limitations associated with these complicated samples?
6. In general, I'm somewhat confused about the length of 1/2N relative to Ns. Based on Fig 1B, this is only slightly longer than Ns, But then in Fig 3, it would seem 1/2N is much longer than Ns. I guess this is important for understanding how much more NC needs to be synthesized for Ns to fold/dock
7. Another place where this occurs: "that Ns is near-natively folded at this chain length." Please specify the chain length
8. In their response to my first round of review, I appreciated the authors' point that co-translational folding is perhaps most interesting and relevant in the context of proteins with complicated domain topologies of which FLuc is a good model, but this discussion did not seem to make its way into the text. I think the authors should be more explicit, perhaps in their discussion, that the findings they report here (such as Ns folding only when a larger part of N is synthesized, and Ns:N destabilization upon synthesis of C-domain) are possibly specific to proteins like luciferase with nested domain architecture. It may be that vectorial domain folding would be more common for multi-domain proteins that are of the more simple tail-to-head nature. We don't need to argue about which one is more common. The point is just that some multidomain proteins are tail-to-head and some are discontinuous. FLuc is archetypal of the latter. And hence the results of this important paper speak to that subclass of multidomain protein.
9. In the author's response to my first round of review, they wrote "The N domain contains 3 folds, which are coloured differently." I have never heard of this notion of a domain with multiple folds. A domain is not an arbitrary structural unit but an evolutionarily conserved unit that we can match with an HMM (e.g., a beta-roll, a Rossmann). Hence, FLuc is – by that definition – a 4-domain protein in which the first 3 exhibit high levels of discontinuity. Inspecting the uniprot entry's InterPro shows it as having four domains. Hence, Ns is not really a domain, but a structural unit that contains two halves of two different domains. I think that this description is more correct. And it really does explain why Ns would not be expected to be very well folded by itself until a larger chunk of the protein is available (namely 1/2N). I would ask the authors in Figure 1 to make this clearer by coloring FLuc in 3-D in two different ways, in one way that is based on the bioinformatic domains (as it is shown in the linear representation above it) and in a second way in which Ns and N (which are structural units rather than strict domains) are colored. I would also ask that Ns be explicitly referred to as a composite of two half-domains, to provide more context to why it does not fold as an independent unit and requires elements only present in 1/2N to be stabilized. This does not take away from the authors interesting discoveries that the entirety of N needs to be present for Ns to dock, and that Ns then undocks at T-RNC to redock again at FL-RNC. But I will hold my ground that it is not surprising that Ns is unstructured at Ns-RNC, and I think this simple point should be made explicitly by the authors so that the truly interesting results can be more greatly appreciated by their future readers.
10. Related to the above point, it seems like the "N-domain" sometimes is used to refer to the parts of the first 3 domains excluding Ns (as in Figure 1C), but sometimes it is used to refer to the first 3 domains including Ns (as in Figure 1A). Could a consistent nomenclature be used? Perhaps what is sometimes called N-domain could be called N2, so that Ns + N2 = N? The nomenclature just seems awkward to say that at N-RNC, there is both Ns and N.
11. Figure 8 does not have a panel D (typo in the caption)

Reviewer #3 (Remarks to the Author):

The manuscript by Pellowe and co-authors has been revised. Additional information has been added, which was missing in the original version of the manuscript. In sum, I feel that the authors have responded to the majority of the previous concerns and the manuscript has been substantially improved. I have no additional comments. I believe the manuscript provides sufficiently novel and interesting data to warrant its publication in NSMB.

Reviewer #5 (Remarks to the Author):

I have primarily reviewed the HDX-MS data in the submitted manuscript, and the authors have done an excellent job of responding to reviewer's comments. They now have included error bars on the graphs wherever possible and have included standard deviations for all the data in supplementary tables. Given the scarcity of sample, it is acceptable that the authors only present 3 technical replicates and do not have a biological replicate. The differences they observe are well outside of the standard deviation. It is also acceptable that they suggest using a cut-off of 0.5D even though the standard deviation is smaller (0.11 D). The differences are all much larger than these errors and the conclusions based on the HDX-MS data are robust. Overall the combination of all the different experiments including structure determination, limited proteolysis, HDX-MS and the use of various constructs is comprehensive. The results are super interesting for understanding co-translational folding on human ribosomes and publication is strongly recommended.

Version 4:

Decision Letter:

Our ref: NSMB-A49828D

11th Jul 2025

Dear Dr. Balchin,

Thank you for submitting your revised manuscript "The human ribosome modulates multidomain protein biogenesis by delaying cotranslational domain docking" (NSMB-A49828D). It has now been seen by the original referees and their comments are below. The reviewers find that the paper has improved in revision, and therefore we'll be happy in principle to publish it in Nature Structural & Molecular Biology, pending minor revisions to satisfy the referees' final requests and to comply with our editorial and formatting guidelines.

We are now performing detailed checks on your paper and will send you a checklist detailing our editorial and formatting requirements in about 2-3 weeks. Please do not upload the final materials and make any revisions until you receive this additional information from us.

Sincerely,

Katarzyna Ciazynska, PhD
(she/her)
Senior Editor
Nature Structural & Molecular Biology
<https://orcid.org/0000-0002-9899-2428>

Reviewer #1 (Remarks to the Author):

The authors have made suggested changes to the manuscript, which I believe has improved it significantly. I would recommend publishing this manuscript in NSMB.

Version 5:

Decision Letter:

14th Aug 2025

Dear Dr. Balchin,

We are now happy to accept your revised paper "The human ribosome modulates multidomain protein biogenesis by delaying cotranslational domain docking" for publication as an Article in Nature Structural & Molecular Biology.

Your paper will be published online soon after we receive proof corrections and will appear in print in the next available issue. You can find out your date of online publication by contacting the production team shortly after sending your proof corrections.

Authors may need to take specific actions to achieve compliance with funder and institutional open access mandates. If your research is supported by a funder that requires immediate open access (e.g. according to [Plan S principles](https://www.springernature.com/gp/open-science/plan-s-compliance) or the [NIH public access policy](https://www.springernature.com/gp/open-science/us-federal-agency-compliance)) then you should select the gold OA route, and we will direct you to the compliant route where possible. Because authors warrant under our subscription licensing terms that they haven't committed to licensing any version of their article under a licence inconsistent with the terms of our agreement – including the applicable embargo period – publication under the subscription model isn't suitable for authors whose funders require no embargo.

Sincerely,

Katarzyna Ciazynska, PhD
(she/her)
Senior Editor
Nature Structural & Molecular Biology
<https://orcid.org/0000-0002-9899-2428>

Reviewers' Comments:

We thank the reviewers for their thoughtful and constructive comments. Our point-by-point response is below.

Text changes are highlighted in green in the revised manuscript, and reproduced below.

Reviewer #1 (Remarks to the Author):

In this manuscript Pellowe et al. explore the co-translational folding of the multidomain protein firefly luciferase (FLuc) on both human and bacterial ribosomes. They harnessed a range of biochemical and structural biology assays, particularly mass spectrometry, to characterise the folding process and the intermediate states of FLuc on the ribosome at different stages during translation.

Overall, the manuscript is well-written, data are robust and of high quality, coherently supporting their conclusions. This is an exciting study, particularly showcasing how mass spectrometry can be utilised to reveal the dynamic process of a multidomain protein folding on the human ribosome – a process that has not been well-characterised before. Multiple orthogonal analyses, including HDX-MS, limited proteolysis and the F5M accessibility assay, consistently demonstrate delayed folding of the Ns domain and the destabilisation of the domain interfaces by the C-terminal residues on the human ribosome, whereas the data from the 70S ribosome reveal noticeably different trends. Given the interest of the field in multidomain protein folding, co-translational intermediates and related concepts such as delayed folding, domain interfaces, and the entropic contributions of the ribosome, I believe this manuscript will likely to attract well-deserved attention from the field. In principle, I would recommend publishing this manuscript in NSMB, provided my questions and suggestions below are adequately addressed.

We appreciate the positive overall assessment of our manuscript, highlighting the consistent results from orthogonal approaches and general interest to the field.

The main weakness of the paper appears to lie in the cryo-EM data and its interpretation in the context of the main conclusions of the manuscript. The authors used Ns-RNC for their cryoEM structure to explain the structural basis of the stalling mechanism of the modified XBP1. However, this is neither particularly exciting – given its similarity to previously reported data on the original sequence – nor useful in supporting their main conclusions. Understandably the NC is dynamic for this short construct, as most of the Ns-RNC is unfolded, as evidenced by the MS data. In the manuscript why this specific length, rather than longer chains (such as T-RNC, or even N-RNC to investigate the mechanism and the NC structure for the folded Ns domain), was chosen is not explained.

The reviewer's point about the usefulness of the cryo-EM structure for understanding FLuc folding is well-taken. In the responses below, we better explain the value of the structure for characterising our RNC system, and substantiate our interpretation of the density map.

Regarding the choice of construct, we focused on Ns-RNC because of a prior study showing that this subdomain folds cotranslationally (DOI: 10.1038/10754). This is now mentioned in the text:

"To further characterise the stalled RNCs, we solved the structure of Ns-RNC to a global resolution of 2.2 Å by cryoEM (Fig 2A, S3A-D and S4A). Ns-RNC was chosen because the Ns subdomain was previously shown to fold cotranslationally."

We agree that the Cryo-EM structure was not specifically informative regarding the folding pathway of FLuc, and we were careful not to claim otherwise in the text. Nonetheless, we still think that the structure is useful for two reasons:

1. The subsequent experiments reported in the manuscript all rely on human RNCs produced using the new protocol that we describe. We therefore think it is important to show structural evidence for the composition and integrity of a representative RNC. We also think that this will be useful for future work by others using the same system.
2. The mechanism of enhanced stalling by XBP1u+ is of interest, additionally because the previous structure with wild-type XBP1u was of the rabbit (not human) ribosome.

The structural analysis of the visible NC (8 residues) is not convincing. This is partly due to the poor resolution caused by NC dynamics (Fig. S4), and partly because of the NC in this short construct would not fold into a stable structure. Because of this the absence of clear density (line169-170) in the vestibule and outside the tunnel exit can easily be predicted. Furthermore, the detailed structural interpretations – such as the predicted electrostatic potential-based interaction between the NC and the ribosome surface, or the location of the NC in the groove between eL39 and rRNA – are not convincing for these reasons. Also, NCs of different lengths would exhibit different local charges at this location of the tunnel, likely resulting in distinct interactions with the ribosomal tunnel. Consequently, conclusions (such as line339-341) are not supported by their cryoEM structure.

To quantitatively assess the map-model fit for the NC in the tunnel, we calculated Q-scores in Chimera X (doi.org/10.1038/s41592-020-0731-1). Based on this we have now deleted Gly-200 from our model, the only residue with a Q-score below 0.5. In addition, we removed the side chains of residues 201 and 202 for which there was no clear side-chain density.

As shown in Figure S4G and below, the density is of sufficient quality to place side-chains for Ile-204, Lys-206 and Val-208, allowing us to unambiguously trace the backbone of the nascent chain. We do not make claims about specific FLuc residues in the manuscript, but rather the overall path of the nascent chain in the exit tunnel.

As shown in Fig. 2E, there is clear NC density from the PTC to the end of the exit tunnel, allowing the path of the NC to be traced near eL39. Moreover, the density for eL39 is of high enough quality to model side-chains, allowing us to confidently describe its surface electrostatic properties. Taken together, we think that these data support our description of the backbone conformation of the NC and its position in the exit tunnel near the positively charged surface of eL39. This interpretation is supported by HDX-MS analysis showing protection of eL39 by the NC (Fig. 7).

We acknowledge that the NC density is not of sufficient quality to discuss specific electrostatic interactions at the residue level, and we agree that interactions with the exit tunnel are likely to change as the NC elongates. For these reasons, we avoid referring to specific NC-ribosome contacts. Interestingly, our HDX-MS analysis showed that eL39 is protected at all chain lengths (Fig. 7). In the manuscript we therefore suggest that the path of the NC may not depend only on its sequence.

Using CryoEM structure of N-RNC or T-RNC may provide far more convincing data to support their argument.

We have attempted this for several systems, including stably folded NCs, and have not been able to visualise the NC outside the tunnel in any case. The difficulty likely stems from the fact that even a folded NC is dynamic relative to the ribosome, and would thus also need to be stably bound to the ribosome surface to be resolved. To our knowledge a cryo-EM structure of an NC outside the exit tunnel has not yet been reported.

Importantly, other approaches used here (HDX-MS, limited proteolysis, F5M labelling) report on the folding behaviour of the NC.

Minor comments

- L125

Enzymatic activity of RNCs in S1b. FL-RNC appears to have greater activity than isolated FLuc. Is this biologically plausible? Could this be related to any released full-length NC in the RNC sample, as suggested by the released product in Fig1e and FigS1b?

FLuc enzyme activity is coupled to domain dynamics, and it is not impossible that the environment near the ribosome surface could influence its activity. However, we cannot exclude that this difference simply arises from errors in calculating sample concentration,

particularly since the concentrations of the RNC and off-ribosome samples are determined in different ways (A_{260} vs A_{280}).

We are therefore cautious about overinterpreting the difference in activity. We have now added the following note to the legend for Fig S1:

“Note that the apparent difference in enzyme activity between FL-FLuc and FL-RNC may result from differences in concentration, which is estimated using different approaches for off-ribosome and RNC samples (see Methods)”.

Why is there stronger signal for the FL-FLuc on the gel after RNaseA treatment?

We are not sure exactly, but it is possible that the first lane simply did not transfer well during the immunoblot. The activity measurements show that the amount of FLuc is the same \pm RNase. In replicate blots (below) we found no difference in band intensity \pm RNase.

- Fig4a

Why is there no change at all in T-RNC, despite the structural destabilisation of T-RNC observed in Fig3? Is this due to limitations of the proteolysis experiments, or could it be a result of the lack of digestion sites in those regions (e.g. Ns-N interface and b-roll)?

Predicted PK digestion sites (ExPASy PeptideCutter) are well distributed throughout FLuc – coloured red on the structure below.

Two possible explanations for the apparent stability of T-RNC in these experiments are:

1. Limited accessibility of the Ns:N/ β -roll interfaces to the bulky protease enzyme. Compare to the F5M small-molecule labelling experiments, which show increased labelling of T-RNC compared to N-RNC and FL-RNC (Fig. S7D).
2. After PK treatment, the destabilising effect of the C-domain is lost. When treated with PK, we observe that the partially-synthesised C-domain of T-RNC is rapidly cleaved, leaving only the structured N-domain (Fig 4a). The apparent stability of the N-domain in isolation would be consistent with our interpretation that it is destabilised on the ribosome by the unstructured C-domain fragment.

We now mention these possibilities in the manuscript as follows:

“Destabilisation of T-RNC was not reflected by increased susceptibility of the N domain to proteolysis (Fig 4A). This could be due to limited protease accessibility to cleavage sites at FLuc domain interfaces, or result from the fact that the destabilising C-terminal sequence is cleaved immediately after adding protease.”

What are the bands in the 30-50 kDa range for 1/2N-RNC? Could these represent dimers of Ns as shown in Fig4b-e?

Since the bands increase then decrease in intensity over the time course of PK treatment, they are presumably proteolysis intermediates *en route* to the stable Ns fragment that we highlight.

We do not think the bands are covalent dimers that survive SDS-PAGE, since purified Ns did not show any additional bands (Figure S6B).

- line214-215

Does a shallow transition always indicate low cooperativity? Do they refer to global or local cooperativity? Is the data fit to a simple two-state unfolding kinetic model? Additional clarification and explanation would be helpful.

We have rephrased this sentence as follows:

“Moreover, isoNs showed a shallow transition during thermal denaturation, consistent with low global unfolding cooperativity (Fig 4C).”

The data were fit to simple sigmoidal functions in order to extract the T_m at the midpoint of the unfolding transition. This is now clarified in the legend for Fig. 4C:

“Thermal denaturation of FL-FLuc ($T_m = 42$ °C) and isoNs ($T_m = 50.5$ °C). Data were fit to sigmoidal functions in Graphpad Prism.”

We did not fit the data to a kinetic model since they are not from a time course, but rather measurements at different temperatures. We could not fit a thermodynamic model, since the system is not at equilibrium. As shown below for isoNs (and typical for most proteins), the thermal denaturation was not reversible.

- line 237

Where in the figure is the partially folded population evident?

We had initially mentioned a partially folded population because the fit of the spectra to two populations was imperfect. In retrospect there is no clear evidence for an additional population, and we have not been able to convincingly fit the data to 3 Gaussian distributions. We have removed this claim from the revised manuscript.

- line 249

Do the data demonstrate an alteration in the conformational ensemble? Or is this simply a change in population between two distinct states (folded and unfolded)? Is there any evidence supporting the presence of multiple conformational states revealed by the data?

This is perhaps an issue of semantics: are unfolded and folded populations not part of the conformational ensemble? The bimodal spectra do support co-existence of unfolded and folded populations. To avoid confusion, we have reworded the sentence as follows:

“Together, these data show that the distribution of states populated by nascent FLuc is altered by synthesis of a C-terminal unstructured sequence, which shifts the equilibrium towards the unfolded state and destabilises pre-formed domain interfaces.”

- FigS9a

It would be helpful to show the two interface regions with dashed boxes as in main figures

Done.

- line346

Previous studies on the bacterial ribosome (e.g., reference 26) suggest C-terminus-mediated destabilisation of the nascent chain on bacterial ribosome. Can the ‘insulation of entropic destabilisation’ observed on the bacterial ribosome in this study generalised?

This is a good point. We now add a more nuanced discussion, and refer to work from the Kaiser lab showing that Trigger factor can protect folded domains from unstructured C-terminal sequences (<https://doi.org/10.1016/j.molcel.2019.01.043>). We now write:

“The extent of entropic destabilisation would be expected to depend on the stage of translation and stability of the nascent domain. In cases where a substantial unstructured C-terminal segment is exposed outside the exit tunnel of the bacterial ribosome, Trigger factor has been shown to protect upstream folded domains from denaturation²³.”

- HDX-MS data (Fig3a,h, Fig5d, Fig6a)

Are there uncertainties associated with these data? How many experimental repeats were performed?

All HDX-MS experiments were repeated at least 3 times. Exact numbers of replicates are given in supplementary Data S3 (hRNCs) and S5 (bRNCs) and mentioned in the figure captions.

In the Methods section we now write:

“Data were collected in at least technical triplicate. The standard deviation associated with every differential HDX measurement is reported in Supplementary Data S3, S5, S6 and S7. Mean s.d. = 0.11 Da, interquartile range: 0.04-0.14 Da for hRNCs. Mean s.d. = 0.10; IQR: 0.03-0.13 for bRNCs.”

Fig. 3h shows the behaviour of several representative peptides for each region of interest. These plots show error bars corresponding to the standard deviation.

In the revised manuscript we have added error bars to Fig. 6a.

We tried adding error bars to Fig. 3a and 5c, but these plots are already quite complex, and with error bars become overly crowded and difficult to read. We think that showing the errors for key peptides in Fig. 3h is a good compromise. Fig 3h is also now updated with additional statistics (see response below related to the weak Δ Da for the Ns:N interface).

- Fib5ab Ex1 data

Can these experiments be performed under different conditions (e.g., varying temperature or pulse labelling times) to provide more detailed kinetic information on the dynamics between the two states?

In principle it is possible to determine the kinetics of interconversion between the 2 states by varying the labelling time. However, in our case we do not think that we will be able to achieve the necessary time resolution. Because the hRNCs are so dilute, the labelling is performed by buffer exchange using a spin column, which sets a lower limit on the labelling time (~3min). As can be seen in Fig 5a, at this time point approximately half of the population is already deuterated.

It is also possible in principle to determine thermodynamic parameters for the interconversion by varying temperature. The limitation in this case is that, unlike for bRNCs, the integrity of the hRNCs at different temperatures is yet to be established.

Another point to add is that the low yield of purified hRNCs makes these experiments a substantial (and expensive) undertaking – each triplicate time point requires a new hRNC purification from 1 litre of suspension HEK cells. We are certainly interested in quantifying peptide-level folding thermodynamics/kinetics in a future study.

- line183

Why was 0.5Da chosen as the cutoff? The Ns:N interface of T-RNC shows a weak Δ Da. How can this be justified as meaningful evidence of destabilisation of this region?

We appreciate this comment and have thought carefully about how to approach it. In the revised manuscript we no longer apply a specific Da cutoff value. Rather, we describe the specific changes in Da associated with each comparison, and add some notes to guide the reader's interpretation. The relevant section now reads:

“We next used HDX-MS to probe the conformation of FLuc on the ribosome. We measured peptide-resolved deuterium uptake of NCs, and used isolated (off-ribosome) full-length FLuc (FL-FLuc) as a reference for the native state (Fig 3A). The FLAG tag did not affect deuterium uptake of FL-RNC and was retained (Fig S5A). Sequence coverage of NCs was >83% and most peptides were detected across different RNCs, allowing quantitative comparison between states (Fig S5B and Data S3). HDX-MS data for each RNC are summarised in Fig 3A-G and representative peptides are shown in Fig 3H for quantitative comparison. Data were measured in triplicate, with an average s.d. = 0.11 Da across the entire dataset. Note that a difference in uptake of 0.5 Da would correspond to protection from exchange of a single amide hydrogen, considering an average back exchange of 50% (Data S3).

We found that peptides in 1/2Ns were deprotected by 2-6 Da relative to the same region in FL-FLuc, indicating that 1/2Ns was unfolded (Fig 3B). Ns was also globally unfolded, and peptides from this subdomain reached near-native levels of deuterium exchange (within 0.5 Da) only when a larger part of the N domain was synthesised (1/2N-RNC, Fig 3C,D). The interface between Ns and N remained deprotected by 1-3 Da until the N-domain was complete (N-RNC), stabilising the β -roll that connects the N domain to the extreme N-terminus of FLuc (Fig 3E). At the final stage of translation, prior to termination, the extreme C-terminal residues of FLuc are within the ribosomal exit tunnel and therefore unavailable to fold with the rest of the NC. In our T-RNC construct which mimics this step, C domain peptides were highly deprotected (up to 7 Da) relative to FL-FLuc, as expected. Surprisingly, the N domain β -roll and the Ns:N interface of T-RNC were also deprotected by 1-3 Da compared to RNCs mimicking earlier stages of translation (Fig 3F,H). Extending the C-terminus of FLuc from the ribosome on a 50 aa linker (FL-RNC) resulted in native-like folding of C-domain peptides, and recovered the stability of N-domain peptides to within 0.5 Da of FL-FLuc (Fig 3G). Together, these data resolve length-dependent local folding of partially-synthesised FLuc on the human ribosome (Fig 3I). We find that Ns folding is delayed relative to synthesis, and the NC is destabilised prior to translation termination.”

To complement this, we have now calculated p-values for key comparisons and added these to Fig. 3H. For the representative Ns:N interface peptides, the uptake of T-RNC is different from N-RNC with p-values of 0.0029 and 0.0048. See revised Fig. 3H, below.

Two additional points regarding the Ns:N interface in T-RNC: 1. The absolute change in Da is not particularly small - up to 1 Da. 2. The Δ Da is calculated from the centre-of-mass of bimodal spectra, which underestimates the size of the effect. The population shift is more obvious when looking at the spectra themselves (Fig. 5A).

- FigS2d

The cryoEM structure of Ns-RNC shows an interaction between the NC and eL39. Was this interaction observed in the crosslinking data?

We did not detect crosslinks between eL39 and the NC. This might be due to the fact that DSBU-reactive side-chains in eL39 (K/T/Y/S) are predominantly positioned towards the ribosome core/interacting with rRNA rather than lining the exit tunnel. Alternatively, access of DSBU to the narrow tunnel may be sterically disfavoured. The interaction between eL39 and the NC is supported by HDX-MS (Fig. 7).

- line210-219

It is not clear how the role of the ribosome in Ns folding is being explained using data from isolated proteins. The CD and HDX-MS data are derived from the dimer. Are these data intended to show that the protein cannot fold independently without the ribosome and self-assemble? Is it clear that the isolated dimer is formed by the same interface as in the ribosome-bound state?

Previous work showed that Ns folds cotranslationally (DOI: 10.1038/10754), and our data indicated that Ns folding required synthesis of some downstream elements in the N-domain. We therefore wanted to test whether Ns is capable of folding in isolation and should be considered an independent folding unit. We have reworded the introduction to this section of the manuscript as follows:

To test whether Ns behaves as an independent folding unit, we purified isolated Ns (isoNs, residues 1-190) off the ribosome (Fig S6A,B).

When comparing deuterium uptake of isoNs and Ns-RNC, we noticed that although they were highly similar overall, isoNs was protected at specific peptides. Protection was localised to residues that, in FL-Fluc, would contact the rest of the N domain (Fig. 4E). We therefore speculate that isoNs dimerises via the same interface residues.

This is only speculation, since we do not know the structure of dimeric Ns. The protein is not soluble at concentrations required for structural analysis using NMR/X-ray crystallography. To emphasise this, we now write:

"We speculate that IsoNs may therefore dimerise via the same interface to form a non-native intermolecular β -sheet."

Reviewer #3 (Remarks to the Author):

While many proteins fold efficiently when refolded from a denatured state, many at the same time are unable to do so. Protein folding in vivo starts cotranslationally and hence is widely believed to facilitate efficient folding of especially those proteins that are unable to refold efficiently in vitro. Many recent studies shed light on the general principles governing cotranslational protein folding, however, many details of this process remain poorly understood. New techniques for both measuring folding kinetics and detecting the conformations of partially folded intermediates on the ribosome emerged recently thus allowing to get important information on cotranslational folding trajectories of several single and multidomain proteins. In this manuscript, Pellowe and co-authors used cryo-electron microscopy and hydrogen/deuterium exchange-mass spectrometry to characterize the structure and dynamics of partially-synthesized intermediates of a model multidomain protein, Firefly Luciferase (Fluc), previously shown to fold cotranslationally. The authors used HEK293 cells to characterize folding intermediates of FLuc on the human ribosome and E. coli cells to further understand whether cotranslational folding intermediates differ between human and bacterial ribosomes. Homogeneously-stalled Ribosome-nascent chain complexes (RNCs) were obtained using an arrest-enhanced variant of XBP1u, or a bacterial arrest-enhanced SecM sequence, respectively. The authors found that FLuc nascent chain subdomains fold progressively during synthesis on the human ribosome, templated by interactions across domain interfaces and that the conformational ensemble of the nascent chain is tuned by its unstructured C-terminal segments, which keep interfaces between folded domains in dynamic equilibrium until translation termination. Interestingly, the authors found that in case of bacterial ribosomes, domain interfaces appeared to form early and remained stable during synthesis. The authors concluded that delayed domain docking may avoid interdomain misfolding to promote the maturation of multidomain proteins in eukaryotes. This is an elegant and comprehensive study providing important insights into the mechanism of protein folding in the cell.

We appreciate the positive overall assessment and thoughtful comments below.

I have only a few minor comments:

1. The authors write (page 7) that “Digestion of ½N-RNC resulted in weak accumulation of the previously-described ~22 kDa intermediate corresponding to Ns...” and then state that “Ns is not an independent folding unit, but is held on the ribosome in a folding-competent state until a larger part of the Rossmann fold is available”. Please note that Frydman et al (Nature, 1994) in their seminal publication not only observed the formation of this N-terminal intermediate, but also described the time-dependent nature of its formation during protein synthesis and hypothesized that during the course of translation it further engages in interactions with the rest of the synthesized protein. This earlier observation needs to be mentioned and discussed in a bit more detail.

Thank you for pointing this out. We now write:

“In summary, we find that Ns folds cotranslationally, in agreement with previous reports ^{21,35}. Our data are also consistent with the time course of Ns formation in synchronised translation reactions ³⁵. A discrete protease-resistant species corresponding to Ns was shown to form early, then disappear upon incorporation into full-length FLuc. By HDX-MS analysis of purified RNCs and isoNs, we further show that Ns is not an independent folding unit, but is held on the ribosome in a folding-competent state until a larger part of the Rossmann fold is available. The Ns:N interface stabilises later, when the entire N domain is synthesised.”

2. The authors revealed that human and bacterial ribosomes differentially affect the conformational ensemble of FLuc NCs. They have concluded that “different ribosomes can shape the conformational ensemble of a model NC.” I would be a bit more cautious with this conclusion and consider alternative explanations too. While different ribosome structures may certainly provide different folding environment for the emerging nascent chains (both inside and outside the ribosome tunnel), another important factor to consider (besides, also the different cellular environment discussed by the authors) is the local and global synthesis rates that are known to modulate protein folding in the cell (see e.g. Buhr et al, Mol. Cell, 2016, etc). It is my understanding that the same Fluc sequence was used to obtain RNCs in HEK293 cells and E. coli and hence due to different repertoire of the tRNAs (and different codon usage bias) in these systems, the differential translation kinetics of Fluc may have resulted in different (structural and functional) intermediates. Please note that Liu and colleagues previously showed that specific luciferase activity can be affected by altered dynamics of its synthesis (see Yu et al, Mol. Cell, 2016) and Kolb et al (JBC 2000) in an earlier study also demonstrated that specific luciferase activity may differ in prokaryotic and eukaryotic translation systems. Interestingly, an about 10-20% higher specific luciferase activity was observed in E. coli S30 system compared to wheat germ extract system (Kolb et al, JBC, 2000) back then. It was also shown that refolding of luciferase in E. coli and wheat germ cytosol proceeds with different efficiency (Kolb et al, JBC, 2000), suggesting that besides the difference in translation kinetics, differences in the cellular environment (mentioned by the authors) are important contributors to the process of luciferase folding. In case, I am reading Figs. S1B and S8D correctly, FL-RNCs have about 20% higher specific activity compared to FL-bRNCs. This difference additionally implies that there exists a different spatial arrangement of the ribosome bound chains between FL-RNC and FL-bRNCs in the two systems used. As shown by the authors, this conformational difference persists in the Fluc chains fully extruded out of the ribosome (Fig. 6). However, the lower specific activity in case of FL-bRNC suggests that the differential engagement of the Fluc C-terminus (required for protein activity, see Sala-Newby et al., Biochem. Biophys. Res. Commun., 1990) with the rest of the protein (C-domain was substantially deprotected in FL-bRNC compared to FL-RNC as revealed by the authors) may account for this difference. These considerations and issues related to Fluc activity need to be also discussed by the authors.

The relationship between codon usage/translation rate and folding is often difficult to interpret unambiguously, especially for an aggregation-prone protein like FLuc. Faster translation leads to higher protein levels, which can result in decreased yield if aggregation is a competing reaction. Nonetheless, we agree that translation kinetics should be discussed, and have added a new section to the Discussion, citing Yu et al.:

“Translation elongation is faster in bacteria than eukaryotes and it is possible that this also contributes to the difference in folding outcomes⁶⁰, which may persist even in stalled complexes at equilibrium. Indeed, previous work has noted that the yield of active FLuc is sensitive to codon usage, especially in regions encoding the Ns:N interface⁶¹.”

Regarding the specific activity of the FL-RNCs on the bacterial versus human ribosome, we are cautious about overinterpreting this rather small difference. It could for example result from small errors in the estimation of RNC concentration, or slight differences in the proportion of ribosomes occupied by nascent chains.

Regarding Kolb et al (JBC, 2000), the authors conservatively concluded that E.coli lysate, wheat germ extract and RRL produce active FLuc approximately equally efficiently, stating: “all three translation systems are almost equally effective in providing the proper folding of

the enzyme". However, Agashe et al ([https://doi.org/10.1016/S0092-8674\(04\)00299-5](https://doi.org/10.1016/S0092-8674(04)00299-5)) find that only 10% of FLuc synthesised in E. coli is active, compared to ~100% in yeast. They further show that FLuc folding in E.coli, but not yeast, is partially post-translational. In our experience these assays are highly sensitive to the specific reaction conditions, and we do not wish to wade into this debate. In the revised manuscript we write:

"Whereas the unfolded conformation was rarely sampled on the human ribosome and in isolated FL-FLuc, the same peptide was predominantly unfolded on the bacterial ribosome. The C-domain was nonetheless folding-competent, as FL-bRNC was enzymatically active (Fig S8D), although slightly less so than the equivalent FL-hRNC (Fig. S1B). Previous comparisons of FLuc biogenesis efficiency in bacteria and eukaryotes have arrived at conflicting conclusions^{29,51}. However, the observation that FLuc folding is partially post-translational in E. coli²⁹ is consistent with our finding that the bacterial ribosome interferes with folding of the C-domain of FLuc near the ribosome surface."

3. It is also my understanding that RNCs were purified using either the 3xFLAG-tag (in case of HEK293 cells) or the muGFP(green)-tag (in case of E. coli cells; as described previously by the authors (Wales et al, 2024)). Could this difference in the N-terminal tagging affect the initial events of luciferase folding in the different systems used? The authors found that the FLAG-tag does not measurably affect the conformation of FL-RNCs (Fig. S5), however they didn't provide any evidence regarding the muGFP(green)-tag. This issue also needs to be discussed. Obviously, the muGFP(green)-tag has been cleaved off after RNCs purification. However, before it has been cleaved off, it may have affected the folding of a (comparable in size) Ns domain, which is not an independent folding unit and due its dynamic nature may have been engaged in interactions with muGFP(green)-tag.

Although our previous study (Wales et al.) used an muGFP tag to isolated bacterial RNCS, in the present manuscript we used a 3xFLAG tag to match the human system. We realise that this was not clearly described in the Methods section. Now fixed.

Minor:

Figure 8. There is no D panel, D in the text should be replaced with C.

Fixed, thank you.

Reviewer #4 (Remarks to the Author):

The manuscript by Pellowe et al investigates how human ribosomes influence the folding of multidomain proteins during synthesis. Using mainly cryo-electron microscopy and HDX-MS experiments on a model 550 res two-domain protein FLuc, the authors effectively illustrate how delayed domain docking helps prevent misfolding, highlighting a key difference between eukaryotic and bacterial ribosomes. The experimental work appears thorough, and speaking mainly for the HDX-MS experiments, I believe these are performed well and the data is generally adequately analyzed and interpreted.

Overall, the paper provides new interesting insights into folding of multi-domain proteins and ribosome-specific folding mechanisms in eukaryotes. I find the work interesting and the results from the combination of the many techniques used (including cryo-EM, XL-MS, limited proteolysis, HDX-MS etc) convincing. I am positive towards publication provided the below comments are addressed.

We appreciate the positive assessment and constructive suggestions.

Comments:

Line 183 – the authors need to substantiate and support the assertion that peptides are natively folded only when they differed in D uptake from full-length FL by less than 0.5 Da. In general it is not clear to me how the authors approached error in their HDX-MS experiments.

We appreciate this comment and have thought carefully about how to approach it. In the revised manuscript we no longer apply a specific Da cutoff value. Rather, we describe the specific changes in Da associated with each comparison, and add some notes to guide the reader's interpretation. The relevant section now reads:

“We next used HDX-MS to probe the conformation of FLuc on the ribosome. We measured peptide-resolved deuterium uptake of NCs, and used isolated (off-ribosome) full-length FLuc (FL-FLuc) as a reference for the native state (Fig 3A). The FLAG tag did not affect deuterium uptake of FL-RNC and was retained (Fig S5A). Sequence coverage of NCs was >83% and most peptides were detected across different RNCs, allowing quantitative comparison between states (Fig S5B and Data S3). HDX-MS data for each RNC are summarised in Fig 3A-G and representative peptides are shown in Fig 3H for quantitative comparison. Data were measured in triplicate, with an average s.d. = 0.11 Da across the entire dataset. Note that a difference in uptake of 0.5 Da would correspond to protection from exchange of a single amide hydrogen, considering an average back exchange of 50% (Data S3).

We found that peptides in $\frac{1}{2}$ Ns were deprotected by 2-6 Da relative to the same region in FL-FLuc, indicating that $\frac{1}{2}$ Ns was unfolded (Fig 3B). Ns was also globally unfolded, and peptides from this subdomain reached near-native levels of deuterium exchange (within 0.5 Da) only when a larger part of the N domain was synthesised ($\frac{1}{2}$ N-RNC, Fig 3C,D). The interface between Ns and N remained deprotected by 1-3 Da until the N-domain was complete (N-RNC), stabilising the β -roll that connects the N domain to the extreme N-terminus of FLuc (Fig 3E). At the final stage of translation, prior to termination, the extreme C-terminal residues of FLuc are within the ribosomal exit tunnel and therefore unavailable to fold with the rest of the NC. In our T-RNC construct which mimics this step, C domain peptides were highly deprotected (up to 7 Da) relative to FL-FLuc, as expected. Surprisingly, the N domain β -roll and the Ns:N interface of T-RNC were also deprotected by 1-3 Da compared to RNCs mimicking earlier stages of translation (Fig 3F,H). Extending the C-terminus of FLuc from the ribosome on a 50 aa linker (FL-RNC) resulted in native-like folding of C-domain peptides, and recovered the stability of N-domain peptides to within 0.5 Da of

FL-FLuc (Fig 3G). Together, these data resolve length-dependent local folding of partially-synthesised FLuc on the human ribosome (Fig 3I). We find that Ns folding is delayed relative to synthesis, and the NC is destabilised prior to translation termination.”

To complement this, we have calculated p-values for differences in uptake that are critical for the key conclusion that NCs are destabilised late in translation. See Figure 3H of the revised manuscript, reproduced below. However, we do not want to overemphasise the rigour of this approach. P-value cutoffs (e.g. <0.05) are subject to debate, and tests based on a small number of technical replicates are not statistically powerful. Moreover, effect size (i.e. the absolute change in Da) is important to consider when interpreting differences as biologically meaningful. Importantly, key points in the manuscript are substantiated by orthogonal experiments (e.g. limited proteolysis and F5M labelling).

To give the reader a general idea about technical reproducibility/precision of the HDX measurements, we now write the following in the Methods section:

“Data were collected in at least technical triplicate. The standard deviation associated with every differential HDX measurement is reported in Supplementary Data S3, S5, S6 and S7. Mean s.d. = 0.11 Da, interquartile range: 0.04-0.14 Da for hRNCs. Mean s.d. = 0.10; IQR: 0.03-0.13 for bRNCs.”

Line – 235 The writing on the EX1 kinetics is a bit confusing. Surely, for all proteins folded and unfolded conformations are populated to some extent. What defines EX1 kinetics is that the conversion between these states is slow on the HDX timescale. Please rewrite this part. Also, can the authors say something about what are the likely parts in each peptide that have a slow kcl? Does this make sense structurally? More discussion would add to this part.

We now write:

“Destabilisation of the N-domain in T-RNC was primarily localised to the β -roll and Ns:N interface (Figure 3A,F,H). Mass spectra for peptides in these regions displayed EX1 kinetics and were multimodal, indicating that a subset of residues within the peptide is slow to refold relative to the timescale of H/D exchange (Fig 5A,B). The peptides map to similar structural motifs, consisting of two anti-parallel β -strands and a connecting loop. Since the loops pack against structured elements at the Ns:N interface and β -roll:N interface in native FLuc, they are likely to be substantially deprotected if these interfaces are not formed.”

Line 722 – I am assuming the LC used is an UPLC?

Fixed, thank you.

Line 750 – since the authors use HDMS_e to identify peptides they should make sure that each peptide is identified in several replicate HDMS_e experiments i.e 2 out of 3 or 3 out of 4 – to add further confidence to identification. MS_e is notorious for misidentifications if such a replicate validation threshold is not used.

We did indeed do this. The exact number of undeuterated replicates varied from 3 to 8, and we kept only peptides that were found more often than not (i.e. 2 out of 3, etc). We also cross-checked the peptide identifications across RNCs, since many peptides are expected to be detected in different NC lengths. In addition, we used the initial peptide list to search a “decoy” empty ribosome dataset in DynamX, and discarded any peptides that were found there.

We have now explained our procedure in the Methods section:

“Peptides were identified from HDMS^E analyses of 3-8 replicate undeuterated control samples across each RNC, off-ribosome FLuc and empty 80S ribosomes using PLGS 3.0.3 (Waters).”

“Only peptides that were identified in the majority of undeuterated replicates, as well as in off ribosome FLuc, were retained. In addition, the resulting FLuc peptide list was searched in DynamX against control data collected for empty 80S ribosomes. Putative FLuc peptides that were detected in empty 80S were removed.”

Reviewer #2 (Remarks to the Author):

Review of Pellowe et al.

Pellowe et al. provide a comprehensive experimental treatment characterizing a nascent chain emerging from the eukaryotic 60S ribosome by taking advantage of a eukaryotic-specific stalling sequence, XBP1u+. Their manuscript leverages XL-MS, HDX-MS, Cryo-EM, biochemical labeling methods to interrogate reconstituted stalled ribosome nascent chain complexes (RNCs) of firefly luciferase (FLuc). Two primary observations are noted: Firstly, that the cotranslational folding of FLuc is not strictly vectorial, with the Ns domain remaining largely unfolded until docking onto the N domain is possible; and Secondly, that the human ribosome permits a different set of folding trajectories than the bacterial one, which allows the C-domain to fold more cotranslationally, whereas it remains unfolded on the bacterial ribosome and only completes folding post-translationally.

We thank the reviewer for their time, and for recognising the scope of the experiments.

We would like to note here that the reviewer’s summary omits the major finding of the study, stated in the title and prominent in several figures. Namely, that the Ns:N interfaces are destabilised late in translation on the human ribosome. This observation is also the most important point of difference compared to the bacterial ribosome.

The strength of the manuscript lies in the amount of experimental evidence generated to interrogate the system at hand. On the other hand, this reviewer could make the case that the primary claims are not necessarily surprising and/or may be specific to the case of luciferase, a notoriously complex-to-fold protein with a highly unusual and nested domain structure. The most interesting claim about fundamental differences between the human and bacterial ribosome vis-à-vis folding multi-subunit proteins is interesting but perhaps the one that would require additional evidence to fully substantiate.

The reviewer makes two points. First, that the main claims are not surprising given the structure of FLuc. Second, that the results are not generalisable because FLuc has a highly unusual topology. We address each of these points below, specifically in response to comment #6 and #8.

We argue that our results, accurately described, do not follow obviously from inspecting the structure of FLuc. We also explain that proteins with discontinuous domain architectures, like FLuc, are in fact quite common. More generally, we think that difficult-to-fold proteins are exactly those which should be the focus of cotranslational folding studies, since their maturation is not readily explained by in vitro refolding data.

I'll lay out my comments below.

1. "Multidomain proteins are ~3 times more frequent in eukaryotic compared to prokaryotic proteomes." This statement seems incorrect unless the author is invoking a very specific definition of multidomain. In *E. coli* MC4100, there are 926 proteins without an annotated domain, 2024 with one, and 1743 with more than one. Hence 37% of *E. coli* proteins have more than one domain. So it's not possible for eukaryotes to be 3-fold more, as that would be greater than 100%. Please check.

This is our mistake, thank you for catching it. The figure we quoted was for proteins with 3 or more domains, which are 3x more frequent in eukaryotes. The exact frequency of multidomain proteins in eukaryotes varies from ~70-80% of the proteome depending on how it is estimated (e.g. DOI: 10.1126/science.1085371; <https://doi.org/10.1038/nrm2144>). In the revised manuscript we now simply state:

"Multidomain proteins are more frequent in eukaryotic compared to prokaryotic proteomes."

2. "FLuc consists of 550 residues across two domains." It seems like it has four, based on the drawing in Fig 1A

The domain plot in Fig 1A indicates the 2-domain architecture and domain boundaries as described in the text. The N domain contains 3 folds, which are coloured differently. This is now clarified in the Figure legend.

3. "The P-site tRNA is poorly resolved due to high flexibility in the absence of the small subunit" Is this typical for isolated 60S subunits? Typically it is resolved in many structures of loaded ribosomes. Oftentimes it is the only tRNA that is resolved.

The p-tRNA is resolved, but poorly compared to the rest of the ribosome. We have revised the text to make this clear:

"The P-site tRNA is poorly resolved relative to the ribosome, due to high flexibility in the absence of the small subunit (Fig 2A and S4E)."

This behaviour is typical for isolated 60S unless bound by an additional factor, and logical since tRNAs are stabilised by interactions with both subunits.

4. "The P224V substitution positions one of the side chain methyl carbons within 3.9 Å of the aromatic ring of 28S rRNA residue U4556, compatible with a CH – π interaction." This is too far to be a significant electrostatic interaction. In general, the discussion about these mutations is not too compelling. Maybe could be more interesting if authors could compare the structure of XBP1u and XBP1u+ directly.

CH – π interactions are a type of hydrogen bond for which the electrostatic contribution is small, and 3.9 Å is well within the typical range of distances (<https://doi.org/10.1039/B718656H>).

The structure of XBP1u and XBP1u+ are directly compared in Fig S4H.

5. “The same residues make up an unstructured coil in native FLuc, suggesting that confinement in the eL39/rRNA groove may contribute to stabilising non-native secondary structure in the NC.” The idea that alpha helices can transiently form in the exit tunnel only to be rendered non-alpha later on was pointed out by Marina Rodnina’s team on cold shock protein.

We have now pointed this out in the revised manuscript:

“The same residues make up an unstructured coil in native FLuc, suggesting that confinement in the eL39/rRNA groove may contribute to stabilising non-native secondary structure in the NC. A similar phenomenon was previously noted for the all-β protein CspA, which showed density in the exit tunnel consistent with an α-helix.”

6. “The interface between Ns and N remained deprotected until the N-domain was complete.” “We find that Ns folding is delayed relative to synthesis.” The authors treat this as an important point, and it is one of the major findings of this paper, but in some ways, this is a very logical result especially because FLuc has such a complex nested domain structure. Even if we look at the author’s own domain map of Ns, it would be impossible for this to be folded until all of N is synthesized because Ns itself is not really “a domain” per se, itself being an amalgam comprising of 1/3 of Rossmann 1 and 2/3 of Rossmann 2.

In fact, Ns does fold before all of N is synthesised. Our data show that Ns requires synthesis of only a fraction of N (see ½N-RNC) to fold, but stable docking of Ns against the N domain occurs later (Fig. 3). This is supported by our limited proteolysis data (Fig. 4) and consistent with prior work (DOI: 10.1038/10754).

Thus, our conclusions do not follow obviously from inspecting the structure of FLuc. The results are surprising in two respects.

1. Ns folds when the remainder of N is partially synthesised.
2. The Ns:N interface, stable when N is complete, is then destabilised during synthesis of the C-domain (see data for T-RNC; Fig. 3 and 5).

7. “with reduced α-helix and increased β-sheet content compared to Ns in the context of FL-FLuc” I’m not really sure how CD could tell you this. CD of IsoNs can tell you about the 2° structure of IsoNs. CD of FL-FLuc can tell you about the 2° structure of FL-FLuc. But the structure of Ns in the context of FL-FLuc? This would presumably require some “contrasting” or “backgrounding” method to make the contributions of IsoNs stand out above the remainder of the protein... which one could do with, say, SANS or NMR (isotope editing), but I don’t see how you could learn this from a CD. Maybe this is just worded in a way I am not following.

Secondary structure of Ns in the context of FL FLuc was not measured by CD, but calculated using the crystal structure of the protein. We have now rephrased the legend of Fig. 4 to make this clear:

“Spectral deconvolution showed 4.4% α-helix and 38.5% β-sheet for isoNs. Cf. Ns (residues 1-190) in the context of FL-FLuc, calculated based on the crystal structure (PDB: 1LCI): α-helix = 24.7% and β-sheet = 23.5%.”

8. “Ns is not an independent folding unit, but is held on the ribosome in a folding-competent state until a larger part of the Rossmann fold is available.” Reiterating point 6. Whilst I agree with this, it is maybe not surprising given the nature of the domain topology. I think this point deserves to be made and I agree with it. But I also think it must be qualified with the fact that luciferase is just a very unusual case.

In this point and above, the reviewer characterises FLuc as an unusual protein due to its “nested domain structure”. In fact, ~28% of multidomain proteins have discontinuous domains (doi/abs/10.1002/pro.5560070202).

Moreover, even if FLuc were particularly unusual, we would argue that proteins with complex architectures are exactly those which should be the focus of cotranslational folding studies, since their maturation is not readily explained by in vitro refolding data.

9. As a stylistic point I don't like how so many plots are missing numbers, like how mass spectra never have masses on x-axis, Delta mass plots never show residue numbers on x-axis, etc.

We have now added m/z values to the x-axes of mass spectra.

The peptide uptake plots are too densely packed to include legible labels for each peptide. Linear residue numbering is not possible, since many peptides are overlapping. We would prefer to keep the current format which indicates different FLuc regions on the summary plots, and includes detailed additional plots/spectra for peptides of interest.

10. “The Ns:N interface was not destabilised in T- bRNC, indicating that the unfolded conformation is sampled much less frequently than on the human ribosome.” If this claim is based on the size of the colour bar on Fig 6A in the dotted boxed region called Ns:N interface, then this does not look like a particularly substantial effect/difference, and without any statistics, it is hard to evaluate if the effect is statistically significant. From inspecting the Fig 6A the only effects that look large are the negative feature in the beta-roll for T-RNC and the positive feature in C-domain for FL-RNC.

We have now added error bars corresponding to standard deviation to Fig. 6a.

Also, note that these data points are calculated from the centroid of bimodal spectra. The difference between bRNCs and hRNCs in this region is even clearer when comparing the mass spectra – see Fig. 5a and 6c.

11. “The bacterial ribosome therefore interferes with folding of the C-domain of FLuc near the ribosome surface.” DO we think this is exclusively because of differences in the ribosome? Could the bacterial ribosome be carrying a trigger factor chaperone? Is there a way to rule out that possibility?

The bRNCs were purified from Trigger factor-knock out *E. coli* (Methods) and do not contain any major contaminants. The composition of the bRNCs was assessed by SDS-PAGE (Fig. S8) and quantitative proteomics (Data S4).

We now clarify this in the Results section of the revised manuscript:

“bRNCs were purified from Δ *tig* cells to avoid copurifying Trigger factor, and RNC composition was verified by MS (Data S4).”

12. The most interesting claim in the paper is the idea that the bacterial ribosome exit tunnel fundamentally destabilizes folding near the exit tunnel thereby explaining why C-domain is more unfolded in the FL-bRNC but more folded in FL-hRNC. One important caveat to point out is that C-domain in T-hRNC is also highly unfolded, which shows evidence that the folding of the C-domain is exquisitely sensitive to how much “slack” it has coming out of the exit tunnel. Hence it seems possible that some of this difference arises from the difference in the stalling sequences and the fact that FL-RNC has 12 aa difference in length between these two systems. Nevertheless, the specific hypothesis that the loops of uL23 and uL24 are specifically responsible for this destabilization is an interesting one, and the best

experiment to test this idea is not to compare bRNC to hRNC, but rather to mutate out the loops of uL23 and uL24 of bRibos, so that way there can be an apple-to-apple comparison using the same stalling sequence.

The C-domain of T-hRNC is indeed unfolded, but this is not a question of “slack” at the C-terminus. Rather, it is simply because part of the C-domain is unavailable for folding - it is still in the exit tunnel/replaced by the stalling sequence (See Fig. 1B,C).

Although FL-hRNC is longer than FL-bRNC, T-hRNC and T-bRNC are the same length and can be compared. We show that the distribution of NC conformations is quite different in these two cases, favouring the unfolded state on the human ribosome (Fig. 6).

Removing the loops of bacterial L23/L24 is an interesting idea. Unfortunately, this would not result in a “humanised” ribosome, since eL39 is absent. Rather, it would only serve to further widen the exit tunnel, which is already wider in bacterial compared to human ribosomes. We are certainly interested in systematically testing the determinants of folding on human vs bacterial ribosomes, but we think that this would be better suited to a separate study.

13. “Multidomain proteins often fold more efficiently in eukaryotes compared to bacteria”
There may be isolated examples of this, but I don’t think we know it as a general fact.

We agree. Although this has been shown in a number of cases where it was explicitly tested (e.g. DOI: 10.1038/41024; <https://doi.org/10.1016/j.jmb.2005.08.052>), we are not aware of any proteome-wide studies. We have now rephrased this statement as follows:

“Indeed, several multidomain proteins have been shown to fold more efficiently in eukaryotes than bacteria”.

14. “Delayed folding of C-terminal domains may contribute the general tendency of multidomain proteins to complete folding post-translationally when expressed in bacteria^{22,29}.” This can also be because translation is faster in bacteria.

We now write:

“Together with differences in elongation kinetics, delayed folding of C-terminal domains may contribute to the general tendency of multidomain proteins to complete folding post-translationally when expressed in bacteria.”

Reviewers' Comments:

Reviewer #1 (Remarks to the Author):

It is clear the authors have improved the text in various areas. However, there are still a few points that remain unconvincing to me.

First, regarding the cryoEM data: the Q-score based assessment for model fitting looks good and justifies the structural modelling. However, I still cannot see that these structures support their main conclusion well. The two reasons they offer – demonstrating the integrity of one of the RNCs and insights into the stalling mechanism – do not seem particularly interesting or coherent in supporting the central findings of the manuscript. As they acknowledge, density for the nascent chain (NC) outside exit is challenging to capture, even when the NC is well folded. Contrary to their response, however, NC density outside the exit has been reported in several studies (e.g., Zhang et al. 2015 Elife; Tian et al. 2018 PNAS), though these were based on bacterial ribosomes. Even if the T-RNC or N-RNC lack NC density outside the exit as they suggest, the density inside the tunnel would be more readily visualised and more informative than what is shown for the Ns-RNC in Figure 2 – especially since the manuscript's conclusions primarily rely on HDX data of 1/2N-RNC and T-RNC.

We appreciate the reviewer's comments on the quality of the structural modelling. Regarding the usefulness of the CryoEM structure: our study reports a new way to prepare human RNCs, and the structure supports the rigour of subsequent experiments that rely on these complexes. However, since cryoEM resolves folded states, it is not appropriate for describing folding intermediates. Even the observation of NC density outside the tunnel would be of limited benefit, unless it can be used to produce a structural model. For this reason, the central findings of our study are based on HDX-MS analyses, corroborated by XL-MS, limited proteolysis, and Cys labelling.

It is useful to see statistical analysis added to some figures. However, the key HDX figures (Fig 3a and 5c) would better include uncertainties for each data point. These data underpin the main conclusions, and I do not think displaying uncertainties will make the figures overly complex. Rather, including error bars is essential for allowing readers to interpret and assess the data in a more rigorous and transparent manner.

We agree and have now added standard deviations to Figures 3A, 5C and Extended data 9A.

Other responses appear reasonable.

Reviewer #2 (Remarks to the Author):

I still think the paper should be accepted eventually and I appreciate the authors comments to my first round of review. Here are some additional comments based on a close second read and my response to some of their responses in a few cases where I really do think a change is required/merited.

We appreciate the feedback and have endeavoured to thoroughly address the new and original points, which we hope will bring the review process to a timely conclusion.

1. "FLuc refolds extremely slowly ($t_{1/2} \sim 75$ min)." Slowly and inefficiently, I would add. (Slowly by itself implies that it eventually gets to 100% just slowly, but really even after 75 min, the corresponding author showed in the cited paper the final recovered activity is 10-50%).

We now write:

"FLuc refolds extremely slowly ($t_{1/2} \sim 75$ min) and inefficiently (yield 10-50%) from denaturant in vitro, and populates aggregation-prone intermediates."

2. Extended Data Fig 1B: Any comment why FL-RNC has a 3x activation relative to isolated FL-FLuc?

We previously addressed this point made by reviewer 1 in the first round of revisions. Our response is reproduced below.

FLuc enzyme activity is coupled to domain dynamics, and it is not impossible that the environment near the ribosome surface could influence its activity. However, we cannot exclude that this difference simply arises from errors in calculating sample concentration, particularly since the concentrations of the RNC and off-ribosome samples are determined in different ways (A_{260} vs A_{280}). We are therefore cautious about overinterpreting the difference in activity.

The legend for Extended Data Fig. 1 contains the following comment:

"Note that the apparent difference in enzyme activity between FL-FLuc and FL-RNC may result from differences in concentration, which is estimated using different approaches for off-ribosome and RNC samples (see Methods)."

3. Line 136 and 139: Extended Data Fig SA, missing a number

Fixed, thank you.

4. I would appreciate residue numbering in Fig 3A

We have updated figures 3A, 5C and Extended data Fig. 9A to label peptides on the x-axis. So that the axes are readable, we labelled every 2nd peptide in Fig 3A and ED Fig 9A, and every 3rd peptide in Fig 5C. All peptide-specific uptake data are also supplied in supplementary files.

5. Average back exchange of 50%. This seems higher than normal for HDX-MS. Could authors comment why? Is this due to a technical limitation associated with these complicated samples?

Correct. Our workflow is adapted to the complexity of the samples, requiring “offline” pepsin digestion and a long HPLC gradient, both of which contribute to increased back exchange.

In general, different HDX workflows are associated with different levels of back exchange. This does not affect data interpretation, provided that the data are either collected under identical conditions (as we do) or corrected for back exchange using additional control samples.

6. In general, I’m somewhat confused about the length of 1/2N relative to Ns. Based on Fig 1B, this is only slightly longer than Ns, but then in Fig 3, it would seem 1/2N is much longer than Ns. I guess this is important for understanding how much more NC needs to be synthesized for Ns to fold/dock

In addition to the main text, we have now included the construct boundaries in the legend for Fig 1B, which reads:

“Design of FLuc stalling constructs. 1/2NS, residues 1-123; NS, 1-208; 1/2N, 1-388; N, 1-458; T, 1-528; FL, 1-550+50aa. Positions at which XBP1u+ was inserted are indicated by red arrows.”

7. Another place where this occurs: “that Ns is near-natively folded at this chain length.” Please specify the chain length

We now write:

“...that Ns is near-natively folded after synthesis of residues 1-388...”

8. In their response to my first round of review, I appreciated the authors’ point that co-translational folding is perhaps most interesting and relevant in the context of proteins with complicated domain topologies of which FLuc is a good model, but this discussion did not seem to make its way into the text. I think the authors should be more explicit, perhaps in their discussion, that the findings they report here (such as Ns folding only when a larger part of N is synthesized, and Ns:N destabilization upon synthesis of C-domain) are possibly specific to proteins like luciferase with nested domain architecture. It may be that vectorial domain folding would be more common for multi-domain proteins that are of the more simple tail-to-head nature. We don’t need to argue about which one is more common. The point is just that some multidomain proteins are tail-to-head and some are discontinuous. FLuc is archetypal of the latter. And hence the results of this important paper speak to that subcase of multidomain protein.

Regarding the choice of model, in the introduction we now write:

“To study how eukaryotic ribosomes shape multidomain protein folding, we focused on Firefly Luciferase (FLuc) as a nascent chain model. FLuc is a conformationally labile 2-

domain protein with a complex topology, the efficient biogenesis of which strongly depends on the cellular environment.”

We now mention the relationship between domain discontinuity and Ns folding in both the results and discussion section, as follows:

“...we further show that Ns is not an independent folding unit, but is held on the ribosome in a folding-competent state until a larger part of the discontinuous Rossmann fold is available...”

“As a result of the discontinuous architecture of the Rossmann fold, the Ns subdomain folds only when a larger part of the N-domain is synthesised.”

In the discussion we now mention that some of the phenomena we describe may not apply to multidomain proteins with continuous domains.

“How do multidomain proteins fold efficiently in vivo? Previous work has shown that interdomain misfolding can be circumvented by separating domain folding on the ribosome²², or rescued by the Hsp70 chaperone system^{3,34,57}. Here, we suggest two additional factors that optimise de novo folding of multidomain proteins (Fig. 8B). First, (sub)domain folding is triggered when the native domain interface first becomes available. This might allow the correct interface to form before additional sequence, encoding competing interactions, is synthesised. Such a mechanism may be less relevant for simpler multidomain proteins with continuous domain architectures.”

We have no evidence or reason to believe that interdomain interface destabilisation is specific to proteins with discontinuous domain arrangements (the C-domain is continuous with the N-domain), and would prefer not to suggest this.

9. In the author’s response to my first round of review, they wrote “The N domain contains 3 folds, which are coloured differently.” I have never heard of this notion of a domain with multiple folds. A domain is not an arbitrary structural unit but an evolutionarily conserved unit that we can match with an HMM (e.g., a beta-roll, a Rossmann). Hence, FLuc is – by that definition – a 4-domain protein in which the first 3 exhibit high levels of discontinuity. Inspecting the uniprot entry’s InterPro shows it as having four domains. Hence, Ns is not really a domain, but a structural unit that contains two halves of two different domains. I think that this description is more correct. And it really does explain why Ns would not be expected to be very well folded by itself until a larger chunk of the protein is available (namely 1/2N). I would ask the authors in Figure 1 to make this clearer by coloring FLuc in 3-D in two different ways, in one way that is based on the bioinformatic domains (as it is shown in the linear representation above it) and in a second way in which Ns and N (which are structural units rather than strict domains) are colored. I would also ask that Ns be explicitly referred to as a composite of two half-domains, to provide more context to why it does not fold as an independent unit and requires elements only present in 1/2N to be stabilized. This does not take away from the authors interesting discoveries that the entirety of N needs to be present for Ns to dock, and that Ns then undocks at T-RNC to redock again at FL-RNC. But I will hold my ground that it is not surprising that Ns is unstructured at Ns-RNC, and I think this simple point should be made explicitly by the authors so that the truly interesting results can be more greatly appreciated by their future readers.

In the revised manuscript we have expanded the introduction to better explain the domain annotation of FLuc. We tried to take a balanced approach that respects both the original description of the structure of FLuc, and bioinformatic/experimental evidence for alternative (sub)domains.

The original report of the structure of FLuc (doi: 10.1016/S0969-2126(96)00033-0) described it as a 2-domain protein, with the N-domain consisting of 3 subdomains (the 2 Rossmann folds and β -barrel roll). We have kept this nomenclature, but additionally now mention the current InterPro annotation and refer to prior literature suggesting that Ns is an independent folding unit.

At the reviewer's suggestion we have also included a new panel in Figure 1 (new Fig 1B) which shows Ns in a separate colour. We did this by colouring the structure according to InterPro/TED (<https://www.ebi.ac.uk/interpro/protein/UniProt/P08659/>; <https://ted.cathdb.info/uniprot/P08659>), which identifies 3 domains in total and distinguishes Ns as a separate domain.

We now write:

“FLuc consists of 550 residues and was initially described to contain two domains: a large N-terminal domain containing the active site, connected via a flexible linker to a smaller C-terminal domain (Fig. 1A)³⁸. The N-domain has a complex topology and can be subdivided into three discontinuous subdomains with distinct folds: two inverted Rossmann-like folds and a β -barrel roll (β -roll). Despite containing only part of each Rossmann fold, the N-terminal part of the N-domain (Ns, residues 15-190) was shown to form a protease-stable substructure during translation but not refolding from denaturant, suggesting that it may behave as an independent folding unit²¹. Consistent with this, InterPro divides FLuc into three consensus domains with residues 54-186 constituting an independent domain (CATH Superfamily 3.40.50.980) (Fig 1B).”

We agree that the architecture of the Rossmann folds explains why Ns folding is delayed. In the section describing Ns folding we write:

“In summary, we find that Ns folds cotranslationally, in agreement with previous reports^{21,35}. Our data are also consistent with the time course of Ns formation in synchronised translation reactions³⁵. A discrete protease-resistant species corresponding to Ns was shown to form early, then disappear upon incorporation into full-length FLuc. By HDX-MS analysis of purified RNCs and isoNs, we further show that Ns is not an independent folding unit, but is held on the ribosome in a folding-competent state until a larger part of the discontinuous Rossmann fold is available (Fig 1A). The Ns:N interface stabilises later, when the entire N domain is synthesised.”

10. Related to the above point, it seems like the “N-domain” sometimes is used to refer to the parts of the first 3 domains excluding Ns (as in Figure 1C), but sometimes it is used to refer to the first 3 domains including Ns (as in Figure 1A). Could a consistent nomenclature be used? Perhaps what is sometimes called N-domain could be called N2, so that Ns + N2 = N? The nomenclature just seems awkward to say that at N-RNC, there is both Ns and N.

“N-domain” always refers to the complete N-domain as originally annotated, of which Ns is a fragment. We realise that the schematics in Figures 1C and 3I may have been confusing, and are now changed for clarity.

11. Figure 8 does not have a panel D (typo in the caption)

Fixed, thank you.

Reviewer #3 (Remarks to the Author):

The manuscript by Pellowe and co-authors has been revised. Additional information has been added, which was missing in the original version of the manuscript. In sum, I feel that the authors have responded to the majority of the previous concerns and the manuscript has been substantially improved. I have no additional comments. I believe the manuscript provides sufficiently novel and interesting data to warrant its publication in NSMB.

We thank reviewer 3 for their constructive comments which substantially improved the manuscript.

Reviewer #5 (Remarks to the Author):

I have primarily reviewed the HDX-MS data in the submitted manuscript, and the authors have done an excellent job of responding to reviewer's comments. They now have included error bars on the graphs wherever possible and have included standard deviations for all the data in supplementary tables. Given the scarcity of sample, it is acceptable that the authors only present 3 technical replicates and do not have a biological replicate. The differences they observe are well outside of the standard deviation. It is also acceptable that they suggest using a cut-off of 0.5D even though the standard deviation is smaller (0.11 D). The differences are all much larger than these errors and the conclusions based on the HDX-MS data are robust. Overall the combination of all the different experiments including structure determination, limited proteolysis, HDX-MS and the use of various constructs is comprehensive. The results are super interesting for understanding co-translational folding on human ribosomes and publication is strongly recommended.

We thank reviewer 5 stepping in to review the manuscript at this stage, and for their positive comments on the HDX-MS experiments.